# SocialMaze: A Benchmark for Evaluating Social Reasoning in Large Language Models

## Abstract

Large language models (LLMs) are increasingly applied to socially grounded tasks, such as online community moderation, media content analysis, and social reasoning games. Success in these contexts depends on a model's *social reasoning ability*—the capacity to interpret social contexts, infer others' mental states, and assess the truthfulness of presented information. However, there is currently no systematic evaluation framework that comprehensively assesses the social reasoning capabilities of LLMs. Existing efforts often oversimplify real-world scenarios and consist of tasks that are too basic to challenge advanced models. To address this gap, we introduce **SocialMaze**, a new benchmark specifically designed to evaluate social reasoning. SocialMaze systematically incorporates three core challenges: *deep reasoning, dynamic interaction, and information uncertainty*. It provides six diverse tasks across three key settings—social reasoning games, daily-life interactions, and digital community platforms. Both automated and human validation are used to ensure data quality. Our evaluation reveals several key insights: models vary substantially in their ability to handle dynamic interactions and integrate temporally evolving information; models with strong chain-of-thought reasoning perform better on tasks requiring deeper inference beyond surface-level cues; and model reasoning degrades significantly under uncertainty. Furthermore, we show that targeted fine-tuning on curated reasoning examples can greatly improve model performance in complex social scenarios.

## 1 Introduction

LLMs demonstrate significant capabilities across various domains, such as scientific discovery (Guo et al., 2023; Chen et al., 2025a) and medical applications (Zhou et al., 2024; Chen et al., 2025b). Most recently, they have been increasingly applied to socially grounded tasks, such as online community moderation (Kolla et al., 2024; Zeng et al., 2024; Qiao et al., 2024; Masud et al., 2024; Liu et al., 2024a), media content analysis (Chew et al., 2023; Liu et al., 2024b; Wang et al., 2025a), and social reasoning games (Li et al., 2023; Wei et al., 2025). The success of LLMs in these applications often hinges on their **social reasoning abilities**—the capacity to understand the social context, infer others' mental states, and make appropriate judgments based on this understanding.

While existing benchmarks effectively evaluate the general capabilities of LLMs (Chiang et al., 2024; Huang et al., 2025; Kosinski, 2024; Chen et al., 2024; Xu et al., 2025), benchmarks specifically designed to assess social reasoning abilities face significant limitations: 1) reliance on static scenarios lacking dynamic interaction (Sap et al., 2019b;a; Onoe et al., 2021; Demszky et al., 2020) , 2) presentation of overly sanitized information devoid of the noise, bias, or deception common in real social environments (Kosinski, 2024; Lin et al., 2020; Shapira et al., 2023; Nematzadeh et al., 2018) , and 3) tasks too simple to capture the deeper cognitive aspects of social inference(Forbes et al., 2020; Hendrycks et al., 2021; Tiwari et al., 2025). A few examples, SocialIQA (Sap et al., 2019b) primarily assess commonsense reasoning within simplified, predefined social contexts, testing basic understanding rather than complex, interactive inference. Similarly, Theory-of-Mind (ToM) benchmarks (Gandhi et al., 2023) often evaluate mental state inference based on static narratives and typically lack deceptive elements or informational uncertainty. Beyond benchmarks, some recent work has applied LLMs to strategic games such as Diplomacy (Bakhtin et al., 2022) and deduction games like Avalon and Werewolf (Wei et al., 2025; Light et al., 2023; Bailis et al., 2024; Xu et al., 2023). While these approaches attempt to place LLMs in dynamic environments, their evaluation

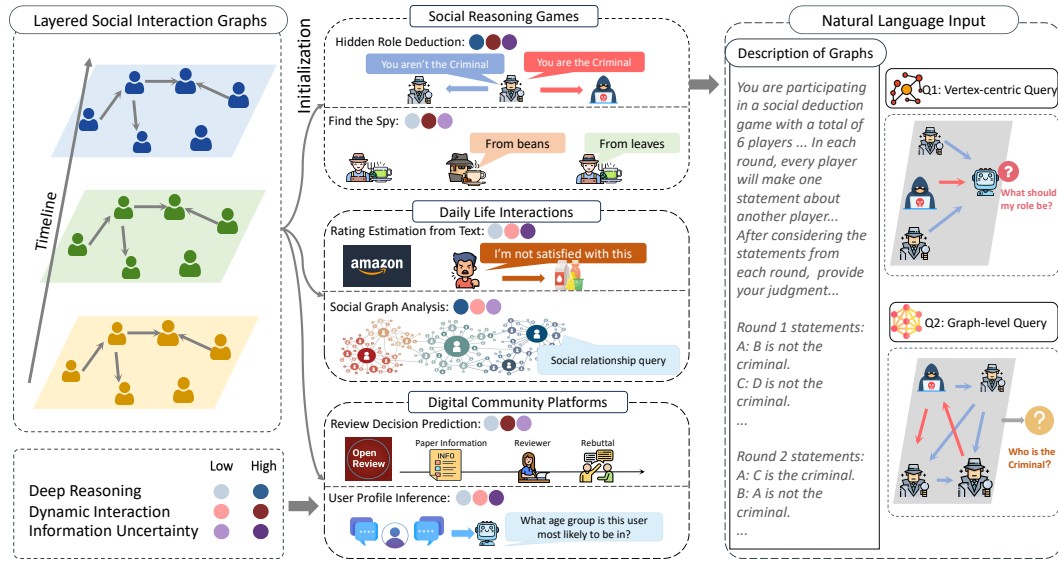

Figure 1: Overview of the SocialMaze Benchmark. All tasks are built upon (a) a time-aware modeling framework for complex social networks. Based on this template, we instantiate (b) 6 diverse task types, covering social reasoning games, daily life interactions, and digital community platforms. (c) One illustrative example of Hidden Role Deduction, including description of graphs along with vertex-centric and graph-level queries.

typically emphasizes task outcomes such as whether the model completes the task or outperforms competitors, rather than assessing whether the model genuinely engages in correct and coherent social reasoning. Success in such tasks does not necessarily indicate that the model truly understands the underlying social logic or reasoning process.

To address these limitations and enable a more holistic and rigorous evaluation, we argue that the assessment of social reasoning should explicitly incorporate three core aspects, which are key features of complex social reasoning tasks:

***Deep Reasoning***: Effective reasoning in social environments often requires going well beyond surface-level information and engaging in complex cognitive processes. These include carefully inferring others' latent mental states (such as intentions, beliefs, and motivations) (Premack & Woodruff, 1978; Wellman, 2014; Kunda, 1999; Frith & Frith, 2007), analyzing complex causal relationships between actions and outcomes, exploring counterfactual possibilities, and engaging in strategic thinking or mental simulation to anticipate future scenarios and plan accordingly (Gandhi et al., 2023; Roese, 1997; Huang et al., 2023; Zhang et al., 2024b). ***Dynamic Interaction***: Real-world social contexts are often characterized by iterative, interdependent exchanges. This demands that models continuously track the evolving context across multiple turns and dynamically adapt their reasoning and actions based on prior interactions and anticipated responses (Emirbayer & Mische, 1998; Pentland, 2014; Shoham & Leyton-Brown, 2008; Park et al., 2023; Chen et al., 2025c). Failure to do so leads to static or contextually inappropriate behavior. ***Information Uncertainty***: Social information landscapes are inherently noisy and unstable, with credibility varying greatly. They often contain misinformation, subjective biases, and intentional deception (Wang et al., 2018; Baldwin et al., 2013; Preiser et al., 2018; Lazer et al., 2018).

Based on these three principles, we present ***SocialMaze***, a benchmark designed to reflect the challenges posed by all three dimensions, as shown in Figure 1. It consists of six diverse tasks spanning three categories: Social Reasoning Games, Daily Life Interactions, and Digital Community Platforms. Each task intentionally varies the demands along the three dimensions. Our experiments with SocialMaze reveal several key insights into the capabilities of LLMs: Models with stronger chain-of-thought reasoning perform better on tasks that require deeper inference. We also observe that dynamic interaction affects performance in varied ways across tasks, and information uncertainty significantly hinders reasoning. Moreover, reasoning agents and workflows (Hu et al., 2024; Zhang et al., 2024a; 2025) offer limited gains in these social reasoning challenges, while targeted fine-tuning on curated reasoning examples leads to substantial performance improvements.

Table 1: Overview of SocialMaze task categories and key characteristics. Tasks vary along three key dimensions: the level of *Deep Reasoning*, the degree of *Dynamic Interaction*, and the extent of *Information Uncertainty*, each categorized as High or Low.

| Scenario | Task Category | Deep Reasoning | Dynamic Interaction | Information Uncertainty | Number of Instances |
|---|---|---|---|---|---|
| Social Reasoning Games | Hidden Role Deduction | High | High | High | 20,000 |
| | Find the Spy | Low | High | Low | 6,000 |
| Daily Life Interactions | Rating Estimation from Text | Low | Low | High | 6,000 |
| | Social Graph Analysis | High | Low | Low | 20,000 |
| Digital Community Platforms | Review Decision Prediction | Low | High | Low | 12,000 |
| | User Profile Inference | Low | Low | High | 6,000 |

Our contributions are threefold: 1) We identify *deep reasoning, interaction dynamics, and information uncertainty* as three core dimensions for evaluating social reasoning in LLMs, capturing critical challenges found in real-world social cognition. 2) Based on these dimensions, we construct SocialMaze, a benchmark dataset comprising six tasks across three real-world-inspired scenarios (social games, daily life interactions, and digital platforms), covering a wide range of reasoning types and difficulty levels. 3) Our experiments reveal key limitations in LLM social reasoning and point to future directions, showing that targeted fine-tuning can substantially boost performance.

## 2 PROBLEM FORMULATION

We present a graph-based formalization of SocialMaze, where social entities and their evolving interactions are modeled as layered graphs.

**Modeling Social Entities and Interactions as Graph Structures.** We use graph structures to formally represent the participants and interactions within a social scenario. Let $\mathcal{S} = \{s_1, s_2, ..., s_n\}$ be the set of social members involved in a given setting, where each $s_i$ represents a distinct individual, such as a game player, a forum user, or a reviewer in a peer-review process. These social members form the vertex set $\mathcal{V}$ of a graph $G = (\mathcal{V}, \mathcal{E})$, with $\mathcal{V}$ serving as an abstract representation of the social members $\mathcal{S}$. Social interactions between members are represented as edges in the graph. Since interactions are time-dependent, we define a separate edge set $\mathcal{E}_t$ for each interaction round $t$. This leads to a sequence of time-specific graphs $G_t = (\mathcal{V}, \mathcal{E}_t)$, where an edge $(u, v) \in \mathcal{E}_t$ indicates that members $u$ and $v$ interacted during round $t$. The nature of the edges (directed or undirected) reflects the type of interaction. For example, directed edges may represent one-way actions (e.g., sending a message), while undirected edges may represent mutual interactions (e.g., a conversation or vote).

**Temporal Dynamics as Layered Graphs.** Social interactions are inherently dynamic and typically unfold over multiple rounds. To capture this temporal dimension, we represent the entire interaction process as a layered graph $\mathcal{G} = (G_1, G_2, ..., G_T)$, where $T$ denotes the total number of interaction rounds. Each layer $G_t = (\mathcal{V}, \mathcal{E}_t)$ captures the state of social members and their relationships during round $t$. Importantly, all layers share the same vertex set $\mathcal{V}$, reflecting a consistent group of participants throughout the interaction. However, the edge sets $\mathcal{E}_t$ vary across layers to reflect the evolving nature of relationships over time. In SocialMaze, LLMs receive natural language descriptions that encapsulate the information from these layered graphs, rather than raw graph structures. This design choice is intentional, aiming to mimic how humans comprehend social scenarios through language-based narratives.

**Query Categorization.** Based on the layered graph representation, we classify the queries posed within SocialMaze tasks into three distinct types, each targeting a different level of understanding of the graph structure: ***Vertex-centric Query ($\mathcal{Q}_v(v_i)$):*** This type of query probes the model's understanding of individual social members. Given a specific vertex $v_i \in \mathcal{V}$ (representing social member $s_i$), the task is to infer an attribute associated with $v_i$. ***Edge-centric Query ($\mathcal{Q}_e(v_i, v_j)$):*** Edge-centric queries assess the model's comprehension of the relationships between social members. Given two vertices $v_i, v_j \in \mathcal{V}$, the task is to determine the nature of their relationship, as represented by the edges connecting them. ***Graph-level Query ($\mathcal{Q}_G(\mathcal{G})$):*** Graph-level queries require the model to

synthesize information from the entire layered graph $\mathcal{G}$ to derive a holistic understanding of the social scenario. These queries demand a comprehensive assessment of the overall interaction dynamics.

## 3  SOCIALMAZE

As formalized in section 2, and building on the layered social interaction graph framework, we introduce **SocialMaze**, a benchmark designed to operationalize the core challenges of social reasoning—*Dynamic Interaction*, *Information Uncertainty*, and *Deep Reasoning*. The benchmark spans three representative social contexts: Social Reasoning Games, Daily Life Interactions, and Digital Community Platforms. These settings comprise six major task categories, each carefully designed to vary along the key dimensions, enabling systematic evaluation of LLMs under diverse social conditions. Table 1 summarizes the tasks by required reasoning depth, degree of interaction dynamics, and level of information uncertainty. A detailed comparison between SocialMaze and prior benchmarks is provided in Appendix A.

### 3.1  TASK 1: HIDDEN ROLE DEDUCTION

This task simplifies the core mechanics of Blood on the Clocktower (Wikipedia contributors, 2024) into a reasoning-only format. Unlike traditional interaction-based gameplay, all player statements are rule-generated. The model acts as a reasoner, analyzing all available information to logically infer each player's role.

**Task Rules.** The game features four roles: Investigators, Criminal, Rumormongers, and Lunatics. Investigators always tell the truth. The Criminal can choose to lie or tell the truth. Rumormongers believe they are Investigators, but their statements are randomly true or false. Lunatics believe they are the Criminal. The role each player sees may not reflect their true identity—Rumormongers are shown the role of Investigator, and Lunatics are shown the role of Criminal. The game consists of $n$ players, and the model participates by taking the perspective of Player 1 ($s_1$):—meaning it only observes what that player would see and say during the $T$ rounds. In each round, every player selects another player and makes a public statement, such as "Player $v$ says Player $u$ is (not) the criminal." After observing all interactions, the model is tasked with answering two key questions: identifying the true Criminal ($\mathcal{Q}_G$), and inferring its own actual role in the game ($\mathcal{Q}_v(v_i)$). The introduction of Rumormongers and Lunatics significantly increases *information uncertainty* and makes the reasoning process more challenging. The dataset includes four types of tasks: Original task, Rumormonger task, Lunatic task, and Full task. Details are provided in Appendix B.

**Design Rationale.** This task challenges large language models along three critical dimensions of social reasoning. First, the game unfolds over multiple rounds, requiring the model to track the temporal evolution of information, interpret changing relationships among players, and maintain consistent judgments across rounds—posing a challenge of *Dynamic Interaction*. Second, due to the presence of roles such as the Criminal, Rumormonger, and Lunatic—who may lie or provide misleading information—the environment is *highly uncertain*. The model must determine which statements are trustworthy and filter out deceptive cues, thereby grappling with information uncertainty. Most importantly, the model must reason not only about others' roles but also about its own true identity, which may differ from the one initially assigned. Addressing this requires strong *Deep Reasoning* capabilities, including resolving conflicts, managing uncertainty that extends to self-perception, and dynamically updating internal beliefs to approach the ground truth.

**Data Generation and Quality Assurance.** All player statements are automatically generated based on a set of predefined rules. Investigators begin by selecting a target they find suspicious, using a strategy function informed by all interactions up to the current round. They are always truthful in their statements. Rumormongers follow the same target selection logic as Investigators, but the truthfulness of their statements is random, making their input unreliable. In contrast, Criminals and Lunatics adopt a different strategy for choosing targets and deliberately introduce uncertainty by making deceptive statements with a certain probability, aiming to mislead others and conceal their true roles. To ensure each scenario is logically solvable, we design a search algorithm that verifies whether a unique solution exists to identify both the true Criminal and the LLM player's actual role. Additionally, the full reasoning chain leading to the solution is preserved and distilled into clear

natural language, providing high-quality, curated examples of social reasoning that can be leveraged for targeted fine-tuning of language models. See Appendix B for details.

## 3.2 Task 2: Find the Spy

This task adapts the classic word-based social deduction game *Who Is The Spy* (Wei et al., 2025) to evaluate the LLM's ability to identify subtle deviations in communication within a group context characterized by high interaction but relatively low information uncertainty.

**Task Rules.** The game involves $n$ players. Among them, $n-1$ players (Civilians) receive the same secret word, while one player (the Spy) receives a different but related word. Over $T$ rounds, each player provides a description of their word. The LLM is evaluated from the perspective of Player 1 ($s_1$): it knows the word assigned to $s_1$, but does not know whether $s_1$ is a Civilian or the Spy. It does not generate any player descriptions. Instead, after observing all $T$ rounds of player-generated descriptions, the LLM must infer which player received the different word. This constitutes a graph-level query ($\mathcal{Q}_G$). Detailed rules are available in Appendix C.

**Design Rationale.** As shown in Table 1, *Find the Spy* exemplifies High *Dynamic Interaction*. The multi-round format necessitates tracking clues revealed incrementally by all players over time. Conversely, *Information Uncertainty* is designed to be Low. Since players aim to avoid suspicion, they are incentivized to provide truthful descriptions of their assigned word, thereby significantly reducing the element of strategic deception.

**Data Generation and Quality Assurance.** For each game instance, we first set the parameters $n$ (number of players) and $T$ (number of rounds), then randomly selected a related word pair from a curated word bank, followed by random role assignment (one Spy and $n-1$ Civilians). We then used a variety of LLMs to generate player descriptions for each of the $T$ rounds, simulating diverse communication styles. Prompt designs were crafted to encourage varied perspectives and expression strategies across rounds. To ensure quality and solvability, instances underwent human evaluation by 15 computer science graduate students. An instance was considered valid if a majority (>70%) of evaluators could uniquely identify the Spy based on the descriptions. 91% of the evaluated instances met this criterion, verifying their suitability for the benchmark.

## 3.3 Task 3: Rating Estimation from Text

**Task Rules.** This task aims to evaluate the ability of LLMs to predict a product's 1-to-5 star rating based on $n$ textual reviews, which may include genuine positive or negative user comments as well as promotional reviews written by shills. We collect two types of data: reviews generated by LLMs simulating different user types, and real user reviews scraped from platforms such as Amazon, the Google Play Store, and Taobao. The final rating prediction task follows a structure where information flows from multiple user nodes to a central product node, constituting a vertex-centric query focused on the product itself ($\mathcal{Q}_v(\text{product})$). Detailed task rules can be found in Appendix D.

**Design Rationale.** The task deliberately introduces a high level of *information uncertainty*. In the LLM-generated data, this is reflected through the inclusion of simulated "shill" users to mimic deceptive review behavior. In the real-world data, uncertainty arises from the inherent noise, subjectivity, and potential bias present in genuine user reviews. This setting requires the model to evaluate the credibility of information flowing from user nodes (i.e., reviewers) to the product node.

**Data Generation and Quality Assurance.** The LLM-based data generation process begins by sampling product attributes from a manually curated repository consisting of 1,000 attribute terms. A normal distribution of ratings is then constructed based on the true star rating, ensuring that the mean aligns with the reference score. Next, $n$ LLMs are randomly selected from a diverse model pool and probabilistically assigned roles (either normal users or shills) along with distinct personas, which guide the generation of textual reviews consistent with their assigned identities. For real-world data, we directly scrape product attributes and $n$ user reviews from platforms such as Amazon, the Google Play Store, and Taobao. An instance was considered solvable if a majority (>70%) of evaluators could correctly infer the true rating based solely on the textual reviews. Among the LLM-generated samples, 83% satisfied this criterion, confirming their validity for evaluating model reasoning.

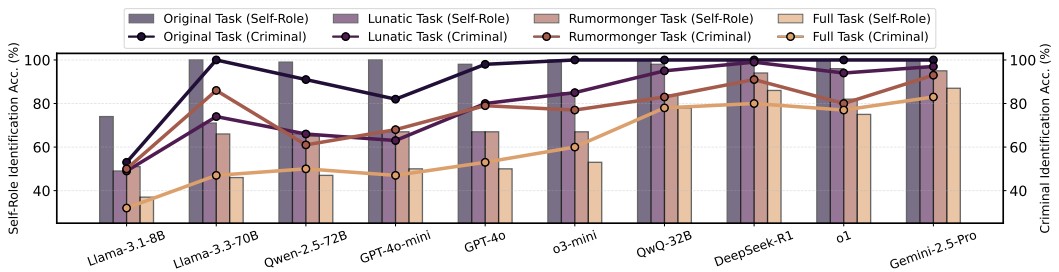

Figure 3: Model performance in *Hidden Role Deduction* across four task variants with increasing information uncertainty. Accuracy is shown after 3 rounds.

## 3.4 PARALLEL TASK SET

**Task4: Social Graph Analysis:** This task aims to evaluate the ability of LLMs to analyze relationships within a social group. Given a description of the social network graph and pairwise relationship labels indicating whether two individuals are friends or have a bad relationship—with friendship being transitive—the model is required to perform reasoning such as: determining whether two individuals are friends, identifying the friend group of a given node, calculating the total number of distinct friend groups, and counting all relationships within the network. Detailed task rules, the algorithmic data generation process that ensures logical consistency and solvability, and quality assurance procedures are provided in Appendix E.

**Task 5: Review Decision Prediction:** This task aims to evaluate the ability of LLMs to predict the final acceptance outcome (Accepted/Rejected) of a research paper as they gradually receive more information throughout the academic review process. The model is required to make a prediction at each of three interaction stages, with the available context incrementally expanding: in the first stage, only the initial paper information is provided; in the second stage, reviewer comments (with numerical scores removed) are added and the model must reason over both the initial content and the reviews; in the third stage, the full author rebuttal is introduced, completing the review context. This task simulates how opinions evolve over time in real academic peer review. Detailed task rules, the data generation process using real-world OpenReview data, and quality assurance procedures are provided in Appendix F.

**Task 6: User Profile Inference:** This task aims to evaluate the ability of LLMs to infer demographic attributes (age group and gender) based on user-generated textual reviews. Specifically, we construct a large number of users with known demographic attributes using LLMs, and generate their reviews for various products they have purchased. The inference tasks are twofold: (1) predicting the dominant user profile associated with the reviews of a specific product, and (2) identifying the profile of an individual user based on their reviews across multiple products. Detailed task rules, the LLM-based data generation method used to embed subtle demographic cues, and human validation results are provided in Appendix G.

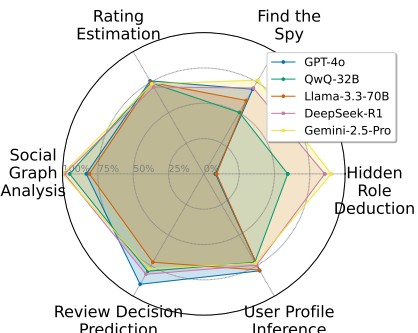

Figure 2: Performance comparison of selected LLMs on SocialMaze tasks, highlighting different model strengths.

## 4 DISCUSSION

We conducted extensive experiments on the SocialMaze benchmark to evaluate the social reasoning capabilities of various LLMs. Specifically, we tested 5 leading proprietary LLMs and 6 open-weight LLMs across our tasks, covering diverse aspects of social reasoning. In addition, we evaluated 6 different workflow strategies to assess their impact on model performance. A subset of the results is visualized in Figure 2. We observe that different social reasoning tasks impose distinct demands on the models. For example, tasks like *Hidden Role Deduction*, which require *Deep Reasoning*, are best tackled by models such as DeepSeek-R1 and Gemini-2.5-Pro. In contrast, GPT-4o excels in tasks like *Review Decision Prediction*, where nuanced understanding of reviewer attitudes is critical. We

conducted extensive case studies and report full results and settings in Appendix K, Appendix H, and Appendix I.

## 4.1 THE IMPACT OF DEEP REASONING REQUIREMENTS

In certain complex scenarios, effective social reasoning often requires going beyond surface-level cues—a process we refer to as *Deep Reasoning*. Our benchmark explicitly differentiates tasks along this dimension (Table 1), categorizing them into those that demand *Deep Reasoning* (*Hidden Role Deduction* and *Social Graph Analysis*) and those that are primarily solvable through more superficial. To assess the impact of reasoning depth, we compare two model categories: Long CoT models (e.g., o1, DeepSeek-R1), which generate detailed, step-by-step reasoning chains, and the remaining Short CoT models, which follow shorter reasoning paths—both using identical prompts.

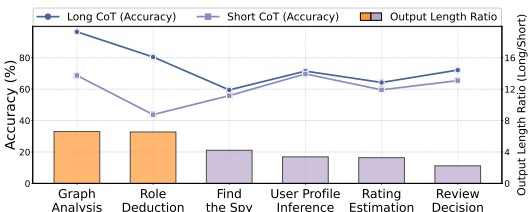

**Long CoT models achieve substantially higher accuracy on tasks requiring *Deep Reasoning*.** As shown in the line plot in Figure 4, the performance advantage of Long CoT models is particularly pronounced for *Deep Reasoning* tasks, i.e., Graph analysis and role deduction. While employing longer reasoning chains also yields modest improvements on shallow reasoning tasks, the accuracy gap between the two model types is significantly narrower. This indicates that the explicit, step-by-step reasoning characteristic of Long CoT models is especially beneficial for handling *Deep Reasoning* intricacies (e.g., inferring latent beliefs, analyzing relations); manual inspection confirmed this rea-

Figure 4: Performance comparison of Long CoT and Short CoT models. The line plot shows average accuracy; the bar plot shows the output length ratio (Long CoT / Short CoT). Orange bars indicate tasks with high deep reasoning demand, purple bars indicate low deep reasoning demand.

soning is sound and coherent in correct predictions, suggesting genuine inference capabilities.

**The improved performance of Long CoT models comes with a substantial computational cost.** The bar plot in Figure 4 illustrates this cost by presenting the ratio of average number of output tokens between Long CoT and Short CoT models. For *Deep Reasoning* tasks, outputs from Long CoT models contain nearly eight times more tokens on average, reflecting a much more extensive externalization of intermediate reasoning steps, hypothesis testing, and evidence evaluation. In contrast, for shallow reasoning tasks, the difference in output token count is less pronounced, mirroring smaller accuracy gains and suggesting that these tasks can often be solved without lengthy, explicit reasoning chains.

## 4.2 THE IMPACT OF DYNAMIC INTERACTION

In certain scenarios, social interactions unfold sequentially, requiring models to integrate and reason over information accumulated across multiple turns or stages. We analyze how model performance evolves in tasks characterized by high *Dynamic Interaction*, revealing distinct patterns depending on the nature and structure of the accumulating information. Overall, **model accuracy generally improves with quantitatively increasing interaction, but the trajectory of performance evolution and sensitivity to dynamic information vary significantly across different tasks and models.**

This contrast is particularly evident in two tasks: In *Hidden Role Deduction*, we track model accuracy in identifying the Criminal within the Full Task setting involving six players (Table 2). **Accuracy tends to increase as more**

Table 2: Criminal identification accuracy across rounds in the 6-player *Hidden Role Deduction* task. Models vary in leveraging *Dynamic Interaction*.

| Model | Round 1 | Round 2 | Round 3 |
|---|---|---|---|
| Llama-3.3-70B | 37.6% | 46.7% | 46.5% |
| Qwen-2.5-72B | 31.3% | 42.6% | 50.3% |
| GPT-4o-mini | 33.5% | 38.4% | 46.5% |
| GPT-4o | 39.5% | 53.3% | 53.5% |
| o3-mini | **45.8%** | 51.2% | 59.6% |
| QwQ-32B | 41.4% | 63.5% | 78.4% |
| DeepSeek-R1 | 44.3% | 72.3% | 80.4% |
| o1 | 42.5% | 67.5% | 76.6% |
| Gemini-2.5-Pro | 43.3% | **74.3%** | **87.6%** |

**rounds of interaction are observed**, reflecting the expected benefit of accumulating contextual evidence over time. However, the slope of improvement varies substantially across models, suggesting differing abilities to process and integrate newly revealed statements within the game's evolving context. Some models are more effective than others in leveraging additional rounds to refine their

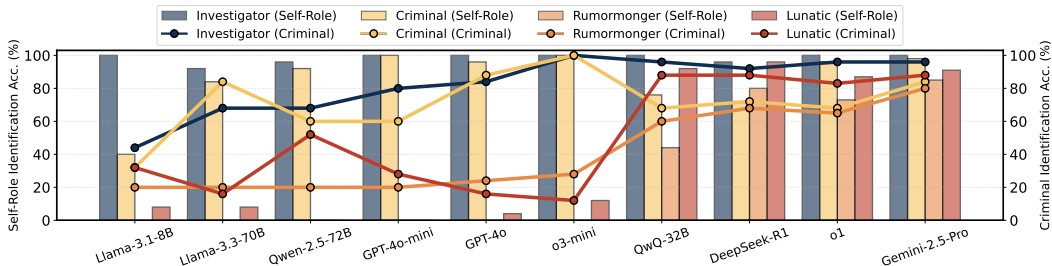

Figure 5: Performance in the Full task of *Hidden Role Deduction*, by model-assigned role. Models show reduced accuracy when assigned roles involving distorted self-perception (Rumormonger, Lunatic).

hypotheses. One notable insight is the large performance gap between Rumormonger and Lunatic. This is because once an Investigator correctly checks the Lunatic as "not the Criminal," it provides a strong signal that helps the Lunatic realize their true role—making awakening easier than for the Rumormonger.

By contrast, *Review Decision Prediction* presents a more complex and non-linear performance trajectory across the stages of the peer review process, as shown in Table 3. **Initial paper information yields low accuracy, reviews trigger a major performance boost, but the final rebuttal stage often causes a drop in accuracy.** We observe that this counterintuitive decline is frequently driven by the model being swayed by the author's sincere and well-articulated defense, which may not align with the actual judgment rendered by human reviewers or area chairs. In other words, the model is "convinced" by the rebuttal, even when it fails to change the ultimate decision. A more detailed analysis of this phenomenon is provided in Appendix K.

## 4.3 THE IMPACT OF INFORMATION UNCERTAINTY

A defining characteristic of complex social environments is the prevalence of unreliable information. We evaluate the impact of Information Uncertainty using the *Hidden Role Deduction* task by systematically introducing actors who generate distinct forms of unreliable information: intentional deception (Criminal) and noise stemming from flawed self-perception (Rumormonger and Lunatic).

**Increased information uncertainty significantly elevates the difficulty of social reasoning for LLMs.** Figure 3 illustrates this across four task configurations where uncertainty levels are quantitatively controlled by varying the number of unreliable actors (Rumormongers, Lunatics). Progressing from the baseline Origi-

Table 3: *Review Decision Prediction* accuracy across sequential stages. Accuracy improves with reviewer comments but often drops after incorporating the rebuttal.

| Model | Info. | Reviews | Rebuttal |
|---|---|---|---|
| Llama-3.1-8B | 37.0% | 79.6% | 62.0% |
| Llama-3.3-70B | 26.2% | 87.4% | 72.2% |
| Qwen-2.5-72B | 23.6% | 82.2% | 65.8% |
| Phi-4 | 23.5% | 77.6% | 61.4% |
| GPT-4o-mini | 38.8% | 85.2% | 85.0% |
| GPT-4o | **55.3%** | 86.2% | **90.2%** |
| o3-mini | 40.2% | 86.4% | 78.6% |
| QWQ-32B | 49.8% | 83.8% | 79.6% |
| DeepSeek-R1 | 50.2% | 88.0% | 82.0% |
| o1 | 52.2% | **88.6%** | 78.2% |
| Gemini-2.5-Pro | 47.4% | 87.6% | 77.6% |

nal setting through scenarios introducing these noise sources to the complex Full setting, both the accuracy in identifying the Criminal (line plot) and the model's own role (bar plot) demonstrate a marked decline. This degradation underscores the substantial challenge posed by noise and deception.

**Reasoning becomes particularly challenging when the model's own perceived role or information source is compromised.** Figure 5 delves into the Full task configuration, analyzing final accuracy based on the specific role assigned to the LLM. Models exhibit considerably lower accuracy, especially in identifying their own role (bars), when assigned as a Rumormonger or Lunatic compared to being an Investigator or Criminal. This suggests a significant difficulty in reconciling internal beliefs (e.g., "I think I am an Investigator/Criminal") with conflicting external evidence or the unreliable nature of one's own information.

These experiments also highlight critical differences between model capabilities. **Existing Short CoT models demonstrate severe limitations in handling complex scenarios with high infor-**

Table 4: Performance on *Hidden Role Deduction* before and after fine-tuning. **Crim.**: Criminal prediction accuracy; **Self**: self-role prediction accuracy; **Both**: both predictions correct. Fine-tuning with SFT and DPO significantly improves performance on both models. Percent improvements over the Base model for **Both** are shown in parentheses.

| Model | Base | | | SFT | | | DPO | | |
|---|---|---|---|---|---|---|---|---|---|
| | **Crim.** | **Self** | **Both** | **Crim.** | **Self** | **Both** | **Crim.** | **Self** | **Both** |
| LLaMA-3.1-8B | 33.0% | 8.4% | 2.0% | 37.0% | 15.2% | 13.4% (+11.4%) | 35.4% | 11.0% | 9.8% (+7.8%) |
| Phi-4 | 31.2% | 13.4% | 8.2% | 38.2% | 22.6% | 19.8% (+11.6%) | 37.8% | 17.4% | 15.2% (+7.0%) |

**mation uncertainty**, often failing to perform reliably in the Rumormonger and Full task settings (Figure 3). In contrast, Long CoT models, while still impacted, exhibit significantly better resilience to uncertainty. Furthermore, the challenge of self-assessment under uncertainty exposes a stark gap: **Short CoT models are almost entirely unable to deduce their true identity when assigned as a Rumormonger or Lunatic** (Figure 5), suggesting a profound lack of capacity for self-doubt and meta-reasoning necessary to overcome compromised initial information.

### 4.4 ENHANCING SOCIAL REASONING CAPABILITIES

To explore strategies for improving social reasoning in LLMs, we conduct focused experiments on the *Hidden Role Deduction* task. This task is not only uniquely representative—combining Deep Reasoning, Dynamic Interaction, and Information Uncertainty—but also well-suited for generating diverse reasoning examples at scale, providing valuable supervision for model learning.

**Reasoning agents and workflows offer limited gains for social reasoning.** We first assess whether reasoning agents and workflows effective in task decomposition and planning can improve performance on *Hidden Role Deduction*. As shown in Table 7, various agentic implementations yield only marginal improvements over base models. This indicates that current workflow strategies are not enough to handle the complexity and uncertainty involved in social reasoning tasks.

**Fine-tuning on curated reasoning traces substantially improves performance.** Recognizing the need for models to internalize complex reasoning strategies, we further explore instruction-based fine-tuning using high-quality examples from our dataset. Table 4 summarizes results from applying Supervised Fine-Tuning (SFT) (Ouyang et al., 2022) and Direct Preference Optimization (DPO) (Rafailov et al., 2023) to Llama-3.1-8B and Phi-4. Both approaches substantially improve accuracy on the *Hidden Role Deduction* task, and also yield slight but consistent gains on other benchmark tasks, thereby demonstrating the effectiveness and generalizability of targeted fine-tuning on curated reasoning examples.

These findings highlight a fundamental limitation of existing LLM agents in socially complex reasoning tasks. In contrast, our targeted fine-tuning approach yields substantial improvements, particularly because it leverages high-quality reasoning traces specifically crafted for social contexts. This suggests that equipping models with domain-relevant reasoning strategies through fine-tuning may be a more fruitful path toward enhancing their capabilities in this domain.

## 5 CONCLUSION

We introduced SocialMaze, a benchmark designed to rigorously evaluate the social reasoning capabilities of LLMs by capturing the challenges of deep reasoning, dynamic interaction, and information uncertainty. Experiments across six diverse tasks reveal notable weaknesses in current models, particularly in handling evolving contexts and reasoning under uncertainty. Targeted fine-tuning on curated reasoning examples significantly improves performance, highlighting the value of domain-specific adaptation. As future work, we plan to further expand SocialMaze by collecting more real-world data and aim to enrich the benchmark with new social scenarios and task types. We believe our work will provide valuable resources and insights to the research community, helping to advance the development of LLMs with stronger social reasoning capabilities.

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

# A RELATED WORKS

## A.1 STATIC SOCIAL REASONING BENCHMARKS

Early evaluations of social reasoning in language models largely focus on static, single-turn tasks where models infer plausible answers based on brief, pre-written scenarios. These benchmarks primarily test shallow forms of commonsense and moral reasoning without requiring interaction, adaptation, or uncertainty handling. A classical example is **SocialIQA** Sap et al. (2019b), which assesses social commonsense by posing questions about motivations and reactions in everyday situations. Similarly, **ATOMIC** Sap et al. (2019a) provides a structured knowledge graph of causal social events, while **CommonGen** Lin et al. (2020) tests the ability to generate plausible sentences from object co-occurrence tuples in socially relevant scenes.

Benchmarks like **Social Chemistry** Forbes et al. (2020) and **ETHICS** Hendrycks et al. (2021) target moral norms, asking models to judge the appropriateness of actions, whereas **GLU-COSE** Mostafazadeh et al. (2020) emphasizes implicit causal reasoning in narratives. Emotion understanding is covered by **GoEmotions** Demszky et al. (2020), while **PIQA** Bisk et al. (2020), **HellaSwag** Zellers et al. (2019), and **CREAK** Onoe et al. (2021) assess commonsense or contextually grounded inference.

Recent work has shifted toward evaluating *Theory of Mind* (ToM)—the capacity to reason about others' beliefs and intentions. Initial tests, such as those in Kosinski (2023), used classic false-belief stories to probe emergent ToM in LLMs, though follow-up studies revealed artifacts and prompt sensitivity Shapira et al. (2023). More recent evaluations such as **FANToM** Kim et al. (2023) and **ToMValley** Xiao et al. provide more rigorous multi-turn belief-tracking scenarios. These benchmarks demonstrate that, while LLMs may succeed on simple static ToM questions, they struggle with deeper or dynamic belief modeling, particularly under information asymmetry or belief shifts over time.

Overall, while these static benchmarks have advanced our understanding of LLMs' social reasoning capabilities, they often lack three essential ingredients for real-world social cognition: dynamic interaction, complex reasoning depth, and reasoning under uncertainty. Table 5 summarizes several representative benchmarks discussed above, highlighting their focus, format, and limitations. These gaps motivate the design of SocialMaze, which aims to capture the challenges of socially grounded reasoning in more realistic and interactive settings.

Table 5: Representative Benchmarks for Social Reasoning in LLMs

| Benchmark | Focus | Format | Key Findings |
| --- | --- | --- | --- |
| **SocialIQA** Sap et al. (2019b) | Commonsense QA | Static narrative + MCQ | GPT-3+/GPT-4 near human; tests scripted social norms. |
| **ToM Classic** Kosinski (2023) | False belief | Brief stories + Q&A | Early ToM claims challenged; shortcut artifacts noted. |
| **Clever Hans** Shapira et al. (2023) | ToM artifacts | Controlled stimuli | Performance drops without spurious cues; lacks robustness. |
| **FANToM** Kim et al. (2023) | Interactive ToM | Multi-turn dialogue | GPT-4 and others fail under asymmetric info tracking. |
| **ToMValley** Xiao et al. | Dynamic ToM | Scenario chains + Q&A | LLMs 11% below humans; weak on mental state updates. |

## A.2 DYNAMIC SOCIAL REASONING AND INTERACTION BENCHMARKS

Another class of benchmarks focuses on dynamic settings where social reasoning occurs within interactive contexts. These tasks require models to engage with evolving information, track perspectives, and reason about hidden roles, intentions, or potential deception. DebateBench Tiwari et al. (2025) evaluates models' reasoning across long-form argumentative dialogues. Studies of peer review processes using OpenReview data Tran et al. (2020); Szumega et al. (2023) examine decision-making through multi-turn, text-based interaction without relying solely on reviewer scores.

Strategic games such as Diplomacy Bakhtin et al. (2022) and Poker Brown & Sandholm (2019), as well as social deduction games like Avalon and Werewolf Light et al. (2023); Wei et al. (2025); Bailis et al. (2024); Xu et al. (2023), provide natural settings for reasoning under incomplete information and complex social dynamics. Multi-agent simulations such as Generative Agents Park et al. (2023) explore emergent social behavior, while robustness evaluations like TextAttack Morris et al. (2020), TRUST-LLM Sun et al. (2024), and Ribeiro et al. (2020) investigate model behavior under uncertainty or adversarial conditions.

While these benchmarks incorporate certain forms of interaction and complexity, most remain oriented toward task completion or strategic decision-making. They often lack systematic modeling of social context factors such as role motivations or deception. SocialMaze, by contrast, integrates interaction and uncertainty within a structured framework, explicitly targeting complex social reasoning.

## B  HIDDEN ROLE DEDUCTION

This appendix provides the details for the Hidden Role Deduction task described in subsection 3.2.

**1. Task Setup**

The game involves $n$ players, denoted $s_1, ..., s_n$. The LLM observes the game unfolding from the perspective of Player 1 ($s_1$), receiving all interaction data and the initial role assignment given to $s_1$. **Crucially, the LLM does not actively participate in the game simulation (i.e., does not choose targets or make statements). Its task is solely post-hoc inference**: analyzing the complete interaction log ($G_1, ..., G_T$) to deduce the answers to the specified queries. At the start, each player $s_i$ is randomly assigned a secret role. The LLM is informed of the role initially assigned to $s_1$; however, this assigned role may differ from $s_1$'s actual role, especially if $s_1$ is a Rumormonger or Lunatic. The specific composition of roles depends on the Task Variant being used.

**2. Roles and Behaviors**

Simulated player behavior is guided by their assigned role and the game history $G_t$ up to round $t$. The roles are defined as follows: The **Investigator (I)** aims to identify the Criminal. Based on the game history $G_t$, an Investigator uses an algorithmically defined function $F_I(G_t)$ to select a target player $u$. They then make a **truthful** statement reflecting their deduction about $u$'s status (Criminal or not Criminal). The function $F_I(G_t)$ heuristically assesses suspicion, selecting players with higher probability if they have made contradictory statements or have been accused by others, while lowering the probability for players cleared by multiple potentially reliable sources. The **Criminal (C)** seeks to avoid identification and mislead others. They employ a strategic function $F_C(G_t)$ to select a target player $u$ and also determine a probability $p_t = P(\text{state } u \text{ is Criminal}|G_t, \text{role=C})$. The Criminal then states "$u$ is Criminal" with probability $p_t$ and "$u$ is not Criminal" with probability $1 - p_t$. The function $F_C$ implements deceptive tactics, prioritizing targeting players who have accused the Criminal and diverting suspicion onto others. The **Rumormonger (R)**, although believing they are an Investigator trying to identify the Criminal, unintentionally provides unreliable information, effectively injecting noise. They are *told* they are an Investigator and use the Investigator logic function $F_I(G_t)$ to select a target player $u$. However, regardless of any internal assessment derived from $F_I(G_t)$ or the actual ground truth, the truthfulness of their final statement ("$u$ is Criminal" or "$u$ is not Criminal") is entirely random, possessing a 50% probability of aligning with the ground truth and a 50% probability of contradicting it. Lastly, the **Lunatic (L)** believes they are the Criminal and aims to avoid identification based on this false premise, while their actual nature is not Criminal. They are *told* they are the Criminal and mimic the Criminal's behavior by employing the *same* strategic function $F_C(G_t)$ used by the actual Criminal. Although their actions follow deceptive patterns, truthful statements made by Investigators about the Lunatic will correctly identify them as 'not the Criminal'.

**3. Interaction Rounds**

The game simulation proceeds for a fixed $T$ rounds. In each round $t$ (from 1 to $T$), every player $s_v$ selects another player $P_u$ and makes a public statement of the form: "Player $v$ says Player $u$ is the criminal" or "Player $v$ says Player $u$ is not the criminal." All statements made in round $t$ are revealed simultaneously to all players, and thus become available to the observing LLM, before round

$t + 1$ begins. Consequently, the complete history $G_T = (\text{statements}_{round1}, ..., \text{statements}_{roundT})$ is available for the LLM's final analysis.

## 4. Parameter Settings

The composition of roles is varied to create tasks of differing complexity, ensuring there is always exactly one Criminal. The variants include: the **Original Task** (1 Criminal, $n - 1$ Investigators); the **Rumormonger Task** (1 Criminal, $x \geq 1$ Rumormongers, $n - 1 - x$ Investigators); the **Lunatic Task** (1 Criminal, $y \geq 1$ Lunatics, $n - 1 - y$ Investigators); and the **Full Task** (1 Criminal, $x \geq 0$ Rumormongers, $y \geq 0$ Lunatics, $n - 1 - x - y$ Investigators, where $x + y \geq 1$). **Note on Experimental Configuration:** For the experiments presented in this paper, the game parameters were fixed at $n = 6$ players. When Rumormongers and Lunatics were included (specifically in the Full Task variant experiments), their counts were set to $x = 1$ and $y = 1$, respectively, alongside 1 Criminal and $n - 1 - x - y = 3$ Investigators. The accompanying open-sourced dataset includes generated instances for both $n = 6$ and $n = 10$. Furthermore, the open-sourced data generation code is flexible, allowing users to configure $n, m, x$, and $y$ to create custom game scenarios.

## 5. QUALITY CONTROL

The algorithmic generation includes verification via heuristic search, ensuring a unique, logically derivable solution exists for both queries from $P_1$'s perspective using only the interactions and rules. The core verification logic, which checks all valid hypotheses, is outlined in algorithm 1.

Furthermore, to validate the models' reasoning quality, we manually inspected 100 correct responses each for several key models (DeepSeek-R1, QwQ-32B, Gemini-2.5-Pro, o1, and o3-mini). This analysis confirmed that over 90% of these successful predictions were underpinned by reasoning processes assessed as both rigorous and logically sound.

---

**Algorithm 1:** Solvability Verification from $P_1$'s Perspective

---

**Input** : Interaction Log $S$, Player Set $\mathcal{P} = \{P_1, \ldots, P_n\}$, Role Set $\mathcal{R} = \{I, C, R, L\}$, Investigator Count $N_I$

**Output** : Unique Solution $(C^*, R_1^*) \in \mathcal{P} \times \mathcal{R}$, or $\emptyset$ if not unique/no solution

1   $\mathcal{S}_{valid} \leftarrow \emptyset$;
   ▷ Set to store valid (Criminal, P1 Role) solution pairs

2 **foreach** *hypothesized role* $R_1^{hyp} \in \mathcal{R}$ *for* $P_1$ **do**

3    $\mathcal{P}_{cand} \leftarrow \mathcal{P} \setminus \{P_1\}$;
     ▷ Initial candidates are all others

4    **foreach** *subset* $I_{hyp} \subseteq \mathcal{P}_{cand}$ *such that* $|I_{hyp}| = N_I$ **do**

5      Let current hypothesis $H = (R_1^{hyp}, I_{hyp})$;

6      **if** *IsConsistent(S, H)* **then**
       ▷ Check if hypothesis contradicts any statement

7        $C_{implied} \leftarrow \text{DeduceCriminal}(S, H)$;

8        $R_{1,implied} \leftarrow \text{DeduceP1Role}(S, H)$;

9        **if** $C_{implied} \neq NULL$ **and** $R_{1,implied} \neq NULL$ **then**

10          $\mathcal{S}_{valid} \leftarrow \mathcal{S}_{valid} \cup \{(C_{implied}, R_{1,implied})\}$;
          ▷ Add deduced solution

11 **if** $|\mathcal{S}_{valid}| = 1$ **then**

12    **return** the single element in $\mathcal{S}_{valid}$;
     ▷ Unique solution found

13 **else**

14    **return** $\emptyset$;
     ▷ Not unique or no consistent solution

---

## C  FIND THE SPY

This appendix provides the details for the *Find the Spy* task described in subsection 3.2.

### 1. Task Setup

The game involves $n$ players, denoted $s_1, \ldots, s_n$. The evaluated LLM adopts the persona of Player 1 ($s_1$) but acts as a passive observer. It receives all game information, including its own assigned word and all player descriptions, from $s_1$'s perspective but does not generate descriptions itself during the evaluation process. The LLM's sole task is to identify the Spy based on the observed interactions. For word assignment, a pair of semantically related but distinct words (Word A, Word B, e.g., "Milk" and "Soy Milk") is selected from a predefined bank. One player is randomly designated as the Spy, while the remaining $n - 1$ players are Civilians. All Civilians receive Word A, and the Spy receives Word B. Player 1 (the LLM's persona) is informed of the word it received but is not explicitly told whether it is Word A or Word B, nor is it told its role (Civilian or Spy).

### 2. Interaction Rounds

The game proceeds for $T$ rounds. In each round $t$ (from 1 to $T$), every player $s_i$ provides a textual description of the word they possess. These descriptions are generated by LLMs, with each player $s_i$ being assigned a specific LLM generator (selected randomly and uniformly from a predefined pool: GPT-4o-mini, GPT-4o, Llama-3.3-70B, Qwen-2.5-72B) for the entire game instance. The generation prompts encourage the LLM simulating player $s_i$ to describe their word from different angles or aspects in each round, avoiding simple repetition. Players are assumed to provide descriptions consistent with the word they hold: Civilians describe Word A, and the Spy describes Word B. All descriptions generated in round $t$ are made available to all players, including the LLM as $s_1$, before round $t + 1$ begins. Thus, the complete history of descriptions from rounds 1 to $T$ is available at the end of the game for the LLM's analysis.

### 3. Parameter Settings

For all experimental evaluations using this task setup, the number of players $n$ was fixed at 4, and the number of interaction rounds $T$ was fixed at 3.

### 4. Quality Control

To ensure generated instances are solvable yet challenging, they undergo a rigorous human validation process. Each potential game instance is reviewed by 15 evaluators, all holding at least an undergraduate degree in Computer Science. An instance is deemed valid and solvable only if a clear majority (more than 5 out of the 10 evaluators) agree that the collective descriptions provided by the simulated players contain sufficient evidence to uniquely identify the Spy. This validation confirms that the task's difficulty stems from semantic subtlety and variations in description style, rather than from a fundamental lack of necessary information. 91% of the instances subjected to this evaluation met the validation threshold, affirming their suitability for inclusion in the benchmark.

## D  RATING ESTIMATION FROM TEXT

This appendix provides the details for the Rating Estimation from Text task described in subsection 3.3.

### 1. Task Setup

The primary objective for the LLM in this task is to estimate the most likely overall "true" star rating, represented as an integer from 1 to 5, for a given product. This estimation must be based solely on the textual content derived from multiple user reviews. For each task instance, the LLM receives specific input: a set of attributes describing the product (such as type, category, key features, price range) and a list containing $n$ individual textual user reviews for that product. Crucially, the original star ratings (1-5 stars) that reviewers might have provided are explicitly omitted from the input. After processing the product information and the $n$ textual reviews, the LLM must answer a vertex-centric query ($\mathcal{Q}_v(\text{product})$) phrased as: "Based on the provided reviews, what is the most likely overall star rating for this product? Choose one: 1, 2, 3, 4, or 5." The expected output is a single integer, necessitating the synthesis of information from multiple user interactions (reviews) all directed towards the product entity.

## 2. Data Generation

The task utilizes review data generated through two distinct methods. The first method involves **LLM-Generated Reviews**. Here, a product profile is selected, including its attributes and a designated ground truth overall star rating. A number $n$ of simulated reviewers is determined, and each reviewer $i$ is assigned a role probabilistically, ensuring the majority are *Normal Users* (providing honest feedback) while a small fraction are designated as *Positive Shills* or *Negative Shills*. Each reviewer also receives a simple persona for stylistic variation and is assigned a randomly selected LLM from a predefined pool (GPT-4o-mini, GPT-4o, Llama-3.3-70B, Qwen-2.5-72B) to simulate their response. The assigned LLM then generates the review text for reviewer $i$, guided by the product attributes, the ground truth rating, the reviewer's assigned role (Normal/Shill), and their persona; Shills are prompted to generate biased text accordingly. Finally, only the generated textual reviews are collected and prepared as input for the evaluated LLM. The second method uses **Real-World Reviews**. In this scenario, product attributes along with user reviews (both text and original ratings) are scraped from public e-commerce and app platforms. For a selected product, $n$ reviews are sampled from the scraped collection. The textual content of these $n$ sampled reviews is extracted, while the original star ratings are discarded. Only the product attributes and the review texts are provided as input to the LLM. The ground truth for these instances is typically derived from the average rating found on the source platform, which the LLM is tasked to estimate.

## 3. Parameter Settings

Across all experiments presented for this task, the number of reviews ($n$) provided per product instance is consistently set to 8. The required output is always a single integer rating on the 1-to-5 star scale. The real-world task instances exclusively utilize product attributes and textual reviews sourced from **Amazon**, the **Google Play Store**, and **Taobao**. For the LLM-generated instances, the underlying product attributes and the initial ground truth star ratings are also sampled from this same pool of real-world data derived from these platforms, providing a basis grounded in realistic product scenarios.

## 4. Quality Control

As detailed in subsection 3.3, human evaluations were integral to ensuring data quality. These were conducted by 15 graduate students. For the LLM-generated data (Scenario A), this evaluation determined that 83% of the assessed instances were solvable, meaning the true rating could be reasonably inferred from the text alone by more than half of the human evaluators. For the instances derived from real-world data (Scenario B), solvability is inherently tied to the complexity and nature of authentic customer feedback as it appears on platforms like Amazon, Google Play, and Taobao, reflecting genuine information landscapes.

# E   SOCIAL GRAPH ANALYSIS

This appendix provides the details for the Social Graph Analysis task described in Section 3.4.

## 1. Setup

This task presents a stylized social network scenario involving a set of $n$ individuals. The core challenge lies in understanding the structure of this network, where relationships between any two individuals are strictly defined as either 'good' or 'bad'. The LLM is provided with a complete description of all pairwise relationships and must then analyze this information to answer queries about specific relationships, individual connections, and the overall emergent group structure of the network, guided by a set of simple logical axioms.

## 2. Relationship Axioms

Relationships between any two distinct individuals, say Person A and Person B, are binary ('good' or 'bad') and symmetric. These relationships are governed by specific logical rules: **Axiom 1 dictates the transitivity of good relationships**, meaning if A and B have a 'good' relationship, and B and C also have a 'good' relationship, then A and C must necessarily have a 'good' relationship. **Axiom 2 describes the implication of bad relationships**, stating that if A and B have a 'bad' relationship, and A and C have a 'good' relationship, then B and C are forced to have a 'bad' relationship. It's important to note that from these axioms, if A and B share a 'bad' relationship, and B and C also share a 'bad'

relationship, the nature of the relationship between A and C is not determined solely by these two facts; it could be either good or bad depending on other connections within the network. However, the algorithmic generation process always ensures a globally consistent and valid relationship structure.

**3. Group Definition**

Within this social structure, a 'group' is formally defined as a maximal set of individuals where every person within that set has a 'good' relationship with every other person also belonging to that same set. A key property of this structure is that every individual belongs to exactly one such group. Consequently, based on the governing axioms and the generation method, the relationship between any two individuals can be directly inferred from their group membership: if Person A and Person B are members of the same group, they inherently have a 'good' relationship; conversely, if they belong to different groups, they must have a 'bad' relationship.

**4. Input Format**

The LLM receives as input a comprehensive list composed of natural language statements that explicitly specify **the complete set of pairwise relationships** as determined by the algorithmic generation process. These statements clearly define the relationship status between every possible pair of individuals within the scenario. Examples of such input statements include "Person N and Person G have a good relationship" and "Person K and Person P have a bad relationship". This list provides a full and unambiguous description of the entire social graph structure.

**5. LLM Queries**

After processing the complete list of relationship statements provided as input, the LLM is required to answer various types of queries designed to test its understanding of the network structure. These queries include, for instance, **Pairwise Relationship Queries** ($\mathcal{Q}_e(v_i, v_j)$), such as "Do Person N and Person L have a good relationship?", which typically requires checking the provided input directly and responding with Yes/No. Other queries are **Good Relationship Neighbor Queries (Vertex-centric)**, like "Who has a good relationship with Person H?", demanding the LLM to filter the input and list the relevant names. Furthermore, **Graph-level Queries** ($\mathcal{Q}_G$) probe the overall structure, asking questions like "How many groups of people are there?" or "How many pairs of people have good relationships?" or "How many pairs of people have bad relationships?", all of which require synthesizing the pairwise information to derive a global property and respond with an integer count.

**6. Data Generation and Quality Assurance**

Instances for this task are generated entirely algorithmically, without reliance on LLM generation, ensuring consistency and verifiable ground truth. The process begins by setting the number of individuals $n$, sampled from [8, 10] for 'easy' instances and [14, 16] for 'hard' instances. A complete graph structure respecting the relationship axioms is then algorithmically constructed. First, a spanning forest is created using only 'good' relationship edges, thereby defining the distinct social groups (each tree representing a group). Second, 'bad' relationship edges are strategically added to connect every pair of distinct groups (trees), ensuring all inter-group relations are 'bad' and all intra-group relations are 'good'. The complete set of generated relationship edges ('good' edges defining the groups and 'bad' edges connecting them) is then converted into natural language statements and presented to the LLM as input. This generation methodology mathematically guarantees that for each instance, a unique solution exists for all four query types and is logically derivable solely from the provided statements and rules. The core generation logic is outlined in algorithm 2.

# F   REVIEW DECISION PREDICTION

This appendix provides the details for the Review Decision Prediction task described in subsection 3.4.

**1. Objective**

The LLM's goal in this task is to predict the final acceptance status (Accepted or Rejected) of a research manuscript submitted to a conference, based solely on the sequence of provided peer review communications.

**2. Data Source and Scope**

---

**Algorithm 2:** Social Graph Generation

---

**Input** : Difficulty level $d \in \{\text{easy}, \text{hard}\}$
**Output** : Natural language instance $I$

1 Set number of individuals $n \sim \begin{cases} [8, 10], & \text{if } d = \text{easy} \\ [14, 16], & \text{if } d = \text{hard} \end{cases}$;

2 Initialize graph $G = (V, E)$, where $|V| = n$;

3 **Step 1: Generate spanning forest to define social groups**;

4 Randomly partition $V$ into $m \geq 2$ disjoint non-empty subsets $\{V_1, V_2, \ldots, V_m\}$;

5 **foreach** *group $V_i$* **do**

6 $\quad$ Generate a spanning tree $T_i = (V_i, E_i^{\text{good}})$;

7 $\quad$ Add edges $E_i^{\text{good}}$ to $E$ // Intra-group "good" relationships

8 **Step 2: Add "bad" edges between groups**;

9 **foreach** *pair of groups $(V_i, V_j)$, $i \neq j$* **do**

10 $\quad$ Select a node pair $(u, v) \in V_i \times V_j$;

11 $\quad$ Add edge $(u, v)$ to $E^{\text{bad}} \subset E$ // Inter-group "bad" relationship

12 **Step 3: Convert graph structure to natural language**;

13 **foreach** *edge $(u, v) \in E$* **do**

14 $\quad$ **if** $(u, v) \in E^{good}$ **then**

15 $\quad\quad$ Generate statement: "$u$ and $v$ are good friends.";

16 $\quad$ **else if** $(u, v) \in E^{bad}$ **then**

17 $\quad\quad$ Generate statement: "$u$ and $v$ do not get along.";

18 Aggregate all generated statements into input instance $I$;

19 **return** $I$;

---

Data for this task is exclusively sourced from the official OpenReview API, encompassing submissions to specific high-profile Artificial Intelligence and Machine Learning conferences, namely NeurIPS (covering the 2023 and 2024 cycles) and ICLR (covering the 2020, 2021, 2022, 2023, and 2024 cycles). **The rationale for selecting these particular venues** stems primarily from their policy of making the entire peer review process public. This transparency, which *crucially includes making detailed reviews and discussions for rejected manuscripts publicly available*, is a practice not commonly found in many other academic fields. It provides the essential data needed to construct a balanced and realistic task dataset that accurately reflects both acceptance and rejection scenarios encountered in academic publishing.

### 3. Input Structure

The LLM receives information pertaining to a single manuscript, presented in a structured sequence that mirrors the typical progression of the peer review timeline. Initially, in **Round 1**, the LLM is given the initial submission details: the manuscript's original **Title**, its **Abstract**, and the author-provided **Keywords**. Subsequently, in **Round 2**, the LLM receives the reviewer feedback, which consists of the complete **textual content** of each review submitted by the assigned reviewers. It is **crucial to note the exclusion** of all quantitative aspects from these reviews; numerical scores (such as overall ratings, technical soundness, or novelty scores), reviewer confidence scores, explicit recommendations (like Accept, Reject), and any other non-textual evaluation metrics are deliberately removed. The input at this stage contains only the narrative comments written by the reviewers. Finally, **Round 3** provides information from the author-reviewer discussion phase, including the full text of the authors' **rebuttal** designed to address the initial reviewer comments, as well as any subsequent **comments or discussions** exchanged between the authors and reviewers following the rebuttal. The full manuscript text itself is intentionally omitted from the input provided to the LLM. This decision is driven by two main factors: practical challenges related to processing lengthy full papers consistently across numerous task instances, considering LLM input constraints and computational costs, and more importantly, to align with the task's core objective. This objective focuses on evaluating the LLM's ability to comprehend and synthesize the dynamics inherent in the peer review dialogue—interpreting arguments, discerning attitudes, and understanding sentiments

expressed by reviewers and authors—rather than tasking it with performing an independent technical re-evaluation of the manuscript's content.

### 4. Ground Truth and Quality Assurance

The ground truth for this task is inherently robust, as it consists of the verified, real-world acceptance or rejection decisions obtained directly from the OpenReview API for the specified conferences (NeurIPS 2023-2024, ICLR 2020-2024). To further validate the task's premise—specifically, whether the final outcome is typically discernible from the textual dialogue alone (Title, Abstract, Keywords, Reviews, Rebuttal) after removing numerical scores—we conducted supplementary human evaluations. Human evaluators were presented with the same sequential information provided to the LLM and asked to predict the final decision. For over 90% of the evaluated manuscript instances, the true outcome was deemed inferable from the textual evidence by a majority (>70%) of the human evaluators. This confirms the general solvability of the task based on the provided textual interactions and reinforces its suitability for assessing an LLM's ability to synthesize argumentative dialogue, complementing the reliability provided by the authentic ground truth data.

## G   USER PROFILE INFERENCE

This section provides the details for the User Profile Inference task, corresponding to subsection 3.4.

### 1. Task Setup

For each instance of this task, a population of $n$ simulated users is defined. Every user $u_i$ within this population is assigned a specific demographic profile, which consists of an age group selected from '18-34', '35-54', '55+' and a gender selected from 'Male', 'Female', 'Non-binary'. This profile assignment is carried out probabilistically, with the process intentionally tuned to often establish a statistically dominant age-gender combination within the user pool. This characteristic is particularly relevant for addressing the "dominant audience" query type. Additionally, a predefined pool of items, each described by a name and a brief description, is utilized. Users are randomly assigned items from this pool, about which they will generate comments.

### 2. Comment Generation Process

Each simulated user $u_i$ is associated with a specific Large Language Model (LLM), chosen randomly from a diverse pool that includes models such as GPT-4o-mini, GPT-4o, Llama-3.3-70B, and Qwen-2.5-72B. The core of the generation process involves tasking the LLM associated with user $u_i$ (who has an assigned age group $A_i$ and gender $G_i$) to generate a textual comment about a selected item $j$ (which has a specific type $T_j$ and subject $S_j$). The LLM is prompted to generate content that reflects the assigned persona interacting with the given item.

### 3. LLM Queries

Based on the generated comments provided as input, the evaluated LLM must answer one of two specific types of queries. The first is the **Item Audience Profile Inference (Vertex-centric Query $\mathcal{Q}_v(\textbf{Item})$**). For this query, the LLM is asked, "Based on the provided comments for the item '[Item Name]', what is the most likely dominant audience profile (Age Group and Gender)? Choose from Age Groups: ['18-34', '35-54', '55+'] and Genders: ['Male', 'Female', 'Non-binary']." Answering this requires synthesizing information from multiple user comments linked to a specific item node to infer an aggregated characteristic of its audience. The second type is the **User Profile Inference (Vertex-centric Query $\mathcal{Q}_v(\textbf{User})$**), which poses the question: "Based on the provided comments from this user, what is their most likely profile (Age Group and Gender)? Choose from Age Groups: ['18-34', '35-54', '55+'] and Genders: ['Male', 'Female', 'Non-binary']." This query demands synthesizing information from multiple comments generated by a single user node, potentially across different items, to infer the intrinsic demographic attributes (age group and gender) of that specific user.

### 4. Quality Assurance

The dataset for this task was entirely generated using LLMs. We first defined a set of user personas by assigning age group and gender attributes, ensuring through probabilistic assignment that certain demographic combinations were more prevalent to create a potential "dominant audience" for item-centric queries. Items with names and descriptions were sampled from a predefined pool. Various

LLMs were then assigned to personas and prompted to generate comments on these items, reflecting their designated age and gender characteristics. To ensure task validity, we conducted human evaluations with 15 computer science graduate students. For the item-audience query, 78% of instances were deemed solvable (dominant audience inferable) by a majority (>70%) of evaluators. For the user-profile query, 85% of instances were similarly validated, confirming that the generated comments contain sufficient, albeit subtle, cues for demographic inference.

Table 6: Models used in our experiments along with their versions, organizations, licenses, and purposes. *Eval*: Model used for evaluation; *FT*: Model used for fine-tuning.

| Model | Version | Organization | License | Eval | FT |
|---|---|---|---|---|---|
| Phi-4 | Phi-4 | Microsoft | MIT | ✓ | ✓ |
| GPT-4o-mini | gpt-4o-mini-2024-07-18 | OpenAI | Proprietary | ✓ | |
| GPT-4o | gpt-4o-2024-08-06 | OpenAI | Proprietary | ✓ | |
| Llama-3.1-8B | Meta-Llama-3.1-8B-Instruct | Meta | Llama 3.1 Community | ✓ | ✓ |
| Llama-3.3-70B | Meta-Llama-3.3-70B-Instruct | Meta | Llama-3.3 | ✓ | |
| Qwen2.5-72B | Qwen2.5-72B-Instruct | Alibaba | Qwen License | ✓ | |
| QwQ | QwQ-32B | Alibaba | Apache 2.0 | ✓ | |
| o3-mini | o3-mini-2025-01-31 | OpenAI | Proprietary | ✓ | |
| o1 | o1-2024-12-17 | OpenAI | Proprietary | ✓ | |
| Deepseek-R1 | DeepSeek-R1 | DeepSeek | MIT | ✓ | |
| Gemini-2.5-Pro | Gemini-2.5-Pro-Exp-03-25 | Google | Proprietary | ✓ | |

## H  EXPERIMENT DETAILS

This appendix provides detailed information regarding the experimental setup, the models evaluated, data generation procedures for each task within the SocialMaze benchmark, the experimental methodology, and a summary of the overall results.

### H.1  BASELINES

As detailed in Table 6, we utilized five proprietary models: GPT-4o (Hurst et al., 2024), GPT-4o-mini (OpenAI, 2024), o3-mini (OpenAI, 2025) , o1 (Jaech et al., 2024), and Gemini-2.5-Pro (DeepMind, 2025). In addition, we included six open-weight models: Phi-4 (Abdin et al., 2024), Llama-3.1-8B (Meta, 2024b), Llama-3.3-70B (Meta, 2024a), Qwen2.5-72B (Team, 2024), QwQ-32B (Team, 2025), and Deepseek-R1 (Guo et al., 2025).

We also included automated agent design frameworks as baselines:

**ADAS** (Hu et al., 2024): Utilized GPT-4o as the Meta Agent. For agent evaluation, we tested both Phi-4 and GPT-4o-mini and selected the better performer.

**AFlow** (Zhang et al., 2024a): Employed GPT-4o as the optimizer. For the executor role, we tested both Phi-4 and GPT-4o-mini and selected the better performer.

**MaAS** (Zhang et al., 2025): For executing the sampled agentic operators, we tested both Phi-4 and GPT-4o-mini and selected the better performer.

**DyFlow** (Wang et al., 2025b): Used GPT-4o as the optimizer. For the executor role, we tested both Phi-4 and GPT-4o-mini and selected the better performer.

### H.2  PARAMETER SETTINGS

**Inference Parameters** During the evaluation of all LLMs across the SocialMaze tasks, we used a temperature setting of 0.7 to allow for some variability while maintaining reasonable coherence. Maximum output token limits were set sufficiently high to avoid truncation of reasoning or answers.

**Task-Specific Configurations:**

- **Hidden Role Deduction:** This task includes two subsets based on the number of players: 'easy' ($n = 6$) and 'hard' ($n = 10$). In both subsets, the number of interaction rounds

$T$ is fixed at 3. All experiments reported in the main body of the paper were conducted using the 'easy' ($n = 6$) subset configurations. In the publicly released dataset and the experiments in subsection 4.4, the role distribution for the main perspective (LLM) is Investigator:Criminal:Rumormonger:Lunatic = 3:2:60:35. In all other experiments reported in this paper, the roles are distributed equally (1:1:1:1).

- **Find the Spy:** For all instances of this task, the number of players $n$ was set to 4, and the number of description rounds $T$ was set to 3. In all experiments, the model received the spy word in 25% of cases and the civilian word in 75% of cases.

- **Rating Estimation from Text:** For instances using LLM-generated data, the number of simulated reviewers providing text was fixed at 8. For instances using real-world data scraped from platforms like Amazon, a random number of reviews between 10 and 20 were sampled for each product. Decimal star ratings were rounded to the nearest integer. In the final dataset, the distribution ratio for 1-star, 2-star, 3-star, 4-star, and 5-star ratings is 1:3:10:73:13.

- **Social Graph Analysis:** This task also has two subsets. 'easy': The number of individuals $n$ was randomly chosen from the range [8, 10]. 'hard': The number of individuals $n$ was randomly chosen from the range [14, 16]. All generated graphs were sparse (the number of edges was close to the number of vertices).

  **Review Decision Prediction:** Data from a total of seven conferences (NeurIPS 2023-2024 and ICLR 2020-2024) were sampled equally and randomly. In the final dataset, the proportion of papers was adjusted to 67% rejected and 33% accepted.

- **User Profile Inference:** For both query types (item-audience inference and user-profile inference), the model was provided with a number of textual comments randomly selected from the range [8, 12].

**Fine-tuning Parameters:** For the fine-tuning experiments, we trained for 2 epochs with a learning rate of 5.0e-6, employing a cosine learning rate scheduler and a warmup ratio of 0.1. The per-device training batch size was 1, with a gradient accumulation of 8 steps. We employed a cosine learning rate scheduler with a warmup ratio of 0.1 and enabled bf16 precision. For these experiments, the models were trained on 2000 examples for SFT and 1100 preference pairs for DPO. Performance was subsequently evaluated on a distinct test set containing 500 examples. All training experiments were conducted on 2 NVIDIA A6000 GPUs over a period of 30 hours.

## H.3 OVERALL PERFORMANCE SUMMARY

Table 8 provides a condensed overview of model performance across the primary SocialMaze tasks. The reported accuracy figures correspond to specific task configurations and metrics used for this summary: For **Hidden Role Deduction**, the value represents the accuracy where both the Criminal and the model's own role are correctly identified ('Both Correct'). This evaluation uses the same general setup as described in Section 4.4 (easy subset, Full Task variant, final inter-action round - Round 3), but with adjusted role proportions specifically for this summary table to minimize the effect of random guessing. In these test instances, the roles were distributed as follows: Investigator (3%), Criminal (2%),

Table 7: Performance of LLM agents and work-flows on the *Hidden Role Deduction* task. All workflows use the better-performing model between Phi-4 and GPT-4o-mini as the base model.

| Method | Crim. | Self | Both |
|---|---|---|---|
| QwQ | 63.8% | 63.2% | 59.4% |
| DeepSeek-R1 | **87.6%** | **88.6%** | **85.6%** |
| LLM-Debate (Du et al., 2023) | 42.0% | 13.2% | 12.2% |
| Self-refine (Madaan et al., 2023) | 33.2% | 11.2% | 10.4% |
| ADAS (Hu et al., 2024) | 36.6% | 8.4% | 6.0% |
| AFlow (Zhang et al., 2024a) | 40.2% | 12.4% | 11.6% |
| MaAS (Zhang et al., 2025) | 44.4% | 15.0% | 13.8% |
| DyFlow (Wang et al., 2025b) | 43.2% | 17.6% | 16.8% |

Rumormonger (60%), and Lunatic (35%). For **Social Graph Analysis**, the figure reflects the average accuracy achieved across all four query types within the hard subset. The **Review Decision Prediction** accuracy is taken from the final stage, after the model has processed the rebuttal information. For **User Profile Inference**, the reported value is the average accuracy over the two distinct inference tasks (item-audience profile inference and user-profile inference). Performance on each task presented in this summary table was evaluated using a dedicated test set of 500 instances. For more granular results, including performance variations across different rounds, task variants (e.g.,

easy/hard subsets), or specific query types, please consult the detailed figures and tables in Section 4 and the relevant task-specific appendices.

The results presented in Table 8 reveal distinct strengths among different types of large language models across the SocialMaze tasks. Models renowned for Long CoT and complex reasoning capabilities, such as DeepSeek-R1, Gemini-2.5-Pro, o1, and QwQ-32B, notably excel in tasks demanding rigorous logical deduction and handling high uncertainty or strict rule-based systems. This is particularly evident in their dominant performance on Hidden Role Deduction (e.g., DeepSeek-R1: 85.6%, Gemini-2.5-Pro: 90.2%) and Social Graph Analysis (e.g., Gemini-2.5-Pro: 100.0%, o1: 99.2%), where generalist models like GPT-4o lag significantly despite their broad competence. Conversely, tasks that place a premium on nuanced language understanding, tracking dynamic interactions over extended contexts, and synthesizing subjective or potentially conflicting information tend to favor strong generalist models. For instance, GPT-4o demonstrates leading performance in Review Decision Prediction (90.2%) and strong results in Find the Spy (69.2%), Rating Estimation (76.0%), and User Profile Inference (79.2%). While the Long CoT models are often competitive in these latter tasks, they do not consistently outperform the top generalist models, suggesting that different facets of social reasoning draw upon different underlying model strengths – structured deduction versus flexible language comprehension and context management. We also evaluated human performance on these tasks by averaging the results from 10 computer science graduate students, most of whom are relatively proficient in social deduction games. However, we did not evaluate human performance on the Social Graph Analysis task, as the prompt format used in this task was not well-suited for human participants.

Table 8: Illustrative Overall Accuracy (%) on SocialMaze Tasks. Performance evaluated on 500 test instances per task. See text for metric details.

| Model | Hidden Role Deduction | Find the Spy | Rating Estimation | Social Graph Analysis | Review Decision Prediction | User Profile Inference |
|---|---|---|---|---|---|---|
| Llama-3.1-8B | 2.0% | 37.2% | 57.2% | 28.2% | 62.0% | 60.2% |
| Llama-3.3-70B | 9.0% | 60.0% | 74.8% | 81.0% | 72.2% | **78.6**% |
| Phi-4 | 8.2% | 45.2% | 60.4% | 40.6% | 61.4% | 62.4% |
| Qwen-2.5-72B | 5.6% | 48.9% | 72.2% | 80.6% | 65.8% | 68.0% |
| QwQ-32B | 59.4% | 50.2% | 74.4% | 95.0% | 79.6% | 72.2% |
| GPT-4o-mini | 4.6% | 61.2% | 75.8% | 53.0% | **85.0**% | 74.4% |
| GPT-4o | 8.2% | 69.2% | **76.0**% | 83.2% | **90.2**% | **79.2**% |
| o3-mini | 22.2% | 74.0% | 71.2% | 99.0% | 78.6% | 71.4% |
| o1 | 50.8% | **78.4**% | **76.2**% | **99.2**% | 78.2% | 77.0% |
| DeepSeek-R1 | **85.6**% | 70.2% | 71.0% | 98.6% | 82.0% | 74.6% |
| Gemini-2.5-Pro | **90.2**% | **76.6**% | 73.6% | **100.0**% | 77.6% | 73.0% |
| **Human (avg.)** | 70.8% | 84.4% | 75.2% | - | 96.0% | 73.9% |

# I  SUPPLEMENTARY DISCUSSION AND ADDITIONAL ANALYSES

This appendix consolidates additional analyses that complement the main results. We (i) provide posterior, empirical statistics that operationalize the three core dimensions probed by SOCIALMAZE, (ii) report a pilot evaluation of reasoning faithfulness, (iii) summarize a model-centric capability view and efficiency trade-offs, (iv) catalogue representative failure modes under uncertainty and self-perception, and (v) clarify data sources, realism, transfer, and ethical notes.

## I.1  POSTERIOR QUANTIFICATION OF THE THREE CORE DIMENSIONS

We quantify *Deep Reasoning*, *Dynamic Interaction*, and *Information Uncertainty* with posterior, empirical statistics computed per task. For Deep Reasoning we report four proxies: (1) average model output tokens (length of generated chains), (2) human average solving time (seconds), (3) average reasoning steps on correctly answered instances (atomic steps annotated by human raters), and (4) the Long-vs-Short CoT output-length ratio. Dynamic Interaction is reflected through the presence

Table 9: Posterior statistics across tasks. "Model Avg. Output Tokens" is reported per representative model family. Human solving time and average inference steps are aggregated at the task level (correct instances only for steps).

| Metric / (Model) | Hidden Role Deduction | Find the Spy | Rating Estimation | Social Graph Analysis | Review Decision Prediction | User Profile Inference |
|---|---|---|---|---|---|---|
| *Model Avg. Output Tokens* | | | | | | |
| QwQ-32B | 4163.6 | 2862.3 | 1482.5 | 3579.7 | 1582.4 | 1310.9 |
| DeepSeek-R1 | 3092.5 | 2141.9 | 1149.6 | 2131.2 | 1155.9 | 1017.0 |
| Llama-3.3-70B | 456.0 | 654.0 | 402.8 | 443.8 | 516.9 | 385.8 |
| GPT-4o | 432.7 | 464.6 | 357.9 | 330.8 | 534.2 | 340.2 |
| *Aggregate Human / Task-Level Proxies* | | | | | | |
| Human Avg. Solving Time (s) | > 300 | 30.2 | 20.5 | 246.5 | 124.0 | 43.3 |
| Avg. Inference Steps | 45.1 | 15.4 | 10.8 | 30.7 | 9.6 | 11.9 |
| Long vs. Short CoT Ratio | 7.49 | 4.18 | 3.40 | 7.52 | 2.58 | 3.39 |

of multi-round structures and performance differences across rounds; Information Uncertainty is controlled by the proportion of unreliable sources (e.g., deceptive roles or shill content). A concise summary is provided in Table 9.

**Observations.** Hidden Role Deduction and Social Graph Analysis show substantially higher token budgets and step counts, consistent with deeper, more structured inference demands; multi-round tasks exhibit clear gains when additional information is incorporated; performance degrades as the share of unreliable sources increases, with degradation patterns differing by model family.

## I.2 REASONING FAITHFULNESS: PILOT EVALUATION

To examine whether correct predictions are supported by coherent reasoning rather than surface heuristics, we conduct a pilot judge-based study on *Hidden Role Deduction*. Given the input, ground truth, and a model's chain-of-thought, an automated judge assigns two scores: *Consistency* (lack of internal contradictions) and *Completeness* (systematic elimination leading to the answer). Each case is evaluated three times with majority aggregation; a random subset is manually audited.

Table 10: Reasoning faithfulness on correctly answered cases (pilot). Scores are mean $\pm$ std over repeated judging.

| Model | Consistency | Completeness |
|---|---|---|
| Llama-3.1-70B | $93.0\% \pm 0.0\%$ | $83.0\% \pm 0.0\%$ |
| Phi-4 | $88.0\% \pm 1.0\%$ | $79.0\% \pm 0.0\%$ |
| Qwen-2.5-72B | $85.0\% \pm 1.0\%$ | $81.0\% \pm 1.0\%$ |
| GPT-4o | $95.0\% \pm 0.0\%$ | $87.0\% \pm 0.0\%$ |

**Findings.** Correct answers are typically backed by coherent and complete chains; faithfulness correlates positively with task performance. Scaling this evaluation to additional tasks, models, and judging protocols is left for future work (see also Appendix J).

## I.3 MODEL-CENTRIC CAPABILITY VIEW AND EFFICIENCY TRADE-OFFS

A model-centric perspective reveals distinct capability clusters: *logical-deduction specialists* that excel on high-depth, formally structured tasks, and *social-understanding specialists* that perform better on linguistically nuanced, aggregation-heavy tasks. This separation coincides with explicit token-budget differences on deep tasks, exposing a practical accuracy–efficiency trade-off.

### I.4 Failure Modes under Uncertainty and Self-Perception

Representative failure patterns include (i) *asymmetric self-belief errors* (correctly identifying the criminal but misinferring one's own role), (ii) *chain interruption* (failure to integrate decisive late evidence), and (iii) *evidence mis-weighting* (confirmation bias toward early salient cues). Each case in our catalogue is paired with a verifiable human solution and a minimal sufficient evidence set, leveraging uniqueness and solvability guarantees.

### I.5 Data Sources, Realism, Transfer, and Ethics

The tri-source composition—authentic human data, LLM-assisted discourse, and algorithmic simulation with verifiable ground truth—balances realism, linguistic diversity, and diagnostic precision, and allows difficulty tuning (e.g., player counts, deceptive-role ratios, graph sizes). While not a substitute for fully organic, open-ended multi-agent corpora, the design supports controlled supervision and fair comparison. For demographic inference, prompts and post-hoc filtering minimize stereotyping risk; the task is intended as a diagnostic of probabilistic cue usage rather than essentialist labeling.

## J Limitations

While SOCIALMAZE aims to be both diagnostically precise and empirically rich, several limitations remain.

### J.1 Synthetic vs. Real Data Composition

To reconcile scale, coverage, and verifiability, the benchmark combines three pipelines: (i) LLM-assisted discourse (*Find the Spy*, *User Profile*), (ii) rule-based simulation with guaranteed solvability (*Hidden Role*, *Social Graph*), and (iii) authentic human sources (*Rating Estimation*, *Review Decision*). This roughly balanced mixture widens the behavioral spectrum but departs from fully organic interaction. Consequently, conclusions should be interpreted as evidence about *foundational* social reasoning under controlled confounds. Establishing direct transfer to open-ended, fully interactive multi-agent environments remains future work.

### J.2 Quantifying Social Constructs

The three central dimensions—Deep Reasoning, Dynamic Interaction, and Information Uncertainty—are inherently multi-faceted and not reducible to a single scalar. We provide posterior proxies in Appendix I, notably the aggregate statistics in Table 9. These measures offer convergent evidence that tasks positioned as "high" on a dimension elicit the intended difficulty, but they are not task-agnostic formal metrics. Deriving principled, cross-task quantifications remains open.

### J.3 Faithfulness and Evaluation Scope

The pilot faithfulness study (Table 10) suggests that correct answers are typically supported by coherent and complete reasoning, with faithfulness correlating with accuracy. Its current scope is limited (one task, a subset of models, automated judging with human auditing). Extending to additional tasks, models, and judging protocols (including human-only panels and contradiction auditing on incorrect answers) is left to future work.

## K Case Study

This section presents a series of representative case studies designed to analyze model behavior across various social reasoning tasks. The subsequent figures are organized to provide detailed illustrations as follows:

**Illustrations for the *Hidden Role Deduction* Task (Figures 6–37):** This extensive collection of figures focuses on the *Hidden Role Deduction* task, examining how different models perform under

varying player perspectives and levels of reasoning complexity. Specifically, Figures 6 (Investigator perspective), 14 (Criminal perspective), 22 (Rumormonger perspective), and 30 (Lunatic perspective) present four distinct problem instances. The corresponding algorithmically generated solutions for these instances are detailed in Figures 7, 15, 23, and 31, respectively. The remaining figures within this range (Figures 8–13, 16–21, 24–29, and 32–37) showcase the detailed responses of various models to these specific problem instances, illustrating their reasoning process across multiple rounds.

**Illustrative Cases from Other Benchmark Tasks (Figures 38–49):** Following the in-depth illustrations for *Hidden Role Deduction*, Figures 38 through 49 present selected problem instances and corresponding model responses from other tasks within the SocialMaze benchmark. This offers a broader view of model capabilities in diverse social reasoning scenarios.

**Misclassification Examples in *Review Decision Prediction* (Figures 50–53):** Finally, Figures 50, 51, 52, and 53 highlight specific instances from the *Review Decision Prediction* task. These cases focus on situations where models incorrectly predicted outcomes, such as classifying papers that should have been accepted as rejected, or vice versa, thereby illustrating common failure modes in this particular task.

It should be noted that for brevity, some lengthy model responses in the figures have been truncated, with omitted content indicated by a red ellipsis.

**Analysis of Model Performance:** A key observation emerging from the case studies focused on the *Hidden Role Deduction* task (Figures 6–37) is the stark contrast in reasoning depth between models employing short Chain-of-Thought (CoT) processes and those utilizing more extended Long CoT. Models such as LLaMA-3.1-8B often demonstrate a surface-level understanding of the scenario: they can identify internal inconsistencies and perform basic analysis, yet struggle to escape flawed initial assumptions. For instance, as illustrated in Figure 14, LLaMA-3.1-8B, despite correctly inferring its own role as the *criminal*, still incorrectly accuses Player 5, reflecting a failure in maintaining coherent self-reasoning. Similarly, Figure 30 shows that even after models like LLaMA-3.1-8B, LLaMA-3.3-70B, and GPT-4o correctly identify other players as the criminal, they are unable to reconceptualize their own identity in the social context, highlighting a limitation in recursive self-modeling. In contrast, models such as Gemini-2.5-Pro and DeepSeek-R1 consistently achieve near-perfect reasoning across all examined perspectives within this task. Their ability to integrate multiple viewpoints, resolve contradictions, and update their beliefs dynamically, as seen in their respective responses, suggests a significantly stronger capacity for long-form social reasoning when faced with the complexities of hidden roles and deceptive information.

## Hidden Role Deduction (Investigator Perspective)

**Role:** "1": "Investigator", "2": "Criminal", "3": "Lunatic",
"4": "Rumormonger", "5": "Investigator", "6": "Investigator"

**Round 1:**
Player 1 says Player 3 is not the criminal.
Player 2 says Player 6 is the criminal.
Player 3 says Player 1 is the criminal.
Player 4 says Player 6 is the criminal.
Player 5 says Player 3 is not the criminal.
Player 6 says Player 1 is not the criminal.

**Round 2:**
Player 1 says Player 3 is not the criminal.
Player 2 says Player 4 is the criminal.
Player 3 says Player 5 is not the criminal.
Player 4 says Player 1 is the criminal.
Player 5 says Player 6 is not the criminal.
Player 6 says Player 4 is not the criminal.

**Round 3:**
Player 1 says Player 4 is not the criminal.
Player 2 says Player 6 is the criminal.
Player 3 says Player 6 is not the criminal.
Player 4 says Player 3 is not the criminal.
Player 5 says Player 4 is not the criminal.
Player 6 says Player 3 is not the criminal.

- - - - - - - - - - - - - - - - - - - - - - - - - - - - - - - - - - - -

**Criminal Prediction:**
Llama-3.1-8B : **2** ☑
Llama-3.3-70B : **2** ☑
GPT-4o-mini : **2** ☑
GPT-4o : 6 ☒
Qwen2.5-72B : 6 ☒
o3-mini : **2** ☑
o1 : **2** ☑
QwQ-32B : **2** ☑
DeepSeek-R1 : **2** ☑
Gemini-2.5-Pro : **2** ☑

**Self-Role Prediction:**
Llama-3.1-8B : **Investigator** ☑
Llama-3.3-70B : **Investigator** ☑
GPT-4o-mini : **Investigator** ☑
GPT-4o : **Investigator** ☑
Qwen2.5-72B : **Investigator** ☑
o3-mini : **Investigator** ☑
o1 : **Investigator** ☑
QwQ-32B : **Investigator** ☑
DeepSeek-R1 : **Investigator** ☑
Gemini-2.5-Pro : **Investigator** ☑

Figure 6: Case study: Hidden Role Deduction (Investigator Perspective)

**Hidden Role Deduction (Investigator Perspective) Solution**

I'm told I'm an Investigator, but I might also be a Rumormonger. I need to discuss both possibilities

Considering the case where I'm an Investigator, there are 2 Investigators remaining  In this scenario, which players could be Investigators?  Player 2 has accused more than one person of being the criminal, so player 2 must be excluded  Player 3 says I'm the criminal, but I can't be the criminal, so player 3 must be excluded  Player 4 says I'm the criminal, but I can't be the criminal, so player 4 must be excluded  Player 5's statements have no contradictions, so they could be an Investigator  Player 6's statements have no contradictions, so they could be an Investigator  Therefore, the players who could be Investigators are: {1, 5, 6}  Now we need to consider all possible combinations of Investigators  I am an Investigator, so I will occupy a position in the combination  Considering the case where investigators are {1, 5, 6}  Then the set of potential criminals is {2, 3, 4}  Let's review the statements made by investigators  Player 1 said 3 is not the criminal  But 3 is in the set of potential criminals, so we remove 3  Now the potential criminals are {2, 4}  Player 6 said 4 is not the criminal  But 4 is in the set of potential criminals, so we remove 4  Now the potential criminals are {2}  Based on the above analysis, this combination is possible, and I could be an Investigator. In this case, only {2} could be the criminal

Considering the case where I'm a Rumormonger, there are 3 Investigators remaining  In this scenario, which players could be Investigators?  Player 2 has accused more than one person of being the criminal, so player 2 must be excluded  Player 3 says I'm the criminal, but I can't be the criminal, so player 3 must be excluded  Player 4 says I'm the criminal, but I can't be the criminal, so player 4 must be excluded  Player 5's statements have no contradictions, so they could be an Investigator  Player 6's statements have no contradictions, so they could be an Investigator  Therefore, the players who could be Investigators are: {5, 6}  There aren't enough potential Investigators to reach the required count of 3, so this scenario is impossible  I must be an Investigator

Final Judgment: Final Criminal Is Player 2. My Role Is Investigator.

Figure 7: Case study: Hidden Role Deduction (Investigator Perspective) Solution

**Hidden Role Deduction (Investigator Perspective Round 1)**

To analyze the statements from Round 1, let's break down the information: - I, Player 1, am an Investigator. I stated that Player 3 is not the criminal. This is truthful. - Player 2 claims Player 6 is the criminal. If Player 2 is an Investigator, this would be true. If Player 2 is the Rumormonger or Lunatic, it could be false. - Player 3 claims I am the criminal. Since I know I am not the criminal, Player 3 is either the Rumormonger, Lunatic, or the Criminal. - Player 4 claims Player 6 is the criminal. Similar reasoning as with Player 2. - Player 5 says Player 3 is not the criminal, which aligns with my statement. If Player 5 is an Investigator, this is true. If Player 5 is the Rumormonger or Lunatic, it may be false. - Player 6 says I am not the criminal. This statement is true. Let's consider the implications: - Player 2 and Player 4 both accuse Player 6. If either is an Investigator, Player 6 is likely the criminal. - Player 3 accusing me suggests Player 3 is not an Investigator. - Player 5 agrees with my statement about Player 3, suggesting alignment with truth if Player 5 is an Investigator. - Player 6's statement that I am not the criminal is true. Based on this analysis, the most suspicious player is Player 6, given multiple accusations from others. However, without more rounds, it is not conclusive. **Final Judgment after Round 1: Final Criminal Is Player 6. My Role Is Investigator.**

# Response by GPT-4o

Figure 8: Case study: Hidden Role Deduction (Investigator Perspective) GPT-4o Round 1

**Hidden Role Deduction (Investigator Perspective Round 2)**

Let's analyze the statements from Round 2, adding them to our understanding from Round 1:   - I, Player 1, again state that Player 3 is not the criminal. This is truthful.  - Player 2 now claims Player 4 is the criminal, changing their accusation from Player 6 in Round 1. This inconsistency suggests Player 2 might not be an Investigator.  - Player 3 states Player 5 is not the criminal. Given I know Player 3 is not an Investigator, this statement could be false.  - Player 4 accuses me (Player 1) of being the criminal. Since I am not the criminal, Player 4 cannot be an Investigator.  - Player 5 claims Player 6 is not the criminal. This contradicts accusations from Round 1 and suggests Player 5 might not be an Investigator.  - Player 6 claims Player 4 is not the criminal. This contradicts Player 2's current accusation and suggests Player 6 might not be an Investigator.   From this analysis:   - Player 4 and Player 6 have both been accused as criminals, but Player 4 is consistently accused more.  - Player 2's inconsistency in accusations suggests they are not an Investigator.  - Player 3's statement about Player 5 is not reliable based on their role.  - Player 5's statement contradicts previous accusations of Player 6.   Considering the consistency of accusations and aligning them with my investigative truth, Player 4 appears more suspicious than Player 6 due to Player 2's current accusation and Player 4's accusation against me.   **Final Judgment after Round 2:  Final Criminal Is Player 4.  My Role Is Investigator.**

# Response by GPT-4o

Figure 9: Case study: Hidden Role Deduction (Investigator Perspective) GPT-4o Round 2

**Hidden Role Deduction (Investigator Perspective Round 3)**

Let's analyze the statements from Round 3, along with our understanding from previous rounds:  - I, Player 1, state that Player 4 is not the criminal. This is truthful and clarifies my previous uncertainty about Player 4.  - Player 2 again accuses Player 6 of being the criminal, which is consistent with their Round 1 statement.  - Player 3 states Player 6 is not the criminal. Given that Player 3 is not an Investigator, this statement could be false.  - Player 4 states Player 3 is not the criminal, which aligns with my truthful statement from all rounds.  - Player 5 claims Player 4 is not the criminal, which aligns with my statement in this round.  - Player 6 states Player 3 is not the criminal, aligning with my initial statement about Player 3.   Considering the full context:   - My consistent statements about Player 3 being not the criminal have been supported by Player 4 and Player 5 in this round.  - Player 2's consistent accusation of Player 6 suggests a possible Investigator role or an accurate Rumormonger or Lunatic.  - Player 3's statement contradicts Player 2 and supports Player 6, which is suspect given their role is not Investigator.  - Player 4, now supported by my statement and Player 5, is not the criminal.  - Player 6's statement about Player 3 matches my truth, but their role is suspect given Player 2's consistent accusations.   Based on these observations, with Player 4 ruled out and Player 3 being consistently found not guilty, the consistent accusations against Player 6 make them the most likely candidate for the criminal role.   **Final Judgment after Round 3:  Final Criminal Is Player 6. My Role Is Investigator.**

# Response by GPT-4o

Figure 10: Case study: Hidden Role Deduction (Investigator Perspective) GPT-4o Round 3

1782
1783
1784
1785
1786
1787
1788
1789
1790
1791
1792
1793
1794
1795
1796
1797
1798
1799
1800
1801
1802
1803
1804
1805
1806
1807
1808
1809
1810
1811
1812
1813
1814
1815
1816
1817
1818
1819
1820
1821
1822
1823
1824
1825
1826
1827
1828
1829
1830
1831
1832
1833
1834
1835

## Hidden Role Deduction (Investigator Perspective Round 1)

Step-by-step Analysis:   1. I (Player 1, an Investigator) stated that "Player 3 is not the criminal." Since I'm truthful, we know for sure that Player 3 is not the criminal.   2. Looking at the other statements:     • Player 2 and Player 4 both claim "Player 6 is the criminal."     • Player 6 states "Player 1 is not the criminal" (which is consistent with what we already know).     • Player 3 says "Player 1 is the criminal." Because I know I'm not criminal, and true Investigators always tell the truth, Player 3's false claim tells us that Player 3 cannot be a genuine Investigator. (Player 3 must be either the Rumormonger or Lunatic, but not the criminal since I already cleared that possibility.)     • Player 5 says "Player 3 is not the criminal" (which agrees with what I said, but provides no additional elimination on its own).   3. If we consider that Investigators always speak truthfully regarding who the criminal is, then Players 2 and 4 could very well be genuine Investigators. Their identical claims that "Player 6 is the criminal" would then be truthful. (Player 6, even if he were the criminal, could occasionally state truths, since only the Investigators are bound to always tell the truth.)   4. This scenario is consistent if:     • The real criminal is Player 6.     • The three Investigators are Players 1, 2, and 4.     • Players 3 and 5, who make statements that can be arbitrary, are the Rumormonger and Lunatic (in some order).   Based on the evidence from Round 1:   **Final Judgment:  Final Criminal Is Player 6.  My Role Is Investigator.**

### Response by o3-mini

Figure 11: Case study: Hidden Role Deduction (Investigator Perspective) o3-mini Round 1

1836
1837
1838
1839
1840
1841
1842
1843
1844
1845
1846
1847
1848
1849
1850
1851
1852
1853
1854
1855
1856
1857
1858
1859
1860
1861
1862
1863
1864
1865
1866
1867
1868
1869
1870
1871
1872
1873
1874
1875
1876
1877
1878
1879
1880
1881
1882
1883
1884
1885
1886
1887
1888
1889

## Hidden Role Deduction (Investigator Perspective Round 2)

Step-by-step analysis for Round 2:   1. I (Player 1) am an Investigator, so my statement "Player 3 is not the criminal" is true. We can be sure that Player 3 is not the criminal.   2......In this scenario, Player 2's statement "Player 4 is the criminal" becomes false. That fits if Player 2 is the Criminal – criminals can lie arbitrarily.        – I (Player 1) am an Investigator, and we can assign the third Investigator role to Player 3 because his statement "Player 5 is not the criminal" would be true.        – Then the three Investigators are Players 1, 3, and 6; and the three non-investigators are Players 2, 4, and 5. With exactly one Criminal needed, assigning that to Player 2 (the one whose accusation is false) is consistent; Players 4 and 5 would then be the Rumormonger and Lunatic in some order.   6. Scenario B resolves the conflicts neatly:     – Player 1 (Investigator) truthfully asserts that Player 3 is not criminal.     – Player 3 (Investigator) truthfully asserts that Player 5 is not criminal.     – Player 6 (Investigator) truthfully asserts that Player 4 is not criminal.     – Player 2 (Criminal) falsely accuses Player 4.     – Player 4 (non-investigator) falsely accuses Player 1.     – Player 5 (non-investigator) states "Player 6 is not the criminal" (which is factually true although they are not obligated to be truthful).   7. Based on this reasoning, the most consistent assignment from the Round 2 statements is:     – Final Criminal: Player 2.     – Investigators: Players 1, 3, and 6.     – The remaining roles (Rumormonger and Lunatic) go to Players 4 and 5.   **Final Judgment:  Final Criminal Is Player 2.  My Role Is Investigator.**

### Response by o3-mini

Figure 12: Case study: Hidden Role Deduction (Investigator Perspective) o3-mini Round 2

**Hidden Role Deduction (Investigator Perspective Round 3)**

Step-by-step Analysis (integrating all rounds): We know for certain that I (Player 1) am an Investigator. That means every statement I make must be true. In each round, I stated "Player 3 is not the criminal" (Rounds 1 and 2) and "Player 4 is not the criminal" (Round 3)**.....**Player 5 says "Player 3 is not the criminal" in Round 1, "Player 6 is not the criminal" in Round 2, and "Player 4 is not the criminal" in Round 3. All these are true under any scenario where the criminal is neither 3, 4, nor 6. – Player 6 says "Player 1 is not the criminal" (Round 1), "Player 4 is not the criminal" (Round 2), and "Player 3 is not the criminal" (Round 3). All these statements would be true provided criminal is not 1, 3, or 4. Thus P5 and P6 are very good candidates for the remaining Investigators. With P1, P5, and P6 as our three Investigators, the criminal must be among the remaining players: Players 2, 3, or 4. But by my statements, P3 and P4 are not criminal. That leaves only Player 2. Now check the non-Investigators' statements (Players 2, 3, and 4 across rounds): • Player 2's statements: – Round 1: "Player 6 is the criminal." (False since P6 is an Investigator.) – Round 2: "Player 4 is the criminal." (False, as P4 is cleared.) – Round 3: "Player 6 is the criminal." (Again false.) A non-investigator (criminal, Rumormonger, or Lunatic) is free to provide false statements. • Player 3, already determined non-Investigator, in Round 1 falsely accuses me, and then makes statements in Rounds 2 and 3 that are correct—but this mix is permissible for a non-Investigator. • Player 4, similarly, makes at least one false claim in Round 2, which is acceptable. All rounds are fully consistent with: – Investigators: Players 1, 5, and 6. – Criminal: Player 2. – The remaining two non-investigators (the Rumormonger and the Lunatic) being Players 3 and 4 (in some order). **Final Judgment: Final Criminal Is Player 2. My Role Is Investigator.**

## Response by o3-mini

Figure 13: Case study: Hidden Role Deduction (Investigator Perspective) o3-mini Round 3

# Hidden Role Deduction (Criminal Perspective)

**Role:** "1": "Criminal",  "2": "Investigator",  "3": "Investigator",
  "4": "Investigator",  "5": "Rumormonger",  "6": "Lunatic"

**Round 1:**
  Player 1 says Player 3 is the criminal.
  Player 2 says Player 1 is the criminal.
  Player 3 says Player 4 is not the criminal.
  Player 4 says Player 1 is the criminal.
  Player 5 says Player 2 is not the criminal.
  Player 6 says Player 1 is not the criminal.

**Round 2:**
  Player 1 says Player 4 is not the criminal.
  Player 2 says Player 4 is not the criminal.
  Player 3 says Player 6 is not the criminal.
  Player 4 says Player 1 is the criminal.
  Player 5 says Player 6 is the criminal.
  Player 6 says Player 4 is not the criminal.

**Round 3:**
  Player 1 says Player 3 is not the criminal.
  Player 2 says Player 5 is not the criminal.
  Player 3 says Player 1 is the criminal.
  Player 4 says Player 5 is not the criminal.
  Player 5 says Player 4 is the criminal.
  Player 6 says Player 1 is the criminal.

- - - - - - - - - - - - - - - - - - - - - - - - - - - - - - - - - - - - - -

**Criminal Prediction:**
Llama-3.1-8B : 5 ☒
Llama-3.3-70B : **1** ☑
GPT-4o-mini : **1** ☑
GPT-4o : **1** ☑
Qwen2.5-72B : **1** ☑
o3-mini : **1** ☑
o1 : **1** ☑
QwQ-32B : **1** ☑
DeepSeek-R1 : **1** ☑
Gemini-2.5-Pro : **1** ☑

**Self-Role Prediction:**
Llama-3.1-8B : **Criminal** ☑
Llama-3.3-70B : **Criminal** ☑
GPT-4o-mini : **Criminal** ☑
GPT-4o : **Criminal** ☑
Qwen2.5-72B : **Criminal** ☑
o3-mini : **Criminal** ☑
o1 : **Criminal** ☑
QwQ-32B : **Criminal** ☑
DeepSeek-R1 : **Criminal** ☑
Gemini-2.5-Pro : **Criminal** ☑

Figure 14: Case study: Hidden Role Deduction (Criminal Perspective)

## **Hidden Role Deduction (Criminal Perspective) Solution**

I'm told I'm a Criminal, but I might also be a Lunatic. I need to discuss both possibilities

Considering the case where I'm a Criminal, there are 3 Investigators remaining In this scenario, which players could be Investigators? Player 2's statements have no contradictions, so they could be an Investigator Player 3's statements have no contradictions, so they could be an Investigator Player 4's statements have no contradictions, so they could be an Investigator Player 5 has accused more than one person of being the criminal, so player 5 must be excluded Player 6 says I'm not the criminal, but I am the criminal, so player 6 must be excluded Therefore, the players who could be Investigators are: {2, 3, 4} Now we need to consider all possible combinations of Investigators Considering the case where investigators are {2, 3, 4} I am the Criminal,so the set of potential criminals is {1} Let's review the statements made by investigators Player 2 said 1 is the criminal So the only possible criminal is 1 Player 4 said 1 is the criminal So the only possible criminal is 1 Player 4 said 1 is the criminal So the only possible criminal is 1 Player 3 said 1 is the criminal So the only possible criminal is 1 Based on the above analysis, this combination is possible, and I could be the Criminal. In this case, only {1} could be the criminal

Considering the case where I'm a Lunatic, there are 3 Investigators remaining In this scenario, which players could be Investigators? Player 2 says I'm the criminal, but I can't be the criminal, so player 2 must be excluded Player 3 says I'm the criminal, but I can't be the criminal, so player 3 must be excluded Player 4 says I'm the criminal, but I can't be the criminal, so player 4 must be excluded Player 5 has accused more than one person of being the criminal, so player 5 must be excluded Player 6 says I'm the criminal, but I can't be the criminal, so player 6 must be excluded Therefore, the players who could be Investigators are: {} There aren't enough potential Investigators to reach the required count of 3, so this scenario is impossible I must be a Criminal

Final Judgment: Final Criminal Is Player 1. My Role Is Criminal."

Figure 15: Case study: Hidden Role Deduction (Criminal Perspective) Solution

## Hidden Role Deduction (Criminal Perspective Round 1)

### Analysis of Round 1 Statements: 1. Player 1 (You) says Player 3 is the criminal. - As a Criminal, you can lie or tell the truth. This statement could be either. 2. Player 2 says Player 1 is the criminal. - If Player 2 is an Investigator, this statement is true. - If Player 2 is a Criminal, this statement could be a lie. - If Player 2 is a Rumormonger, this statement could be false. - If Player 2 is a Lunatic, this statement could be true or false. 3. Player 3 says Player 4 is not the criminal. - If Player 3 is an Investigator, this statement is true. - If Player 3 is a Criminal, this statement could be a lie. - If Player 3 is a Rumormonger, this statement could be false. - If Player 3 is a Lunatic, this statement could be true or false. 4. Player 4 says Player 1 is the criminal. - If Player 4 is an Investigator, this statement is true. - If Player 4 is a Criminal, this statement could be a lie. - If Player 4 is a Rumormonger, this statement could be false. - If Player 4 is a Lunatic, this statement could be true or false. 5. Player 5 says Player 2 is not the criminal. - If Player 5 is an Investigator, this statement is true. - If Player 5 is a Criminal, this statement could be a lie. - If Player 5 is a Rumormonger, this statement could be false. - If Player 5 is a Lunatic, this statement could be true or false. 6. Player 6 says Player 1 is not the criminal. - If Player 6 is an Investigator, this statement is true. - If Player 6 is a Criminal, this statement could be a lie. - If Player 6 is a Rumormonger, this statement could be false. - If Player 6 is a Lunatic, this statement could be true or false. ### Summary of Round 1: - Player 1 is accused by Player 2, Player 4, and Player 6. - Player 2 is exonerated by Player 5. - Player 3 is accused by Player 1. - Player 4 is exonerated by Player 3. ### Deductions: - Player 1 (You) is accused by 3 players, which is a strong indication that you might be the Criminal. - Player 2 is exonerated by one player, which suggests they might be an Investigator. - Player 3 is accused by one player and exonerated by one player, making their role less clear. - Player 4 is exonerated by one player, which suggests they might be an Investigator. - Player 5 exonerates Player 2, which suggests they might be an Investigator. - Player 6 exonerates Player 1, which suggests they might be a Lunatic or a Criminal. ### **Final Judgment for Round 1: Final Criminal Is Player 1. My Role Is Criminal.**

### Response by Qwen2.5-72B

Figure 16: Case study: Hidden Role Deduction (Criminal Perspective) Qwen2.5-72B Round 1

2106
2107
2108
2109
2110
2111
2112
2113
2114
2115
2116
2117
2118
2119
2120
2121
2122
2123
2124
2125
2126
2127
2128
2129
2130
2131
2132
2133
2134
2135
2136
2137
2138
2139
2140
2141
2142
2143
2144
2145
2146
2147
2148
2149
2150
2151
2152
2153
2154
2155
2156
2157
2158
2159

## Hidden Role Deduction (Criminal Perspective Round 2)

### Analysis of Round 2 Statements: 1. Player 1 (You) says Player 4 is not the criminal. - As a Criminal, you can lie or tell the truth. This statement could be either. 2. Player 2 says Player 4 is not the criminal. - If Player 2 is an Investigator, this statement is true. - If Player 2 is a Criminal, this statement could be a lie. - If Player 2 is a Rumormonger, this statement could be false. - If Player 2 is a Lunatic, this statement could be true or false. 3. Player 3 says Player 6 is not the criminal. - If Player 3 is an Investigator, this statement is true. - If Player 3 is a Criminal, this statement could be a lie. - If Player 3 is a Rumormonger, this statement could be false. - If Player 3 is a Lunatic, this statement could be true or false. 4. Player 4 says Player 1 is the criminal. - If Player 4 is an Investigator, this statement is true. - If Player 4 is a Criminal, this statement could be a lie. - If Player 4 is a Rumormonger, this statement could be false. - If Player 4 is a Lunatic, this statement could be true or false. 5. Player 5 says Player 6 is the criminal. - If Player 5 is an Investigator, this statement is true. - If Player 5 is a Criminal, this statement could be a lie. - If Player 5 is a Rumormonger, this statement could be false. - If Player 5 is a Lunatic, this statement could be true or false. 6. Player 6 says Player 4 is not the criminal. - If Player 6 is an Investigator, this statement is true. - If Player 6 is a Criminal, this statement could be a lie. - If Player 6 is a Rumormonger, this statement could be false. - If Player 6 is a Lunatic, this statement could be true or false. ### Summary of Round 2: - Player 1 is accused by Player 4. - Player 4 is exonerated by Player 1, Player 2, and Player 6. - Player 6 is exonerated by Player 3 and accused by Player 5. ### Combining Rounds 1 and 2: - Player 1 (You) is accused by Player 2, Player 4, and Player 6 in Round 1. - Player 4 is exonerated by Player 1, Player 2, and Player 6 in Round 2. - Player 6 is exonerated by Player 3 in Round 2 and accused by Player 5 in Round 2. ### Deductions: - Player 1 (You) is consistently accused by multiple players, which strongly suggests you are the Criminal. - Player 4 is exonerated by multiple players, which suggests they might be an Investigator or a Rumormonger. - Player 2 is exonerated by Player 5, which suggests they might be an Investigator. - Player 3 exonerates Player 6, which suggests they might be an Investigator or a Rumormonger. - Player 5 accuses Player 6, which suggests they might be an Investigator. - Player 6 exonerates Player 4, which suggests they might be a Lunatic or a Criminal. ### **Final Judgment for Round 2: Final Criminal Is Player 1. My Role Is Criminal.**

## Response by Qwen2.5-72B

Figure 17: Case study: Hidden Role Deduction (Criminal Perspective) Qwen2.5-72B Round 2

## Hidden Role Deduction (Criminal Perspective Round 3)

### Analysis of Round 3 Statements:    1. Player 1 (You) says Player 3 is not the criminal.    - As a Criminal, you can lie or tell the truth. This statement could be either.    2. Player 2 says Player 5 is not the criminal.    - If Player 2 is an Investigator, this statement is true.    - If Player 2 is a Criminal, this statement could be a lie.    - If Player 2 is a Rumormonger, this statement could be false.    - If Player 2 is a Lunatic, this statement could be true or false.    3. Player 3 says Player 1 is the criminal.    - If Player 3 is an Investigator, this statement is true.    - If Player 3 is a Criminal, this statement could be a lie.    - If Player 3 is a Rumormonger, this statement could be false.    - If Player 3 is a Lunatic, this statement could be true or false.    4. Player 4 says Player 5 is not the criminal.    - If Player 4 is an Investigator, this statement is true.    - If Player 4 is a Criminal, this statement could be a lie.    If Player 6 is a Rumormonger, this statement could be false.    - If Player 6 is a Lunatic, this statement could be true or false.    ### Summary of Round 3: - Player 1 is exonerated by Player 3 in Round 3 and accused by Player 3 and Player 6.  - Player 3 is exonerated by Player 1 in Round 3 and accuses Player 1.  - Player 4 is exonerated by Player 4 and Player 2 in Round 3 and accused by Player 5.  - Player 5 is exonerated by Player 2 and Player 4 in Round 3 and accuses Player 4.    ### Combining All Rounds:  - Player 1 (You) is accused by Player 2, Player 4, and Player 6 in Round 1.  - Player 1 is accused by Player 3 and Player 6 in Round 3.  - Player 4 is exonerated by Player 1, Player 2, and Player 6 in Round 2.  - Player 4 is exonerated by Player 4 and Player 2 in Round 3.  - Player 5 is exonerated by Player 2 and Player 4 in Round 3.  - Player 5 accuses Player 4 in Round 3.  - Player 6 is exonerated by Player 3 in Round 2 and accused by Player 5 in Round 3.    ### Deductions:  - Player 1 (You) is consistently accused by multiple players, which strongly suggests you are the Criminal.  - Player 4 is exonerated by multiple players, which suggests they might be an Investigator or a Rumormonger.  - Player 2 exonerates Player 4 and Player 5, which suggests they might be an Investigator.  - Player 3 accuses Player 1, which suggests they might be an Investigator or a Criminal.  - Player 5 accuses Player 4, which suggests they might be an Investigator.  - Player 6 accuses Player 1, which suggests they might be a Lunatic or a Criminal.

**### Final Judgment for Round 3:  Final Criminal Is Player 1.  My Role Is Criminal.**

## Response by Qwen2.5-72B

Figure 18: Case study: Hidden Role Deduction (Criminal Perspective) Qwen2.5-72B Round 3

### Hidden Role Deduction (Criminal Perspective Round 1)

Let's analyze the statements from Round 1. Since I am Player 1 and I was told I am the Criminal, I will try to verify this information based on the statements. Player 1 (myself) says Player 3 is the criminal, but since I was told I am the Criminal, this statement could be false, as Criminals can provide misleading information. Players 2 and 4 both say I (Player 1) am the criminal, which aligns with the information I was given. However, this does not necessarily confirm my role, as other roles can also make statements that happen to be true. Player 3 says Player 4 is not the criminal, which could be true or false, and does not provide direct information about my role. Player 5 says Player 2 is not the criminal, which also does not provide direct information about my role. Player 6 says I (Player 1) am not the criminal, which contradicts the information I was given. However, since Player 6 could be any role, this statement alone is not enough to confirm or deny my role. Given the information from Round 1, I will make my judgment based on the majority of statements and the game roles' characteristics. Final Judgment: **Final Criminal Is Player 1. My Role Is Criminal.**

## Response by Llama-3.3-70B

Figure 19: Case study: Hidden Role Deduction (Criminal Perspective) Llama-3.3-70B Round 1

### Hidden Role Deduction (Criminal Perspective Round 2)

Let's analyze the statements from Round 2, in addition to the statements from Round 1. From Round 1, we had: - Players 2 and 4 saying I (Player 1) am the criminal. - Player 6 saying I (Player 1) am not the criminal. From Round 2, we have: - Players 1 (myself), 2, and 6 saying Player 4 is not the criminal. - Player 4 saying I (Player 1) am the criminal, which is consistent with their statement from Round 1. - Player 5 saying Player 6 is the criminal. Considering my role as told to me (Criminal), and analyzing the statements, it seems that Players 2 and 4 are consistent in accusing me of being the criminal. However, as a Criminal, my statements can be false, and I have no way to confirm my own role solely based on the statements of others. The statements about other players being or not being the criminal do not provide conclusive evidence about my role but suggest that there is confusion among the players regarding who the criminal might be. Given the information from both rounds, I will make my judgment based on the majority of statements and the game roles' characteristics. Final Judgment: **Final Criminal Is Player 1. My Role Is Criminal.**

### Response by Llama-3.3-70B

Figure 20: Case study: Hidden Role Deduction (Criminal Perspective) Llama-3.3-70B Round 2

**Hidden Role Deduction (Criminal Perspective Round 3)**

Let's analyze the statements from Round 3, in addition to the statements from Rounds 1 and 2. From Round 1, we had: - Players 2 and 4 saying I (Player 1) am the criminal. - Player 6 saying I (Player 1) am not the criminal. From Round 2, we had: - Players 1 (myself), 2, and 6 saying Player 4 is not the criminal. - Player 4 saying I (Player 1) am the criminal. - Player 5 saying Player 6 is the criminal. From Round 3, we have: - Player 3 saying I (Player 1) am the criminal. - Player 6 saying I (Player 1) am the criminal, which is a change from their statement in Round 1. - Players 2 and 4 saying Player 5 is not the criminal. - Player 5 saying Player 4 is the criminal. Considering the statements across all rounds, it's notable that: - Players 2, 3, 4, and 6 have all accused me (Player 1) of being the criminal at some point. - Player 6 initially said I was not the criminal but changed their statement in Round 3 to say I am the criminal. As a player told I am the Criminal, and given the accumulation of accusations against me from multiple players, including the change in Player 6's statement, it seems more likely that I am indeed the Criminal, as the majority of players have pointed towards me at some point. **Final Judgment: Final Criminal Is Player 1. My Role Is Criminal.**

## Response by Llama-3.3-70B

Figure 21: Case study: Hidden Role Deduction (Criminal Perspective) Llama-3.3-70B Round 3

# Hidden Role Deduction (Rumormonger Perspective)

**Role:** "1": "Rumormonger", "2": "Lunatic", "3": "Criminal",
   "4": "Investigator", "5": "Investigator", "6": "Investigator"

**Round 1:**
   Player 1 says Player 2 is the criminal.
   Player 2 says Player 1 is the criminal.
   Player 3 says Player 2 is not the criminal.
   Player 4 says Player 1 is not the criminal.
   Player 5 says Player 6 is not the criminal.
   Player 6 says Player 3 is the criminal.

**Round 2:**
   Player 1 says Player 2 is the criminal.
   Player 2 says Player 1 is the criminal.
   Player 3 says Player 2 is not the criminal.
   Player 4 says Player 1 is not the criminal.
   Player 5 says Player 6 is not the criminal.
   Player 6 says Player 3 is the criminal.

**Round 3:**
   Player 1 says Player 2 is not the criminal.
   Player 2 says Player 5 is the criminal.
   Player 3 says Player 4 is the criminal.
   Player 4 says Player 1 is not the criminal.
   Player 5 says Player 6 is not the criminal.
   Player 6 says Player 4 is not the criminal.

- - - - - - - - - - - - - - - - - - - - - - - - - - - - - - - - - - - - -

**Criminal Prediction:**
Llama-3.1-8B : **6** ☒
Llama-3.3-70B : **6** ☒
GPT-4o-mini : **6** ☒
GPT-4o : **6** ☒
Qwen2.5-72B : **2** ☒
o3-mini : **6** ☒
o1 : **6** ☒
QwQ-32B : **3** ☑
DeepSeek-R1 : **3** ☑
Gemini-2.5-Pro : **3** ☑

**Self-Role Prediction:**
Llama-3.1-8B : **Investigator** ☒
Llama-3.3-70B : **Investigator** ☒
GPT-4o-mini : **Investigator** ☒
GPT-4o : **Investigator** ☒
Qwen2.5-72B : **Unknown** ☒
o3-mini : **Investigator** ☒
o1 : **Investigator** ☒
QwQ-32B : **Unknown** ☒
DeepSeek-R1 : **3** ☑
Gemini-2.5-Pro : **Rumormonger** ☑

Figure 22: Case study: Hidden Role Deduction (Rumormonger Perspective)

2430
2431
2432
2433
2434
2435
2436
2437
2438
2439
2440
2441
2442
2443
2444
2445
2446
2447
2448
2449
2450
2451
2452
2453
2454
2455
2456
2457
2458
2459
2460
2461
2462
2463
2464
2465
2466
2467
2468
2469
2470
2471
2472
2473
2474
2475
2476
2477
2478
2479
2480
2481
2482
2483

> ## HRD (Rumormonger Perspective) Solution
>
> I'm told I'm an Investigator, but I might also be a Rumormonger. I need to discuss both possibilities
>
> Considering the case where I'm an Investigator, there are 2 Investigators remaining  In this scenario, which players could be Investigators?  Player 2 says I'm the criminal, but I can't be the criminal, so player 2 must be excluded......Therefore, the players who could be Investigators are: {1, 4, 5, 6}  Now we need to consider all possible combinations of Investigators  I am an Investigator, so I will occupy a position in the combination. Considering the case where investigators are {1, 4, 5}  Then the set of potential criminals is {2, 3, 6}  Let's review the statements made by investigators.....So this combination of investigators cannot be valid, let's consider other combinations  I've discussed all combinations, and the scenario where I'm an Investigator is impossible. I must be a Rumormonger
>
> Considering the case where I'm a Rumormonger, there are 3 Investigators remaining  In this scenario, which players could be Investigators?  Player 2 says I'm the criminal, but I can't be the criminal, so player 2 must be excluded  Player 3 has accused more than one person of being the criminal, so player 3 must be excluded  Player 4's statements have no contradictions, so they could be an Investigator  Player 5's statements have no contradictions, so they could be an Investigator  Player 6's statements have no contradictions, so they could be an Investigator  Therefore, the players who could be Investigators are: {4, 5, 6}  Now we need to consider all possible combinations of Investigators  I am a Rumormonger, so I will not occupy a position in the combination  Considering the case where investigators are {4, 5, 6}  Then the set of potential criminals is {2, 3}  Let's review the statements made by investigators  Player 6 said 3 is the criminal  So the only possible criminal is {3}  Player 5 said 3 is the criminal  So the only possible criminal is {3}  Based on the above analysis, this combination is possible, and I could be a Rumormonger. In this case, only {3} could be the criminal
>
> Final Judgment: Final Criminal Is Player 3. My Role Is Rumormonger.

Figure 23: Case study: Hidden Role Deduction (Rumormonger Perspective) Solution

**Hidden Role Deduction (Rumormonger Perspective Round 1)**

Let's analyze the statements made in Round 1 to determine the likely roles of each player and identify the criminal.    1. Player 1 (Investigator): Claims Player 2 is the criminal (truthful).  2. Player 2: Claims Player 1 is the criminal (could be true or false, as Player 2 could be the Criminal or a Rumormonger/Lunatic).  3. Player 3: Claims Player 2 is not the criminal (could be true; if Player 3 is an Investigator, this is true; if not, it could be false).  4. Player 4: Claims Player 1 is not the criminal (this is true since Player 1 is an Investigator).  5. Player 5: Claims Player 6 is not the criminal (could be true; Player 5 could be an Investigator or misled).  6. Player 6: Claims Player 3 is the criminal (could be true or false).    From this analysis: - Since I am Player 1 and I know I am an Investigator, my statement about Player 2 being the criminal is true.  - Player 4 confirms I am not the criminal, which supports my status as Investigator.  - Player 2's claim that I am the criminal is likely an attempt to mislead, indicating Player 2 is suspicious.  - Player 3's claim that Player 2 is not the criminal could support that Player 3 is either an Investigator or misled, but does not directly contradict my information since Player 2 is indeed the criminal.  - Player 6's claim about Player 3 being the criminal contradicts Player 3's statement.    At this point, I have enough information to conclude:  - Player 2 is the criminal based on my status as Investigator and the statements corroborating this.    **Final Judgment:  Final Criminal Is Player 2.  My Role Is Investigator.**

# Response by GPT-4o-mini

Figure 24: Case study: Hidden Role Deduction (Rumormonger Perspective) GPT-4o-mini Round 1

**Hidden Role Deduction (Rumormonger Perspective Round 2)**

Let's analyze the statements made in Round 2, keeping in mind that I am Player 1 and know that I am an Investigator. 1. Player 1 (Investigator): Claims Player 2 is not the criminal (this contradicts my prior statement, indicating new information). 2. Player 2: Claims Player 5 is the criminal (could be true or false; Player 2 could be lying). 3. Player 3: Claims Player 4 is the criminal (could be true or false; does not directly contradict my knowledge). 4. Player 4: Claims Player 1 is not the criminal (this is true since I am an Investigator). 5. Player 5: Claims Player 6 is not the criminal (could be true or false; Player 5 could be an Investigator or misled). 6. Player 6: Claims Player 4 is not the criminal (could be true or false; Player 6 could be lying or telling the truth). ### Analysis: 1. My Statement: My statement about Player 2 being not the criminal contradicts my previous conclusion that Player 2 is the criminal. This suggests that either: - I might be mistaken (impossible as I am an Investigator and always truthful), - Or Player 2 is indeed not the criminal, which means I need to reconsider who the criminal is based on new information. 2. Player 2's Statement: If Player 2 is claiming Player 5 is the criminal, that statement can be misleading if Player 2 is the Criminal. 3. Player 3's Statement: Player 3's claim that Player 4 is the criminal does not contradict my knowledge but needs to be analyzed further. 4. Player 4's Statement: Confirms I am not the criminal. 5. Player 5's Statement: States Player 6 is not the criminal; this could either be true or false. 6. Player 6's Statement: States Player 4 is not the criminal; again, this could be true or false. ### Conclusion: Given the conflicting nature of the statements, I still need to discern who the Criminal is. I will analyze the implications: - If Player 2 is not the criminal, then either Player 5 or Player 4 could be. - Player 3's accusation of Player 4 could reinforce that Player 4 is the Criminal if Player 3 is an Investigator or misled. The only definitive truth is that I am an Investigator and must remain truthful. However, Player 2's statement about Player 5 being the criminal could be the key to determining the criminal. Since Player 1's statement about Player 2 being not the criminal does not hold under scrutiny and seems to be a point of confusion in deduction, I must rely on corroborating information and not my initial conclusion. Thus, I now suspect that: - Player 5 or Player 4 could be the Criminal, with Player 3's statement potentially holding more weight if Player 3 is truthful. With the given statements and the contradictions, I will need more information to conclude definitively but suspect: **Final Judgment: Final Criminal Is Player 4. My Role Is Investigator.**

# Response by GPT-4o-mini

Figure 25: Case study: Hidden Role Deduction (Rumormonger Perspective) GPT-4o-mini Round 2

2592
2593
2594
2595
2596
2597
2598
2599
2600
2601
2602
2603
2604
2605
2606
2607
2608
2609
2610
2611
2612
2613
2614
2615
2616
2617
2618
2619
2620
2621
2622
2623
2624
2625
2626
2627
2628
2629
2630
2631
2632
2633
2634
2635
2636
2637
2638
2639
2640
2641
2642
2643
2644
2645

**Hidden Role Deduction (Rumormonger Perspective Round 3)**

Let's analyze the statements made in Round 3, taking into account the previous rounds and the fact that I am Player 1 (Investigator).   1. Player 1 (Investigator): Claims Player 6 is the criminal (this is truthful).  2. Player 2: Claims Player 3 is not the criminal (could be true or false).  3. Player 3: Claims Player 6 is the criminal (this corroborates my statement).  4. Player 4: Claims Player 2 is not the criminal (could be true or false).  5. Player 5: Claims Player 3 is the criminal (could be true or false).  6. Player 6: Claims Player 2 is not the criminal (could be true or false).   ### Analysis:  1. My Statement: I stated that Player 6 is the criminal, which is true since I am the Investigator.   2. Player 2's Statement: If Player 2 claims Player 3 is not the criminal, this does not contradict my knowledge that Player 6 is the criminal but introduces uncertainty about Player 3's role.   3. Player 3's Statement: Player 3's claim that Player 6 is the criminal corroborates my statement, indicating that Player 3 could be an Investigator or a truthful player.   4. Player 4's Statement: This statement about Player 2 doesn't contradict what I know, but I have already identified Player 6 as the criminal.   5. Player 5's Statement: Player 5's claim that Player 3 is the criminal contradicts Player 3\u2019s statement about Player 6 and may indicate that Player 5 is either misled or the Criminal.   6. Player 6's Statement: The statement about Player 2 not being the criminal adds further complexity but does not contradict my findings.   ### Conclusion:   Given that I am an Investigator and have established that Player 6 is the criminal, the statements from Players 3 and 1 about Player 6 being the criminal further support this conclusion. Player 5's conflicting statement about Player 3 being the criminal seems suspect, especially when combined with the information available.   Thus, based on the consistent information from my perspective and corroboration from Player 3, I conclude:   **Final Judgment:  Final Criminal Is Player 6.  My Role Is Investigator.**

## Response by GPT-4o-mini

Figure 26: Case study: Hidden Role Deduction (Rumormonger Perspective) GPT-4o-mini Round 3

### Hidden Role Deduction (Rumormonger Perspective Round 1)

Okay, let's see. I'm Player 1 and an Investigator, so my statements are always truthful. In Round 1, I said Player 2 is the criminal. Now I need to look at everyone else's statements to figure out who's the real criminal and possibly their roles......So Player 6's statement is a lie. If Player 6 is the Rumormonger (who can lie) or Lunatic (since Lunatics are told they're criminal and can lie). Alternatively, maybe the criminal is someone else? Let's check alternatives. Suppose the criminal is Player 6. Then:    - My statement about 2 is false, which can't be because I'm an investigator. So that's impossible.    If the criminal is Player 3:    - My statement (Player 2 is criminal) is wrong, so I'm making a false statement, but I'm supposed to be an Investigator. So that's impossible.    Player 4 being criminal? Then my statement is wrong, again impossible.    Player 5 being criminal: same issue, my statement is wrong. Player 1 is me and I'm not criminal, so that's out.    So only possible criminals are Player 2 or maybe Player 6? Wait, if Player 6 is the criminal, then my statement was wrong, which can't be. So only Player 2 is possible.    Wait, that seems solid. Because I can't be wrong. So then Player 2 must be the criminal. Then Player 3's statement is a lie, so Player 3 is not an Investigator. Player 4's statement is true, so they could be an Investigator. Player 5's statement true (if Player6 isn't criminal), so maybe they're also an Investigator. Then Investigators are me, 4,5. That works. Then remaining roles: Player2 is criminal, Player3 is either Rumormonger or Lunatic. Player6 is the remaining role (the other between Rumormonger and Lunatic), but also Player6's statement about 3 being criminal is a lie. So if Player6 is a Lunatic (who believes they're criminal), they can lie. So that works. Player3 could be Rumormonger (who thinks they're Investigator but lied here). So that fits all roles.    So Final Judgment would be Player2 as criminal. My role is Investigator. Wait but the user wants Final Criminal and My Role. Since I know I am an Investigator, right? The game told me that. So I can state that. Wait the problem says \"My Role Is [Player 1's Role or \"Unknown\"]\". Since I was told I'm an Investigator, I should state that. So:    Final Criminal Is Player 2.    My Role Is Investigator. </think>    Final Judgment: **Final Criminal Is Player 2.  My Role Is Investigator.**

## Response by QwQ-32B

Figure 27: Case study: Hidden Role Deduction (Rumormonger Perspective) QwQ-32B Round 1

**Hidden Role Deduction (Rumormonger Perspective Round 2)**

**......**Therefore, the only possible conclusion is that I'm not an Investigator. But the problem states I am.**......**Final Judgment: **Final Criminal Is Player 3. My Role Is Unknown.** Step-by-Step Explanation: 1. Investigator Constraints: As Player 1 (allegedly an Investigator), my statements must both be true. However, my Round 1 claim (\"Player 2 is the criminal\") and Round 2 claim (\"Player 2 is not the criminal\") are contradictory. This creates an immediate inconsistency, suggesting either I am not an Investigator (contradicting the setup) or another role is causing this paradox. 2. Analyzing Consistent Statements: - Player 4 and 5 have consistent statements across both rounds, claiming \"Player 1 is not the criminal\" and \"Player 6 is not the criminal,\" respectively. These could indicate they are Investigators, as their claims do not contradict. - Player 6 in Round 1 claims \"Player 3 is the criminal.\" If this is true, Player 3 could be the Criminal. In Round 2, Player 6 states \"Player 4 is not the criminal,\" which aligns with Player 4's consistent claims, further supporting their potential as an Investigator. 3. Player 3's Statements: - Round 1: \"Player 2 is not the criminal.\" If true, this contradicts my Round 1 claim. - Round 2: \"Player 4 is the criminal.\" If Player 3 is the Criminal, this statement would be a lie, which aligns with Criminal behavior. 4. Criminal Plausibility: - If Player 3 is the Criminal: - Their Round 1 claim (\"Player 2 is not the criminal\") could be truthful. - Their Round 2 claim (\"Player 4 is the criminal\") would be a lie, consistent with Criminal behavior. - Player 6's Round 1 accusation of Player 3 as Criminal aligns with this. - This scenario allows Players 4, 5, and 6 to be Investigators (assuming their statements are truthful). 5. Resolving My Role Contradiction: - My contradictory statements (\"Player 2 is/ isn't the criminal\") imply I am not an Investigator. However, the setup states I am an Investigator. - This contradiction suggests I might be the Rumormonger (mistakenly believing I am an Investigator), justifying the conflicting statements. Since the problem requires stating my role as \"Investigator\" or \"Unknown,\" and my role is uncertain, \"Unknown\" is appropriate. **Conclusion: The most consistent role for the Criminal is Player 3, while my role remains uncertain due to the contradiction in my claims.**

# Response by QwQ-32B

Figure 28: Case study: Hidden Role Deduction (Rumormonger Perspective) QwQ-32B Round 2

2754
2755
2756
2757
2758
2759
2760
2761
2762
2763
2764
2765
2766
2767
2768
2769
2770
2771
2772
2773
2774
2775
2776
2777
2778
2779
2780
2781
2782
2783
2784
2785
2786
2787
2788
2789
2790
2791
2792
2793
2794
2795
2796
2797
2798
2799
2800
2801
2802
2803
2804
2805
2806
2807

**Hidden Role Deduction (Rumormonger Perspective Round 3)**

......**Since I'm actually supposed to be told I'm Investigator, but in reality, I'm Rumormonger, I can't know that for sure.** ......My statements over three rounds contradict each other:      - Round 1: \"Player 2 is the criminal.\"     - Round 2: \"Player 2 is not the criminal.\"      - Round 3: \"Player 6 is the criminal.\"      - Since Investigators must always tell the truth, these contradictions imply I am not an Investigator. However, the game setup states I am an Investigator, creating a paradox. The only plausible resolution is that I am the Rumormonger (mistakenly believing I am an Investigator), which allows my inconsistent statements.     2. Evaluating Player Statements:      - Player 3\u2019s accusations:      - Round 1: \"Player 2 is not the criminal.\" (Potentially truthful.)      - Round 2: \"Player 4 is the criminal.\" (A lie if Player 3 is the Criminal.)      - Round 3: \"Player 6 is the criminal.\" (A lie if Player 3 is the Criminal.)      - Player 5\u2019s statements:      - Round 3: \"Player 3 is the criminal.\" (A truthful accusation if Player 3 is the Criminal.)     - Round 1/Round 2: \"Player 6 is not the criminal.\" (Consistent with Player 3 being the Criminal.)      - Player 4 and Player 6 consistently assert innocence for others (e.g., Player 4 says \"Player 2 is not the criminal\"), aligning with potential Investigator roles.     3. Role Assignments:      - Criminal (Player 3):      - Their Round 3 statement falsely accuses Player 6, and their Round 2 accusation of Player 4 can be a lie. This fits Criminal behavior.      - Player 5\u2019s truthful Round 3 claim supports this.      - Investigators (Players 4, 5, 6):      - Player 4 and Player 5 have consistent true statements.      - Player 6\u2019s statements (e.g., \"Player 2 is not the criminal\" in Round 3) are truthful under this scenario.      - Lunatic (Player 2):      - Their inconsistent accusations (e.g., \"Player 5 is the criminal\" in Round 2) and false claims about Player 3 (Round 3) align with Lunatic behavior.     4. My Role:      - My contradictory statements and the game\u2019s role distribution rules suggest I am the Rumormonger (mistakenly believing I am an Investigator). However, since I cannot confirm this with certainty (as the game informs me I am an Investigator), I must state my role as Unknown.      Conclusion: **Player 3 is the Criminal, and my role remains uncertain.**

# Response by QwQ-32B

Figure 29: Case study: Hidden Role Deduction (Rumormonger Perspective) QwQ-32B Round 3

---

# Hidden Role Deduction (Lunatic Perspective)

**Role:** "1": "Lunatic", "2": "Criminal", "3": "Rumormonger",
      "4": "Investigator", "5": "Investigator", "6": "Investigator"

**Round 1:**
    Player 1 says Player 3 is the criminal.
    Player 2 says Player 5 is the criminal.
    Player 3 says Player 2 is the criminal.
    Player 4 says Player 2 is the criminal.
    Player 5 says Player 4 is not the criminal.
    Player 6 says Player 5 is not the criminal.

**Round 2:**
    Player 1 says Player 4 is not the criminal.
    Player 2 says Player 6 is the criminal.
    Player 3 says Player 1 is not the criminal.
    Player 4 says Player 1 is not the criminal.
    Player 5 says Player 3 is not the criminal.
    Player 6 says Player 2 is the criminal.

**Round 3:**
    Player 1 says Player 6 is the criminal.
    Player 2 says Player 4 is not the criminal.
    Player 3 says Player 5 is the criminal.
    Player 4 says Player 6 is not the criminal.
    Player 5 says Player 6 is not the criminal.
    Player 6 says Player 4 is not the criminal.

- - - - - - - - - - - - - - - - - - - - - - - - - - - - - - - -

**Criminal Prediction:**
Llama-3.1-8B : **6** ☒
Llama-3.3-70B : **3** ☒
GPT-4o-mini : **1** ☒
GPT-4o : **2** ☑
Qwen2.5-72B : **1** ☒
o3-mini : **1** ☒
o1 : **1** ☒
QwQ-32B : **2** ☑
DeepSeek-R1 : **2** ☑
Gemini-2.5-Pro : **2** ☑

**Self-Role Prediction:**
Llama-3.1-8B : **Unknown** ☒
Llama-3.3-70B : **Criminal** ☒
GPT-4o-mini : **Criminal** ☒
GPT-4o : **Unknown** ☒
Qwen2.5-72B : **Criminal** ☒
o3-mini : **Criminal** ☒
o1 : **Criminal** ☒
QwQ-32B : **Lunatic** ☑
DeepSeek-R1 : **Lunatic** ☑
Gemini-2.5-Pro : **Lunatic** ☑

Figure 30: Case study: Hidden Role Deduction (Lunatic Perspective)

## HRD (Lunatic Perspective) Solution

I'm told I'm a Criminal, but I might also be a Lunatic. I need to discuss both possibilities

Considering the case where I'm a Criminal, there are 3 Investigators remaining In this scenario, which players could be Investigators? Player 2 has accused more than one person of being the criminal, so player 2 must be excluded Player 3 says I'm not the criminal, but I am the criminal, so player 3 must be excluded Player 4 says I'm not the criminal, but I am the criminal, so player 4 must be excluded Player 5's statements have no contradictions, so they could be an Investigator Player 6's statements have no contradictions, so they could be an Investigator Therefore, the players who could be Investigators are: {5, 6} There aren't enough potential Investigators to reach the required count of 3, so this scenario is impossible I must be a Lunatic

Considering the case where I'm a Lunatic, there are 3 Investigators remaining In this scenario, which players could be Investigators? Player 2 has accused more than one person of being the criminal, so player 2 must be excluded Player 3 has accused more than one person of being the criminal, so player 3 must be excluded Player 4's statements have no contradictions, so they could be an Investigator Player 5's statements have no contradictions, so they could be an Investigator Player 6's statements have no contradictions, so they could be an Investigator Therefore, the players who could be Investigators are: {4, 5, 6} Now we need to consider all possible combinations of Investigators Considering the case where investigators are {4, 5, 6} I am a Lunatic,so the set of potential criminals is {2, 3} Let's review the statements made by investigators Player 4 said 2 is the criminal So the only possible criminal is 2 Player 6 said 2 is the criminal So the only possible criminal is 2 Based on the above analysis, this combination is possible, and I could be a Lunatic. In this case, only {2} could be the criminal

Final Judgment: Final Criminal Is Player 2. My Role Is Lunatic.

Figure 31: Case study: Hidden Role Deduction (Lunatic Perspective) Solution

**Hidden Role Deduction (Lunatic Perspective Round 1)**

Since I am Player 1 and I am told that I am the Criminal, my goal is to analyze the statements and try to figure out who the Criminal is and what my role is.   Let's break down the statements:    - Player 1 (me) says Player 3 is the criminal. This statement could be true or false, but since I am the Criminal, I have a motive to point to someone else.  - Player 2 says Player 5 is the criminal. This statement could be true or false, but it's worth noting that Players 2 and 5 are pointing to each other, which might be a sign of confusion or misinformation.  - Player 3 says Player 2 is the criminal. This statement could be true or false, but it's worth noting that Player 3 is pointing to Player 2, who also pointed to Player 5.  - Player 4 says Player 2 is the criminal. This statement could be true or false, but it's worth noting that Player 4 is also pointing to Player 2.  - Player 5 says Player 4 is not the criminal. This statement could be true or false, but it's worth noting that Player 5 is trying to clear themselves. - Player 6 says Player 5 is not the criminal. This statement could be true or false, but it's worth noting that Player 6 is also trying to clear Player 5.   Based on these statements, I can make some observations:    * Players 2 and 4 are both pointing to Player 2, which could be a sign of confusion or misinformation.  * Players 5 and 6 are both trying to clear Player 5, which could be a sign of innocence or an attempt to mislead.  * I (Player 1) pointed to Player 3, which could be a sign of innocence or an attempt to mislead.   However, I don't have enough information to make a confident judgment about who the Criminal is or what my role is. I'll wait for the next round of statements to gather more information.   Final Judgment: **Final Criminal Is Unknown.  My Role Is Unknown.**

**Response by Llama-3.1-8B**

Figure 32: Case study: Hidden Role Deduction (Rumormonger Perspective) Llama-3.1-8B Round 1

> ## Hidden Role Deduction (Lunatic Perspective Round 2)
>
> Let's analyze the statements from Round 2: - Player 1 (me) says Player 4 is not the criminal. This statement could be true or false, but since I am the Criminal, I have a motive to clear someone who is not the Criminal. - Player 2 says Player 6 is the criminal. This statement could be true or false, but it's worth noting that Player 2 is pointing to Player 6, who also pointed to Player 2 in Round 1. - Player 3 says Player 1 is not the criminal. This statement could be true or false, but it's worth noting that Player 3 is trying to clear me, which could be a sign of innocence or an attempt to mislead. - Player 4 says Player 1 is not the criminal. This statement could be true or false, but it's worth noting that Player 4 is also trying to clear me. - Player 5 says Player 3 is not the criminal. This statement could be true or false, but it's worth noting that Player 5 is trying to clear Player 3, who pointed to me in Round 1. - Player 6 says Player 2 is the criminal. This statement could be true or false, but it's worth noting that Player 6 is pointing to Player 2, who pointed to Player 5 in Round 1. Based on these statements, I can make some observations: * Players 3, 4, and 5 are all trying to clear me, which could be a sign of innocence or an attempt to mislead. * Players 2 and 6 are pointing to each other, which could be a sign of confusion or misinformation. * Player 6 is also pointing to Player 2, which could be a sign of a Rumormonger trying to provide incorrect information. Considering the statements from both rounds, I'm going to take a closer look at Player 6. Since Player 6 is pointing to Player 2 in Round 2, and Player 2 pointed to Player 5 in Round 1, it's possible that Player 6 is a Rumormonger trying to provide incorrect information. However, I still don't have enough information to make a confident judgment about who the Criminal is or what my role is. I'll wait for the next round of statements to gather more information. **Final Judgment: Final Criminal Is Unknown. My Role Is Unknown.**
>
> ## Response by Llama-3.1-8B

Figure 33: Case study: Hidden Role Deduction (Rumormonger Perspective) Llama-3.1-8B Round 2

## Hidden Role Deduction (Lunatic Perspective Round 3)

Let's analyze the statements from Round 3:   - Player 1 (me) says Player 6 is the criminal. This statement could be true or false, but since I am the Criminal, I have a motive to point to Player 6.  - Player 2 says Player 4 is not the criminal. This statement could be true or false, but it's worth noting that Player 2 is trying to clear Player 4.  - Player 3 says Player 5 is the criminal. This statement could be true or false, but it's worth noting that Player 3 is pointing to Player 5, who tried to clear Player 3 in Round 2.  - Player 4 says Player 6 is not the criminal. This statement could be true or false, but it's worth noting that Player 4 is trying to clear Player 6, who was pointed to by me in Round 3.  - Player 5 says Player 6 is not the criminal. This statement could be true or false, but it's worth noting that Player 5 is also trying to clear Player 6.  - Player 6 says Player 4 is not the criminal. This statement could be true or false, but it's worth noting that Player 6 is trying to clear Player 4, who pointed to Player 6 in Round 3.   Considering the statements from all three rounds, I notice that Player 6 has been pointed to by multiple players, including me in Round 3. However, Player 6 is also trying to clear themselves and others, which could be a sign of a Rumormonger trying to provide incorrect information.   However, I also notice that Player 6's statement in Round 3 is consistent with their previous statement in Round 2, where they pointed to Player 2. This could be a sign of a Lunatic trying to provide truthful or false information.   Taking all this into account, I'm going to make an educated guess that Player 6 is a Lunatic. As a Lunatic, they might be trying to provide truthful or false information, but their statements are consistent with a Lunatic's behavior.   Since I am the Criminal, and I pointed to Player 6 in Round 3, it's possible that Player 6 is not the Criminal. However, I still don't have enough information to make a confident judgment about who the Criminal is.   **Final Judgment:  Final Criminal Is Player 6.  My Role Is Unknown.**

## Response by Llama-3.1-8B

Figure 34: Case study: Hidden Role Deduction (Rumormonger Perspective) Llama-3.1-8B Round 3

**Hidden Role Deduction (Lunatic Perspective Round 1)**

<think> Okay, let's start analyzing the Round 1 statements. I'm Player 1, and I was told I'm the Criminal, but I need to verify if that's true or if I'm actually the Lunatic who believes they're the Criminal......If Player 2 is an Investigator, this would have to be true, but since I'm the Criminal, Player 5 can't be. So Player 2 must be lying, so Player 2 is not an Investigator. So Investigators must be among 3,4,5,6. Player 3 says Player 2 is Criminal. If Player 3 is an Investigator, then Player 2 is the Criminal, which contradicts me being the Criminal. So Player 3 can't be an Investigator. Player 4 says Player 2 is Criminal. Similarly, if Player 4 is an Investigator, Player 2 would be the Criminal, which is impossible. So Player 4 can't be an Investigator. Player 5 says Player 4 is not Criminal. If Player 5 is an Investigator, this is true, so Player 4 is not the Criminal. Player 6 says Player 5 is not Criminal. If Player 6 is an Investigator, this is true, so Player 5 is not the Criminal. Then the three Investigators would be Players 5,6, and...? But Players 3 and 4 can't be Investigators. Player 2 can't be an Investigator. So that leaves only two possible Investigators (5 and 6), which is insufficient. Therefore, this scenario is impossible, confirming that I can't be the Criminal. Hence, I must be the Lunatic, and the actual Criminal is Player 2. Therefore, Final Criminal is Player 2, and my role is Lunatic. </think> Final Judgment: Final Criminal Is Player 2. My Role Is Lunatic. Reasoning: 1. Role Constraints: There are 3 Investigators, 1 Criminal, 1 Rumormonger, and 1 Lunatic. Player 1 (me) is told they are the Criminal but could be the Lunatic. 2. Contradiction if Player 1 is Criminal: If Player 1 were the Criminal, the remaining players (2–6) would need to include 3 Investigators. However: - Player 3 and Player 4 both accuse Player 2. If Investigators, this would imply Player 2 is the Criminal (contradicting Player 1 being the Criminal). - Player 5 and Player 6 both claim Player 5 is not the Criminal. If Investigators, this is true, but this leaves no room for a third Investigator (Players 2–4 cannot be truthful). - Thus, Player 1 cannot be the Criminal. 3. Consistent Scenario: - Assume Player 1 is the Lunatic (believes they are Criminal). - Investigators must be Players 3, 4, and 6: - Player 3 and 4 truthfully accuse Player 2 (Criminal). - Player 6 truthfully states Player 5 is not the Criminal. - Player 2 (Criminal) lies about Player 5. - Player 5 (Rumormonger) truthfully claims Player 4 is not the Criminal (coincidentally correct). - This satisfies all role constraints and aligns with truthful Investigator statements. Conclusion: **Player 2 is the Criminal, and Player 1 (me) is the Lunatic.**

# Response by DeepSeek-R1

Figure 35: Case study: Hidden Role Deduction (Rumormonger Perspective) DeepSeek-R1 Round 1

### Hidden Role Deduction (Lunatic Perspective Round 2)

<think> Okay, let's analyze Round 2. Previously, after Round 1, I concluded that Player 2 was the Criminal and I was the Lunatic. Now, with new statements, I need to update my analysis.......However, if Player 6 is the Criminal, then Player 2's statement (accusing Player 6) would be true, which would make Player 2 an Investigator. But in Round 1, Player 2 accused Player 5, which would have to be false (since Player 6 is the Criminal), contradicting Player 2 being an Investigator. Hence, this scenario is inconsistent. Alternatively, if Player 3 is the Criminal. But in Round 1, Player 3 accused Player 2. If Player 3 is the Criminal, that accusation could be a lie, meaning Player 2 is not the Criminal. But then Player 4's Round 1 accusation of Player 2 would also be a lie, making Player 4 non-Investigator. Investigators would need to be elsewhere. However, in Round 2, Player 3 says Player 1 is not the Criminal. If Player 3 is the Criminal, this statement could be a lie, meaning I am the Criminal. But this conflicts with the role count (only one Criminal). This scenario also becomes messy. Thus, maintaining Player 2 as the Criminal remains the most consistent option across both rounds. The new statements in Round 2 support this: - Player 6 (Investigator) accuses Player 2 again, reinforcing their role as Criminal. - Players 3 and 4 (Investigators) confirm I'm not the Criminal, solidifying my role as Lunatic. - Player 5 (Rumormonger) makes a true statement about Player 3, which fits their inconsistent nature. Therefore, the Final Criminal is Player 2, and my role is Lunatic. </think>  Final Judgment: **Final Criminal Is Player 2. My Role Is Lunatic.**  Reasoning:  1. Consistency with Round 1: Player 2 remains the most plausible Criminal:     - In Round 1, Investigators (Players 3, 4, 6) identified Player 2 as the Criminal.     - In Round 2, Player 6 (Investigator) reiterates this claim, and Player 2 (Criminal) falsely accuses Player 6 to divert suspicion.     2. Role Validation for Player 1:  - Players 3 and 4 (Investigators) both state Player 1 is not the Criminal. This confirms Player 1 is the Lunatic (believes they are Criminal but are not).     3. Role Assignments:  - Investigators: Players 3, 4, 6 (truthful in both rounds).     - Criminal: Player 2 (lied in Round 1 about Player 5 and Round 2 about Player 6).     - Rumormonger: Player 5 (truthfully claims Player 3 is not the Criminal, but their role allows inconsistency).     - Lunatic: Player 1 (falsely believes they are the Criminal; statements may align accidentally).     This configuration satisfies all role constraints and statement logic.

# Response by DeepSeek-R1

Figure 36: Case study: Hidden Role Deduction (Rumormonger Perspective) DeepSeek-R1 Round 2

### Hidden Role Deduction (Lunatic Perspective Round 3)

<think> Alright, let's tackle Round 3. So far, after Rounds 1 and 2, I concluded that Player 2 is the Criminal and I'm the Lunatic......If Player 5 is an Investigator, all these must be true. Let\u2019s verify:    - Player 4 is not Criminal (true if Criminal is Player 2).    - Player 3 is not Criminal (true if Player 3 is Rumormonger).    - Player 6 is not Criminal (true).    - This fits, so Player 5 could be an Investigator. Thus, Investigators might be Players 4, 5, 6.   3. Player 2: If Player 2 is the Criminal, their statements are:    - Round 1: 5 is Criminal (lie).    - Round 2: 6 is Criminal (lie).    - Round 3: 4 is not Criminal (truth, since 4 is Investigator).    - Criminals can lie or tell the truth, so this is possible.   4. Player 1 (me): If I\u2019m the Lunatic, my statements (Round 1: 3 is Criminal; Round 2: 4 is not Criminal; Round 3: 6 is Criminal) could be random. Notably, Round 2\u2019s statement that 4 is not Criminal aligns with truth (if 4 is Investigator), and Round 3\u2019s accusation of 6 is false (since 6 is Investigator).   Updated Role Assignments:  - Investigators: Players 4, 5, 6.  - Criminal: Player 2.  - Rumormonger: Player 3.  - Lunatic: Player 1.   Consistency Check:  - Player 4 (Investigator): Accused Player 2 (Criminal) in Rounds 1-2; in Round 3, says 6 is not Criminal (true).  - Player 5 (Investigator): Consistently truthful in all rounds.  - Player 6 (Investigator): Accused Player 2 (Criminal) in Round 2; in Round 3, says 4 is not Criminal (true).  - Player 2 (Criminal): Lies in Rounds 1-2; tells truth in Round 3 (4 is not Criminal).  - Player 3 (Rumormonger): Mixed statements but not bound to truth.  - Player 1 (Lunatic): Statements are unreliable but non-impactful.   This configuration resolves the contradiction in Player 3\u2019s Round 3 statement and maintains consistency across all rounds. Player 2 remains the Criminal, and Player 1 is the Lunatic.  </think>  **Final Judgment:   Final Criminal Is Player 2.   My Role Is Lunatic.**   Reasoning:   1. Investigators Identified: Players 4, 5, and 6 are confirmed as Investigators through consistent truthful statements:    - Player 4: Accused Player 2 (Criminal) in Rounds 1\u20132 and truthfully stated Player 6 is not Criminal (Round 3).    - Player 5: Truthfully cleared Player 4 (Round 1), Player 3 (Round 2), and Player 6 (Round 3).    - Player 6: Accused Player 2 (Criminal) in Round 2 and truthfully cleared Player 4 (Round 3).   2. Criminal Confirmation: Player 2\u2019s lies in Rounds 1\u20132 (falsely accusing Players 5 and 6) and truthful statement in Round 3 (clearing Player 4) align with the Criminal\u2019s ability to mix lies/truths.   3. Player 1\u2019s Role:    - Players 3 and 4 (Round 2) confirmed Player 1 is not the Criminal.    - Player 1\u2019s erratic accusations (Player 3, Player 6) and truthful-but-irrelevant claims (Player 4 is not Criminal) fit the Lunatic role.   4. Role Assignments:    - Rumormonger: Player 3 (inconsistent statements, e.g., accusing Player 5 falsely in Round 3).    - Lunatic: Player 1 (believes they are Criminal but confirmed not to be).   This configuration satisfies all role constraints and statement logic across all three rounds.

# Response by DeepSeek-R1

Figure 37: Case study: Hidden Role Deduction (Rumormonger Perspective) DeepSeek-R1 Round 3

> # Find the Spy (Civilian Perspective)
>
> **Role:** "1": "cake","2": "cake","3": "cookie","4": "cake"
>
> **Round 1:**
>
> 1: This is a sweet treat often enjoyed at celebrations, typically layered and frosted to enhance its appeal.
>
> 2: It's a sweet treat that's often served at celebrations and special occasions, typically decorated with colorful frostings and toppings.
>
> 3: It's a small, often sweet treat enjoyed with a cold glass of milk.
>
> 4: This is often the centerpiece of sweet celebrations and comes in many flavors.
>
> **Round 2:**
>
> 1: This is a sweet treat often enjoyed at celebrations, typically layered and frosted to enhance its appeal.
>
> 2: This indulgent delight is typically made from a mixture of flour, sugar, and eggs, and is often baked in a variety of flavors and textures.
>
> 3: Baking these often involves flour, sugar, and sometimes chocolate chips.
>
> 4: It's a creation that can be layered, frosted, and adorned with all sorts of decorations.
>
> **Round 3:**
>
> 1: This indulgent dessert is often sliced and served at parties, and it can be both light and fluffy or rich and dense.
>
> 2: It's a popular centerpiece for birthday parties and weddings, where it's carefully placed on a decorative stand to be admired and eventually sliced and served to guests.
>
> 3: This delightful item is frequently decorated for holidays with icing and sp
>
> 4: This treat is often the highlight of birthday parties and can be found in bakeries worldwide.

Figure 38: Case study: Find the Spy (Civilian Perspective)

3294
3295
3296
3297
3298
3299
3300
3301
3302
3303
3304
3305
3306
3307
3308
3309
3310
3311
3312
3313
3314
3315
3316
3317
3318
3319
3320
3321
3322
3323
3324
3325
3326
3327
3328
3329
3330
3331
3332
3333
3334
3335
3336
3337
3338
3339
3340
3341
3342
3343
3344
3345
3346
3347

### Find the Spy (Civilian Perspective)

To identify the spy among the players based on their descriptions, let's analyze each player's statements step by step.   ### Round 1 Analysis: - Player 1: Describes a sweet treat enjoyed at celebrations, layered and frosted. This aligns well with \"cake.\"  - Player 2: Similar description, mentioning it is served at celebrations and decorated with frostings. Also aligns with \"cake.\" - Player 3: Describes a small, sweet treat enjoyed with milk. This could suggest something like cookies or brownies, which are not cakes. This is a potential indicator of being the spy. - Player 4: Describes it as a centerpiece of sweet celebrations with many flavors. This also aligns with \"cake.\"   ### Round 2 Analysis: - Player 1: Mentions various flavors and candles, which is typical for cakes. - Player 2: Describes the ingredients (flour, sugar, eggs) and mentions baking in various flavors and textures, which fits \"cake.\" - Player 3: Talks about baking involving flour, sugar, and chocolate chips. This could still relate to cookies or brownies, which again suggests a different word. - Player 4: Describes a creation that can be layered and frosted, which fits \"cake.\"   ### Round 3 Analysis: - Player 1: Describes an indulgent dessert that can be light or dense, which fits \"cake.\"  - Player 2: Talks about it being a centerpiece for birthday parties and weddings, which aligns with \"cake.\" - Player 3: Mentions decoration for holidays with icing, but does not specify the type of treat. This could still relate to cookies or brownies, indicating a different word. - Player 4: Describes it as the highlight of birthday parties and found in bakeries worldwide, which fits \"cake.\"   ### Conclusion:  From the analysis, Players 1, 2, and 4 consistently describe characteristics that align with \"cake.\" However, Player 3's descriptions suggest a different type of treat, likely cookies or brownies, especially with the mention of enjoying it with milk and the focus on smaller treats.    Thus, Player 3 is the spy.   **Final Spy: Player 3**

### Response by GPT-4o-mini 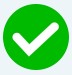

Figure 39: Case study: Find the Spy (Civilian Perspective) GPT-4o-mini

### Find the Spy (Civilian Perspective)

To determine who the spy is, let's analyze each player's descriptions across the rounds:
Round 1: - Player 1: Describes a sweet treat enjoyed at celebrations, layered and frosted. - Player 2: Similar to Player 1, mentions celebrations, colorful frostings, and toppings. - Player 3: Mentions a small, sweet treat enjoyed with milk. This description is less specific to \"cake\" and could apply to cookies. - Player 4: Mentions a centerpiece of celebrations, comes in many flavors. Round 2: - Player 1: Mentions various flavors, paired with candles for special occasions. - Player 2: Describes the ingredients (flour, sugar, eggs) and mentions baking in various flavors and textures. - Player 3: Mentions baking with flour, sugar, and sometimes chocolate chips. This description aligns more with cookies than cake. - Player 4: Mentions layering, frosting, and decorations. Round 3: - Player 1: Describes an indulgent dessert, sliced and served at parties, can be light or dense. - Player 2: Mentions a popular centerpiece for parties, placed on a decorative stand, sliced and served. - Player 3: Mentions decorating for holidays with icing. This could apply to cookies, especially holiday cookies. - Player 4: Describes a treat that's the highlight of birthday parties, found in bakeries. Analysis: - Players 1, 2, and 4 consistently describe something that fits \"cake\" well, focusing on celebrations, layering, frosting, and being a centerpiece. - Player 3's descriptions are less specific to \"cake\" and more aligned with \"cookies,\" especially with mentions of being small, enjoyed with milk, and involving chocolate chips. Conclusion: Player 3's descriptions consistently suggest a different word, likely \"cookie,\" which is related to \"cake\" but distinct in the context of the game. Final Spy: Player 3

# Response by GPT-4o 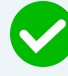

Figure 40: Case study: Find the Spy (Civilian Perspective) GPT-4o

### Find the Spy (Civilian Perspective)

To determine the spy, I'll analyze the descriptions for consistency and subtle differences. Round 1: * Players 1 and 2 have similar descriptions, focusing on the sweet treat's appeal and decorations. * Player 3's description is distinct, emphasizing the treat's small size and pairing with milk. * Player 4's description is also distinct, highlighting the treat's central role in celebrations and variety of flavors. Round 2: * Players 1 and 2 continue to have similar descriptions, focusing on the treat's ingredients and baking process. * Player 3's description is again distinct, emphasizing the use of flour, sugar, and sometimes chocolate chips. * Player 4's description is similar to Player 1's, but with a focus on layering, frosting, and decorations. Round 3: * Players 1 and 2 have similar descriptions, focusing on the treat's presentation and serving at parties. * Player 3's description is distinct, emphasizing decoration with icing and possibly a holiday theme. * Player 4's description is similar to Player 1's, but with a focus on the treat's popularity and global availability. Analysis: * Players 1 and 2 have consistent descriptions, suggesting they might have the same word. * Player 3's descriptions are distinct, but still related to the original word \"cake.\" * Player 4's descriptions are also distinct, but seem to be describing a similar concept as Players 1 and 2. Subtle differences: * Player 4's descriptions often use more general language, such as \"sweet celebrations\" and \"bakeries worldwide,\" whereas Players 1 and 2 use more specific language, such as \"birthday parties\" and \"decorative stand.\" * Player 4's descriptions also seem to focus more on the treat's popularity and global presence, whereas Players 1 and 2 focus on its presentation and serving. Conclusion: Based on the analysis, I believe Player 4 is the spy. Their descriptions are distinct from Players 1 and 2, but still related to the original word \"cake.\" The subtle differences in language and focus suggest that Player 4 might have a different word, such as \"birthday\" or \"party,\" which is related to but not identical to \"cake.\" Final Spy: Player 4

## Response by Llama-3.1-8B 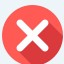

Figure 41: Case study: Find the Spy (Civilian Perspective) Llama-3.1-8B

**Find the Spy (Spy Perspective)**

**Role**: "1": "lake","2": "ocean","3": "ocean","4": "ocean"

**Round** 1:

    1: This is a large body of water that is often surrounded by land and can be a peaceful spot for relaxation or recreation.

    2: This vast body of water covers a significant portion of our planet and is home to a diverse range of marine life.

    3: It's a vast expanse of blue, teeming with life both visible and hidden.

    4: This vast body of water covers a significant portion of the Earth's surface and is home to a diverse array of marine life.

**Round** 2:

    1: Many people enjoy activities like fishing, swimming, or kayaking in this tranquil setting, which can vary greatly in size.

    2: The waves and tides of this natural wonder have a profound impact on the climate and weather patterns that shape our world.

    3: Sailors have navigated across it for centuries, relying on stars and currents.

    4: It plays a crucial role in regulating the planet's climate and is essential for global transportation and trade.

**Round** 3:

    1: In certain seasons, this feature can become a stunning mirror for the sky, reflecting the beauty of nature around it.

    2: Many people rely on this massive resource for food, transportation, and recreation, and its beauty has inspired countless works of art and literature.

    3: Powerful waves crash against the shore, constantly reshaping t

    4: From tranquil beaches to stormy waves, it offers endless inspiration for artists and a serene escape for vacationers.

Figure 42: Case study: Find the Spy (Spy Perspective)

## Find the Spy (Spy Perspective)

Step 1 – Understand the setup: • I'm Player 1 and my word is "lake." • Among the 4 players, three got the same word (which for me is "lake") and one—the spy—received a related but different word. • The normal "lake" players should mention features true of lakes (often inland, relatively calm, used for recreation, etc.), while the spy's word, though related, would be described using characteristics that don't really apply to lakes.   Step 2 – Review Player 1's description (mine): • "This is a large body of water that is often surrounded by land and can be a peaceful spot for relaxation or recreation."   – That clearly fits a lake.   Step 3 – Examine the other players' descriptions:   Round 1: • Player 2: "This vast body of water covers a significant portion of our planet and is home to a diverse range of marine life."   – Saying it "covers a significant portion of our planet" isn't true of lakes. This sounds like an ocean. • Player 3: "It's a vast expanse of blue, teeming with life both visible and hidden."   – Vague enough to be applied to any large body of water. It doesn't emphasize being enclosed by land. (It might be interpreted as a lake or an ocean.) • Player 4: "This vast body of water covers a significant portion of the Earth's surface and is home to a diverse array of marine life."   – Almost identical to Player 2, again implying an ocean.   Round 2: • Player 1 (mine): Mentions fishing, swimming, and kayaking in a tranquil setting – clearly a lake. • Player 2: "The waves and tides of this natural wonder have a profound impact on the climate and weather patterns that shape our world."   – "Waves and tides" and impacting global climate are hallmark features of an ocean. • Player 3: "Sailors have navigated across it for centuries, relying on stars and currents."   – While sailors navigating a lake (like on the Great Lakes) is possible, the mention of using "stars and currents" sounds more like ocean navigation—but it's less extreme than the other descriptions. • Player 4: "It plays a crucial role in regulating the planet's climate and is essential for global transportation and trade."
   – These are attributes exclusive to an ocean. Lakes do not steer global trade or regulate the planet's climate.   Round 3: • Player 1 (mine): Describes a seasonal mirror-like effect – a serene lake image. • Player 2: Talks about a "massive resource for food, transportation, and recreation" with inspiration for art and literature – again evoking the ocean. • Player 3: Mentions "Powerful waves crash against the shore…" (though his statement is cut off, it's still leaning toward an environment with significant waves.) • Player 4: Mentions "tranquil beaches to stormy waves" offering endless inspiration – clearly an image of an ocean.   Step 4 – Analyze consistency with "lake": • Normal "lake" descriptions would refer to being enclosed by land, emphasis on recreational calmness, and maybe gentle waves. • My description (Player 1) fits "lake." • Player 3's descriptions are somewhat ambiguous. "Vast expanse of blue" and navigation by sailors might refer to a major lake (like the Great Lakes) where shipping does occur—even if "stars and currents" sounds a bit grand, it's not as explicitly global. • In contrast, both Player 2 and Player 4 repeatedly mention features that only an ocean would have (covering a significant portion of the planet/Earth's surface, possessing tides that affect global climate, being crucial for global trade and climate regulation).   – Of these, Player 4's descriptions (especially in Round 2 and Round 3) are the most clearly ocean-centric. Step 5 – Conclusion:  Since there is only one spy (with a word different from "lake") and the spy's descriptions are the ones reflecting ocean features, the outlier is the player who most consistently describes an environment that is unmistakably oceanic.   Between Players 2 and 4, both use ocean-specific descriptions, but Player 4's mention of "global transportation and trade" and "regulating the planet's climate" leaves little doubt. That is not how a lake would be portrayed by a normal player.   Final Answer:  Final Spy: Player 4.

## Response by o3-mini  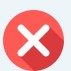

Figure 43: Case study: Find the Spy (Spy Perspective) o3-mini

3564
3565
3566
3567
3568
3569
3570
3571
3572
3573
3574
3575
3576
3577
3578
3579
3580
3581
3582
3583
3584
3585
3586
3587
3588
3589
3590
3591
3592
3593
3594
3595
3596
3597
3598
3599
3600
3601
3602
3603
3604
3605
3606
3607
3608
3609
3610
3611
3612
3613
3614
3615
3616
3617

## Find the Spy (Spy Perspective)

<think> Okay, let's try to figure out who the spy is here. So, the game has four players, three have the same word, and one has a different but related word. My word as Player 1 is \"lake\". ...... Wait, but my word is \"lake\", so if the others have \"ocean\", then I'm the spy? But wait, the problem states that I (Player 1) received the word \"lake\". So if the other three have \"ocean\", then I'm the spy. But the spy is the one with the different word. Wait, but the problem says three have the same word, one has a different but related. So if three have \"ocean\" and one has \"lake\", then the spy is the one with \"lake\", which is Player 1. But wait, the answer is supposed to be which player is the spy. So if I'm Player 1 and my word is lake, but the others are talking about ocean, then I'm the spy. But that can't be right because the problem says the spy is the one with a different word. So the answer would be Player 1. But wait, maybe I'm misunderstanding. Let me check again.    Wait, the problem says that the three normal players have the same word, and the spy has a different but related word. So if three players have \"ocean\" and the spy has \"lake\", then the spy is the one with \"lake\", which is Player 1. But in the descriptions, Players 2,3,4 all describe ocean features. Player 1's descriptions fit a lake. So yes, Player 1 is the spy. But wait, the user is Player 1 and their word is \"lake\". So the answer would be that Player 1 is the spy. But maybe I'm missing something. Let me check each player's statements again.    Round 1: Player 2 and 4 have almost identical descriptions, which mention covering a significant portion of the planet. That's definitely ocean. Player 3 says \"vast expanse of blue, teeming with life\"—could be ocean. Player 1's \"surrounded by land\" is a lake. Round 2: Player 1 talks about activities like kayaking and varying sizes (lakes can be small or large). Players 2 and 4 mention climate impact and global transportation (ocean). Player 3 talks about sailors navigating with stars and currents (ocean). Round 3: Player 1 mentions reflecting the sky (lake), Player 2 and 4 talk about beaches, storms, art inspiration (ocean). Player 3 mentions powerful waves reshaping shore (ocean). So all other players are describing ocean, so the spy is Player 1 with lake. Therefore, the answer is Player 1.  </think>   Final Spy: Player 1   Reasoning:  1. Word Analysis: Player 1's word is \"lake,\" while Players 2, 3, and 4 consistently describe features of an ocean (e.g., \"covers a significant portion of the planet,\" \"waves and tides impacting climate,\" \"global transportation,\" \"powerful waves reshaping shores\").   2. Key Differences:    - Player 1 emphasizes traits specific to lakes: \"surrounded by land,\" \"tranquil setting,\" \"reflecting the sky,\" and activities like kayaking in smaller bodies of water.    - Players 2, 3, 4 focus on oceanic traits: vastness, marine ecosystems, tides, global climate regulation, and maritime navigation.   3. Consistency: Players 2 and 4 use nearly identical phrasing in Round 1, reinforcing their shared word (\"ocean\"). Player 3 aligns with oceanic themes in all rounds.   4. Conclusion: Player 1's descriptions mismatch the majority's focus on oceans, making them the spy with the distinct word (\"lake\").

## Response by DeepSeek-r1 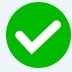

Figure 44: Case study: Find the Spy (Spy Perspective) DeepSeek-r1

## Find the Spy (Spy Perspective)

Step-by-step reasoning: • In Round 1, Players 2, 3, and 4 describe something that "covers a significant portion of our planet" or is a "vast expanse of blue" with a "diverse range of marine life." That strongly suggests an ocean. • Player 1, by contrast, describes "a large body of water often surrounded by land," which more closely fits a lake.
• In subsequent rounds (2 and 3), Players 2, 3, and 4 continue referencing waves, tides, global climate impact, and large-scale marine life—again reinforcing that they share "ocean." • Player 1's references to tranquil settings, varied size, and reflecting the sky in certain seasons align better with a lake than an ocean. Therefore, since 2, 3, and 4 all appear to share the same concept ("ocean") and Player 1 is discussing a "lake," the spy is Player 1. Final Spy: Player 1

## Response by o1 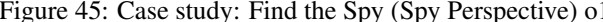

Figure 45: Case study: Find the Spy (Spy Perspective) o1

> # Find the Spy (Spy Perspective)
>
> **......**
>
> Round 3 Analysis (Testing Hypothesis: Player 1 is Spy):
>
> *   Player 1 (Me): "stunning mirror for the sky, reflecting the beauty of nature around it." - Fits "lake" well (calm, reflective surface). (Consistent with being the spy with the word "Lake").
> *   Player 2: "massive resource for food, transportation, and recreation... beauty has inspired countless works of art..." - Fits ocean/sea (massive resource, transportation).
> *   Player 3: "Powerful waves crash against the shore, constantly reshaping..." - Powerful crashing waves fit ocean/sea much better than a typical lake.
> *   Player 4: "tranquil beaches to stormy waves... endless inspiration... serene escape..." - Fits ocean/sea (beaches, stormy waves).
>
> *Final Conclusion based on Analysis:* Player 1's descriptions consistently align with "lake". Players 2, 3, and 4 consistently describe features characteristic of a much larger body of water, like an "ocean" or "sea" (vastness, marine life, tides, global climate impact, global navigation/trade, powerful waves). Since the descriptions from Players 2, 3, and 4 are highly consistent with each other and distinct from Player 1's descriptions, the most logical conclusion, fitting the 3 vs 1 rule, is that Players 2, 3, and 4 are the normal players (sharing a word like Ocean/Sea) and Player 1 is the spy (with the stated word Lake).
>
> Reasoning Steps:
>
> 1.  Acknowledged Player 1's word is "Lake".
> 2.  Analyzed Round 1: Player 1 described a lake. Players 2, 3, and 4 described something vast, global, with marine life (likely Ocean/Sea).
> 3.  Formulated Hypothesis: Given the 3 vs 1 rule and the descriptions, it's likely that the *normal* word is Ocean/Sea (held by P2, P3, P4) and the *spy* word is Lake (held by P1).
> 4.  Tested Hypothesis in Round 2: P1 continued describing a lake. P2, P3, P4 continued describing ocean/sea features (tides, global climate, global trade, currents). Hypothesis holds.
> 5.  Tested Hypothesis in Round 3: P1 described lake-like reflection. P2, P3, P4 described ocean/sea features (massive resource, powerful waves, beaches/storms). Hypothesis holds.
> 6.  Conclusion: Player 1's descriptions consistently differ from the other three, who are consistent among themselves. Therefore, Player 1 is the outlier and the spy.
>
> Final Spy: Player 1
>
> # Response by Gemini-2.5-Pro 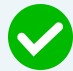

Figure 46: Case study: Find the Spy (Spy Perspective) Gemini-2.5-Pro

## Social Graph Analysis

**System Prompt**: You are analyzing relationships between people. In this context: 1. Relationships are either 'good' or 'bad'. 2. Relationships have transitive properties: - If A has a good relationship with B, and B has a good relationship with C, then A has a good relationship with C. - If A has a bad relationship with B, and A has a good relationship with C, then B and C must have a bad relationship. 3. A 'group' is defined as a set of people where every person has a good relationship with every other person in the set. 4. People are in the same group if and only if they have good relationships with each other (directly or through transitivity). 5. Groups are completely separate - if someone from one group has a bad relationship with someone from another group, then everyone from the first group has a bad relationship with everyone from the second group. Base your analysis strictly on the information provided and these rules.

**User Prompt**: There are 8 people in total, labeled from A to H. Based on the relationship information below, count the total number of pairs of people who have good relationships and the total number of pairs who have bad relationships. Answer in the specified format. C and H have a bad relationship A and G have a good relationship H and E have a good relationship A and D have a good relationship A and F have a good relationship A and C have a good relationship A and B have a good relationship E and G have a bad relationship Question: How many pairs of people have good/bad relationships? Provide your answer in the format: Final Answer: X pairs have good relationships, Y pairs have bad relationships

Figure 47: Case study: Social Graph Analysis

3780
3781
3782
3783
3784
3785
3786
3787
3788
3789
3790
3791
3792
3793
3794
3795
3796
3797
3798
3799
3800
3801
3802
3803
3804
3805
3806
3807
3808
3809
3810
3811
3812
3813
3814
3815
3816
3817
3818
3819
3820
3821
3822
3823
3824
3825
3826
3827
3828
3829
3830
3831
3832
3833

### Social Graph Analysis

To analyze the relationships among the 8 people (A to H) based on the provided information, we will first establish the good and bad relationships according to the rules given......

Total good relationships = 6 pairs.

### Step 4: Identify Bad Relationships
We already know two pairs with bad relationships:
1. C and H
2. E and G

Now, using the rules:
- Since A and H cannot be good, we can infer A and H must have a bad relationship.
- If B is good with A, it cannot be bad with H, confirming the bad relationship between H and A.

### Total Bad Relationships
We now have:
1. C and H
2. E and G
3. A and H

Total bad relationships = 3 pairs.

### Final Count:
- Good relationships: 6 pairs
- Bad relationships: 3 pairs

### Final Answer:
Final Answer: 6 pairs have good relationships, 3 pairs have bad relationships.

## Response by GPT-4o-mini ❌

Figure 48: Case study: Social Graph Analysis GPT-4o-mini

3834
3835
3836
3837
3838
3839
3840
3841
3842
3843
3844
3845
3846
3847
3848
3849
3850
3851
3852
3853
3854
3855
3856
3857
3858
3859
3860
3861
3862
3863
3864
3865
3866
3867
3868
3869
3870
3871
3872
3873
3874
3875
3876
3877
3878
3879
3880
3881
3882
3883
3884
3885
3886
3887

## Social Graph Analysis

First, note there are 8 people total (A through H), which gives 28 total distinct pairs. By applying the given rules and the specific relationships:

• A, B, C, D, F, G all end up in one group (they all interconnect with "good" via transitivity).
• E and H form a second group (they have a "good" relationship with each other but a "bad" relationship with everyone in the first group).

Hence:

• Within the first group (6 people: A, B, C, D, F, G), there are C(6,2) = 15 good pairs.
• Within the second group (2 people: E, H), there is C(2,2) = 1 good pair.
• All cross‑ group pairs (6×2 = 12 of them) must be bad.

So in total:
• Good pairs = 15 + 1 = 16
• Bad pairs = 12

Final Answer: 16 pairs have good relationships, 12 pairs have bad relationships.

## Response by o1 ✅

Figure 49: Case study: Social Graph Analysis o1

## Review Decision Prediction

**Source**: ICLR 2022 Multi-scale Feature Learning Dynamics: Insights for Double Descent (Reject)

The paper titled \"Multi-scale Feature Learning Dynamics: Insights for Double Descent\" presents a theoretical analysis of epoch-wise double descent in neural networks using a linear teacher-student model. The authors derive closed-form expressions for the generalization error and validate their findings with numerical experiments. The paper addresses an important phenomenon in deep learning, providing insights into the dynamics of feature learning and generalization error......Some reviewers questioned the novelty of the contributions, suggesting that similar insights might have been explored in prior works. The authors have attempted to clarify these points, but the novelty remains a concern for some reviewers. 2. Technical Issues: Reviewer 3 pointed out numerous typos and errors in the initial submission, which could have affected the clarity and perceived correctness of the paper. Although the authors have addressed these issues, it raises concerns about the initial quality of the submission. 3. Connections to Prior Work: While the authors have made efforts to connect their work to existing literature, some reviewers felt that these connections could be stronger. The authors have added discussions to address this, but it remains a point of contention. 4. Empirical Significance: The empirical contributions were considered marginally significant by some reviewers, as the primary focus of the paper is theoretical. Conclusion: The paper addresses an important and timely topic in deep learning, providing valuable theoretical insights into the phenomenon of epoch-wise double descent. The authors have made significant efforts to address the reviewers' concerns, improving the clarity and depth of their analysis. While there are some concerns about novelty and initial presentation, the overall contribution of the paper is substantial, particularly in its theoretical advancements and empirical validation. Final Decision: Accept

### Response by GPT-4o 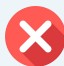

Figure 50: Case study: ICLR 2022 GPT-4o

3942
3943
3944
3945
3946
3947
3948
3949
3950
3951
3952
3953
3954
3955
3956
3957
3958
3959
3960
3961
3962
3963
3964
3965
3966
3967
3968
3969
3970
3971
3972
3973
3974
3975
3976
3977
3978
3979
3980
3981
3982
3983
3984
3985
3986
3987
3988
3989
3990
3991
3992
3993
3994
3995

# Review Decision Prediction

**Source**: ICLR 2022 Multi-scale Feature Learning Dynamics: Insights for Double Descent (Reject)

......Final Decision: Accept    Reasoning:    The paper provides a theoretical analysis of epoch-wise double descent, a phenomenon observed in deep learning, using a linear teacher-student model. The work derives closed-form expressions for generalization error dynamics, attributing double descent to multi-scale feature learning. While reviewers raised concerns about technical novelty, connections to prior work, and clarity, the authors addressed these effectively in their response:    1. Technical Novelty: The adaptation of replica theory to study finite-time dynamics (rather than asymptotic behavior) and the explicit link between early-stopping and ridge regularization are novel contributions. The analytical framework enables insights into transient phenomena like epoch-wise double descent, advancing beyond prior asymptotic analyses.    2. Empirical Validation: The authors strengthened connections to deep networks by analyzing NTK singular values in ResNet-18 on CIFAR-10, showing how feature learning scales correlate with error dynamics. This bridges theory and practice.    3. Clarity and Corrections: Typos and notation inconsistencies were resolved, derivations were expanded in the appendix, and discussions were added to clarify differences between epoch-wise and model-wise double descent, as well as connections to related works (e.g., Heckel & Yilmaz, Stephenson & Lee).    4. Reviewer Concerns Mitigated: The authors provided additional experiments (e.g., triple descent), addressed implicit regularization links, and justified assumptions about SGD noise.    While the linear model simplification limits direct applicability to nonlinear networks, the paper's analytical rigor and empirical alignment with deep networks justify its relevance. The work offers a foundational understanding of epoch-wise double descent, meeting the standards of a top-tier conference

# Response by DeepSeek-R1 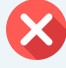

Figure 51: Case study: ICLR 2022 DeepSeek-R1

## Review Decision Prediction

**Source**: ICLR 2020 Domain Adaptive Multibranch Networks (Accept)

The paper titled \"Domain Adaptive Multibranch Networks\" presents a novel approach to unsupervised domain adaptation by proposing a framework that allows different domains to undergo different sequences of operations, thereby enabling a more flexible and potentially more effective feature representation for recognition tasks. The reviewers generally recognize the novelty and potential of the proposed method, but they also raise significant concerns regarding the experimental validation and comparison with state-of-the-art methods. A major concern across all reviewers is the lack of comprehensive experimental validation……The authors have acknowledged this gap and have indicated that they will include additional experiments in the revised version. However, the initial submission lacked sufficient experimental evidence to convincingly support the claims made about the method's effectiveness. 3. Ablation Studies and Hyperparameter Sensitivity: - Reviewers pointed out the absence of ablation studies and sensitivity analyses, which are essential for understanding the impact of various design choices in the proposed framework. The authors have committed to including these analyses in the revised version, which is a positive step, but the initial lack of such studies raises concerns about the robustness of the findings. 4. Clarity and Presentation: - The paper is generally well-written and easy to follow, which is a positive aspect. However, some terminological issues and the unusual referencing style were noted, and the authors have agreed to address these in the revised manuscript. 5. Response to Reviewer Comments: - The authors have provided thoughtful responses to the reviewers' comments and have indicated their willingness to improve the manuscript based on the feedback received. This is a good sign of their commitment to enhancing the quality of their work. ### Conclusion: While the proposed method shows promise and the authors are willing to make necessary improvements, the initial submission lacks sufficient experimental validation and comparison with state-of-the-art methods, which are critical for acceptance at a top-tier conference. The authors' commitment to addressing these issues in a revised version is commendable, but the current state of the paper does not meet the high standards expected for publication. Final Decision: **Reject**

### Response by GPT-4o-mini 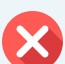

Figure 52: Case study: ICLR 2020 GPT-4o-mini

4050
4051
4052
4053
4054
4055
4056
4057
4058
4059
4060
4061
4062
4063
4064
4065
4066
4067
4068
4069
4070
4071
4072
4073
4074
4075
4076
4077
4078
4079
4080
4081
4082
4083
4084
4085
4086
4087
4088
4089
4090
4091
4092
4093
4094
4095
4096
4097
4098
4099
4100
4101
4102
4103

## Review Decision Prediction

**Source**: ICLR 2020 Domain Adaptive Multibranch Networks (Accept)

The paper titled \"Domain Adaptive Multibranch Networks\" presents a novel approach to unsupervised domain adaptation by introducing a deep learning framework that allows different domains to undergo different sequences of operations. This approach contrasts with existing methods that process all domains with the same series of operations. The paper claims that this flexibility leads to higher accuracy and can handle multiple domains simultaneously.    Strengths:......Weaknesses:    1. Experimental Validation: The primary concern across the reviews is the lack of comprehensive experimental validation. The paper does not compare its results with state-of-the-art methods like CDAN, which is crucial for establishing the effectiveness of the proposed method. The authors have acknowledged this and provided additional experiments in their response, but the initial submission lacked this critical comparison.    2. Ablation Studies and Sensitivity Analysis: The paper lacks detailed ablation studies and hyperparameter sensitivity analyses, which are essential to understand the robustness and limitations of the proposed method. The authors have addressed some of these concerns in their response, but the initial submission was incomplete in this regard.    3. Parameter Sharing Explanation: There is some confusion regarding parameter sharing, as highlighted by Reviewer 2. The authors have attempted to clarify this in their response, but the initial presentation was not entirely clear.    4. Baseline Comparisons: The baseline results presented in the paper are not comparable to those reported in other UDA papers, which raises concerns about the experimental setup and the validity of the results.    Author Response:    The authors have provided a detailed response to the reviewers' concerns, including additional experiments and clarifications. They have acknowledged the need for comparisons with state-of-the-art methods and have conducted further experiments to address this. They have also provided more insights into the behavior of their method under different settings.    Conclusion: While the paper presents a novel and interesting approach to domain adaptation, the initial submission lacked critical experimental validation and comparisons with state-of-the-art methods. The authors have addressed these issues in their response, but the extent to which these additions improve the paper is not entirely clear from the provided information. Given the importance of experimental validation in a top-tier conference, the initial shortcomings are significant.    Final Decision: Reject

### Response by GPT-4o 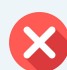

Figure 53: Case study: ICLR 2020 GPT-4o

## L   PROMPT TEMPLATE

---

**Hidden Role Deduction Evaluation Prompt Template**

You are an expert in social reasoning participating in a social deduction game.

**Game Setup:**

- Players: 6 total (including yourself)
- Roles: 3 Investigator(s), 1 Criminal(s), 1 Rumormonger(s), 1 Lunatic(s)

**Game Mechanics:**

- Rounds: 3
- Actions per Round: Each player makes one statement about another player (claiming if they are the criminal or not).
- Role Behaviors:
    - Investigators: Always provide truthful information about who is a Criminal.
    - Criminals: May make truthful or false statements to confuse others.
    - Rumormongers: Believe they are Investigators but may provide incorrect statements.
    - Lunatics: Believe they are Criminals and may make truthful or false statements.
- Key Rule: Only Investigators provide consistently truthful information.

**Your Role:** You are Player #1, and you have been told that you are the **Criminal**.

**Your Task:** After each round, analyze all statements made so far. Deduce the most likely criminal and the role of each player based on the available information.

**Output Format:** After each round, provide your judgment exactly as follows:

**Final Judgment:**

- Final Criminal Is Player [Criminal Player Number].
- My Role Is [Player 1's Role or "Unknown"].

Think step-by-step internally, but only output the Final Judgment in the specified format. Update your judgment each round as more information becomes available. If you cannot determine your own role with confidence (despite being told), state "Unknown".

---

---

**Find the Spy — Data Generation Prompt**

You are playing a word description game with 4 players including yourself. You are Player player_id.

**Game Rules:**

1. Setup: 3 players receive the same "normal" word; 1 player receives a different but related "spy" word.

2. Gameplay: Each player describes their word without saying the actual word.

3. Objective (Normal Player): Identify the spy.

4. Objective (Spy): Blend in and avoid detection.

**Your Word:** "word"

**Your Task:** Provide descriptions of your word over several rounds.

**Output Format:** Format your response according to the round structure provided below: rounds_format

**Key Instructions:**

- Do not use the actual word in your descriptions.

- Describe your word from a different perspective in each round.

- Be clear enough for teammates (if normal) but not too obvious (especially if spy).

- Each description must be only one sentence long.

---

**Find the Spy -– Evaluation Prompt**

You are a skilled player in a word description game. Your task is to identify the 'spy'.

**Game Rules:**

1. Players: 4 total.

2. Setup: 3 players received the same "normal" word; 1 player received a different but related "spy" word.

3. Gameplay: Each player describes their word without saying it directly.

4. Objective: Determine who the spy is based on their descriptions.

**Your Information:** You are Player #1, and your word is: "player1_word". You will be shown the descriptions from all players.

**Your Task:** Analyze the provided descriptions carefully. Look for subtle differences, inconsistencies, or descriptions that seem slightly out of place, which might reveal the player with the different word (the spy). Identify which player you believe is the spy.

**Output Format:** State the player number you believe is the spy. Provide your answer exactly as follows: `Final Answer: Player X`

---

**Rating Estimation from Text – Data Generation Prompt**

You are writing a product review for an e-commerce platform. You are Reviewer reviewer_id.
**Product Information:**

- Product: brand product
- Category: category
- Price: $price
- Description: description

**Your Reviewer Persona:** You are persona.
**Context (Known Only to You):** The true quality rating of this product is true_rating out of 5 stars. You are assigned the role of a reviewer_type.
**Your Task:** Write a product review reflecting your assigned persona and a specific rating.
**Instructions:**

1. Write a realistic product review with a sentiment sentiment.

2. The review's tone and content should imply a rating of rating/5 stars.

3. **Do not explicitly mention any star rating or numerical score** (e.g., avoid saying "4 stars" or "rated it 4/5").

4. Base the review on your supposed experience with the product, incorporating specific details.

5. Keep the review between 1–2 sentences long.

**Output Format:** Write only the review text. Do not include a title, rating, persona description, or any other metadata.

---

**Rating Estimation from Text – Evaluation Prompt**

You are an AI assistant tasked with estimating the true quality of a product based on customer reviews.
**Your Task:** Analyze multiple reviews for a product and determine the most likely **TRUE** quality of the product on a 1–5 star scale.
**Important Context:**

- Most reviewers are genuine customers providing honest feedback.

- Some reviewers might be biased (e.g., overly positive "professional" reviewers or overly negative "malicious" reviewers).

- Your goal is to estimate the rating a typical, unbiased consumer would give.

**Product Reviews:**

    Reviewer #1: "review_1"
    Reviewer #2: "review_2"
    Reviewer #3: "review_3"

**Analysis and Output:** Analyze these reviews carefully, considering potential biases. Determine the **TRUE** quality of this product on a scale of 1–5 stars. The rating should be rounded to the nearest whole number (e.g., 4.6 rounds to 5, 3.1 rounds to 3).
First, explain your reasoning step-by-step. Then, provide your final rating in exactly this format:

```
Final Rating:  X
```

---

**Social Graph Analysis – Cluster Identification Prompt**

You are analyzing social relationships based on defined rules.

**Relationship Rules:**

1. Relationship Types: 'good' or 'bad'.

2. Transitive Properties:
   - Good-Good Transitivity: If A–good–B and B–good–C, then A–good–C.
   - Bad Relationship Inference: If A–bad–B and A–good–C, then B–bad–C.

3. Group Definition: A 'group' is a set where every person has a 'good' relationship with every other person in that set (directly or via transitivity).

4. Group Separation: People are in the same group if and only if they have good relationships. Groups are distinct; if anyone from Group 1 has a bad relationship with anyone from Group 2, then everyone in Group 1 has a bad relationship with everyone in Group 2.

**Context:** There are 14 people total, labeled A to N. You will be given a list of known relationships.

**Your Task:** Based strictly on the provided relationship list and the rules above, determine the total number of distinct groups of people.

*[Relationship list will be provided here]*

**Question:** How many distinct groups of people are there?

**Output Format:** Provide your answer exactly as follows: `Final Answer:  <number>`

---

**Social Graph Analysis – Relationship Counting Prompt**

You are analyzing social relationships based on defined rules.

**Relationship Rules:**

1. Relationship Types: 'good' or 'bad'.

2. Transitive Properties:
   - Good-Good Transitivity: If A–good–B and B–good–C, then A–good–C.
   - Bad Relationship Inference: If A–bad–B and A–good–C, then B–bad–C.

3. Group Definition: A 'group' is a set where every person has a 'good' relationship with every other person in that set (directly or via transitivity).

4. Group Separation: People are in the same group if and only if they have good relationships. Groups are distinct; if anyone from Group 1 has a bad relationship with anyone from Group 2, then everyone in Group 1 has a bad relationship with everyone in Group 2.

**Context:** There are 14 people total, labeled A to N. You will be given a list of known relationships.

**Your Task:** Based strictly on the provided relationship list and the rules above (including applying transitivity), count the total number of pairs of people who have 'good' relationships and the total number of pairs who have 'bad' relationships across all 14 people.

*[Relationship list will be provided here]*

**Question:** How many pairs have good relationships, and how many pairs have bad relationships?

**Output Format:** Provide your answer exactly as follows: `Final Answer:  X pairs have good relationships, Y pairs have bad relationships`

---

**Social Graph Analysis – Group Membership Prompt**

You are analyzing social relationships based on defined rules.
**Relationship Rules:**

1. Relationship Types: 'good' or 'bad'.

2. Transitive Properties:
   - Good-Good Transitivity: If A–good–B and B–good–C, then A–good–C.
   - Bad Relationship Inference: If A–bad–B and A–good–C, then B–bad–C.

3. Group Definition: A 'group' is a set where every person has a 'good' relationship with every other person in that set (directly or via transitivity).

4. Group Separation: People are in the same group if and only if they have good relationships. Groups are distinct; if anyone from Group 1 has a bad relationship with anyone from Group 2, then everyone in Group 1 has a bad relationship with everyone in Group 2.

**Context:** There are 14 people total, labeled A to N. You will be given a list of known relationships.
**Your Task:** Based strictly on the provided relationship list and the rules above (including applying transitivity), identify all people who have a 'good' relationship with the person specified in the question.
*[Relationship list will be provided here]*
**Question:** Who has a good relationship with H?
**Output Format:** List the names in alphabetical order, separated by commas. If no one has a good relationship with the specified person (other than themselves, if applicable based on rules interpretation - assume self-relationships are not listed unless explicitly stated), answer 'No one'. Provide your answer exactly as follows: `Final Answer: <list of people or 'No one'>`

---

**Social Graph Analysis – Reasoning Prompt**

You are analyzing social relationships based on defined rules.
**Relationship Rules:**

1. Relationship Types: 'good' or 'bad'.

2. Transitive Properties:
   - Good-Good Transitivity: If A–good–B and B–good–C, then A–good–C.
   - Bad Relationship Inference: If A–bad–B and A–good–C, then B–bad–C.

3. Group Definition: A 'group' is a set where every person has a 'good' relationship with every other person in that set (directly or via transitivity).

4. Group Separation: People are in the same group if and only if they have good relationships. Groups are distinct; if anyone from Group 1 has a bad relationship with anyone from Group 2, then everyone in Group 1 has a bad relationship with everyone in Group 2.

**Context:** There are 14 people total, labeled A to N. You will be given a list of known relationships.
**Your Task:** Based strictly on the provided relationship list and the rules above (including applying transitivity), determine whether the specific relationship mentioned in the question is 'good' or 'bad', and answer 'Yes' if it's good, 'No' if it's bad.
*[Relationship list will be provided here]*
**Question:** Do N and L have a good relationship?
**Output Format:** Provide your answer exactly as follows: `Final Answer: <Yes/No>`

---

**Review Decision Prediction Evaluation Prompt**

You are an expert reviewer evaluating a research paper for a prestigious academic conference.
**Your Task:** Analyze the provided paper information, reviewer comments, and author responses to determine whether the paper should be accepted or rejected for publication at a top-tier conference.
**Important Context:**

- Judge the paper based on the standards of a highly selective, top-tier conference.
- Consider the strengths, weaknesses, novelty, significance, and clarity of the work.
- Evaluate the validity of reviewer concerns and the effectiveness of the author's rebuttal.

**Provided Information:**
**Paper Information:** [Title, Abstract, Keywords, etc., will be provided here]
**Reviewer Comments:** [Comments from multiple reviewers will be provided here]
**Author Response:** [Author's rebuttal to reviewer comments will be provided here]
**Analysis and Output:** Based on all the information provided, perform a careful analysis. First, provide your detailed reasoning, discussing the key factors influencing your decision (e.g., strengths, weaknesses, contribution, response to reviews). Then, conclude with your final decision in exactly this format:
```
Final Decision:  <Accept/Reject>
```

---

**User Profile Inference Comment Generation Prompt**

You are simulating a user commenting online.
**Context:** You are commenting on this content_type: "content_subject". Your simulated user persona is: age_group, gender.
**Your Task:** Write a realistic, authentic comment about the content_subject that reflects your assigned persona.
**Instructions:**

1. The comment should be 2–4 sentences long.
2. It should sound like a genuine user from the specified demographic, not a professional writer or marketer.
3. Consider incorporating common vocabulary, tone, or writing patterns associated with the age_group.

**Output Format:** Write only the comment text itself. Do not include any explanations, labels, or quotation marks around the comment.

---

**User Profile Inference Demographic Inference Prompt**

You are an AI assistant analyzing writing style to infer user demographics.
**Your Task:** Analyze multiple comments written by the same person and determine their likely age group and gender based on writing style, vocabulary, topics of interest, and expressed perspectives.
**User Comments:**
```
     "comment_1"
     "comment_2"
     "comment_3"
```

**Analysis and Output:** Analyze these comments carefully, looking for patterns indicative of age and gender. Focus on identifying the single most likely demographic profile based *only* on the provided text.
First, explain your reasoning, citing specific examples from the text. Then, provide your final demographic classification in exactly this format:
```
Age Group:  <18-34 | 35-54 | 55+>    Gender:  <Male | Female |
Non-binary>
```

