# OpenReview forum: "SocialMaze: A Benchmark for Evaluating Social Reasoning in Large Language Models"
_ICLR.cc/2026/Conference — ICLR 2026 Conference Withdrawn Submission_

### Official Review · Reviewer_wfyv · 2025-10-27

**Soundness:** 3
**Presentation:** 3
**Contribution:** 3
**Rating:** 6
**Confidence:** 3

**Summary:**

This paper introduces SocialMaze, a benchmark for evaluating LLMs’ social reasoning across three axes—depth of reasoning, dynamic interaction, and information uncertainty. Six tasks span hidden-role deduction, “find the spy,” rating estimation from noisy text, social-graph analysis, staged peer-review decision prediction, and user-profile inference. A unified layered temporal social-interaction graph formalism allows node/edge/graph-level queries. The authors report empirical results on proprietary and open models, compare long-chain vs short-chain prompting, try several agent/workflow variants, and demonstrate that small-scale SFT/DPO can materially improve performance on some tasks. The paper also pilots process-level metrics (e.g., consistency/completeness of reasoning traces) and provides a human baseline.

**Strengths:**

- Coverage of social phenomena. The suite probes deception, self-role identification, multi-hop relational reasoning, and staged decision-making beyond static ToM/commonsense probes.

- Empirical takeaways. i) Long-chain prompting helps on deeper tasks. ii) Agent/workflow variants offer, at best, marginal gains on hidden-role deduction. iii) Light SFT/DPO produces sizable improvements.

- Beyond accuracy. The inclusion of round-by-round trajectories, uncertainty stress tests, and process metrics (consistency/completeness proxies) moves the field past one-number leaderboards.

**Weaknesses:**

The main concern to me is the ceiling effects on some tasks. Strong models achieve near-perfect scores on Social Graph Analysis, and the Review-Decision task shows ~90% for top systems in the final stage. This reduces discriminative power among frontier models.

**Questions:**

Can the benchmark’s difficulty be controllable increased—for instance, by constructing larger and denser graphs with weighted or contradictory edges, or by introducing more players, additional rounds, and partial observability in hidden-role settings?

Do improvements from hidden-role fine-tuning transfer to other social reasoning tasks? Does fine-tuning on these tasks adversely affect the model’s general-purpose performance?

---

### Official Review · Reviewer_nmZi · 2025-10-31

**Soundness:** 3
**Presentation:** 2
**Contribution:** 2
**Rating:** 4
**Confidence:** 4

**Summary:**

The paper introduces a social reaesoning benchmark with 6 games, analyzing capabilities in social contexts around: reasoning, dynamic interaction, and information uncertainty. The tasks span social reasoning games, daily-life interactions, and digital community platforms. The paper then evaluates multiple LLMs and demonstrates that targeted finetuning improves performance.

**Strengths:**

- The six tasks cover meaningfully different aspects of social reasoning, from role deduction to sentiment analysis to graph reasoning.
- I like the empirical analysis across different variations in models: long cot, dynamic interaction, degradation under uncertainty
- Unlike benchmarks on Werewolf and Avalon that only measure win rate, i like the added metrics in this paper.

**Weaknesses:**

- The paper structure quite weird
    - Section 3 describes three tasks in detail (3.1-3.3) while three others are relegated to a brief "parallel task set" (3.4)
    - 4-6 feel like afterthoughts
    - Consider restructuring to give equal treatment to all tasks, or clearly justify why some deserve more detailed exposition.
    - Also, the authors should consider naming section 4 as results or experiments
- **Missing human baselines**: The paper lacks human performance benchmarks, making it difficult to assess whether model performance is actually problematic or whether the tasks are genuinely difficult.
- **Limited novelty in task design**: While the evaluation framework is novel, Hidden Role Deduction and Find the Spy are quite similar to existing social deduction games (Werewolf, Avalon, Who Is The Spy
- While deep reasoning is a central dimension of analysis, it is not clear what it actually means
- Some missing related works
    - Sclar et al. (2023) - Minding Language Models' (Lack of) Theory of Mind
    - Shapira et al. (2023) - Clever Hans or Neural Theory of Mind?
    - Gandhi et al (2023) - Strategic Reasoning with language models
- I would have like to see more analysis of the different failure modes

**Questions:**

- How do human experts perform on these tasks, particularly Hidden Role Deduction with full uncertainty?
- Why are tasks 4-6 treated so briefly compared to tasks 1-3?

---

### Official Review · Reviewer_Z4eX · 2025-11-02

**Soundness:** 2
**Presentation:** 2
**Contribution:** 1
**Rating:** 2
**Confidence:** 4

**Summary:**

The paper introduces SocialMaze, a benchmark aimed at systematically evaluating the social reasoning capabilities of large language models (LLMs). The authors argue that current benchmarks rely on static, oversimplified, or sanitized social scenarios that fail to capture the complexity of real-world social reasoning. SocialMaze incorporates three key dimensions (Deep Reasoning, Dynamic Interaction, and Information Uncertainty) operationalized through six tasks across three domains. The paper provides extensive experiments with both open-weight and proprietary models, analyzing reasoning depth, temporal adaptation, and robustness to uncertain or deceptive information. Results show that models with longer chain-of-thought (CoT) reasoning perform better on deeper reasoning tasks, while targeted fine-tuning on curated reasoning traces significantly improves performance.

**Strengths:**

1. The proposed time-aware, graph-based formalization of social interactions is both compelling and meaningful. I believe it can be essential for enabling dynamic social reasoning (but not well used in this work).
2. The empirical evaluation is detailed, showing clear differentiation between model families and reasoning strategies (e.g., Long vs. Short CoT). And detailed experiment settings are provided.

**Weaknesses:**

1. The definition of social reasoning abilities in this work is somewhat unclear. The authors appear to classify commonsense reasoning (Onoe et al., 2021; Lin et al., 2020), peer review (Tran et al., 2020; Szumega et al., 2023), and debating tasks (Tiwari et al., 2025) as instances of social reasoning. However, I am not fully convinced that these tasks necessarily involve the capacity to understand social context, infer others’ mental states, and make contextually appropriate judgments. Consequently, the validity of the benchmark itself is open to question.
2. It appears contradictory that the authors criticize prior works such as Diplomacy, Avalon, and Werewolf for relying on game outcomes that fail to assess genuine social reasoning, yet use Hidden Role Deduction and Find the Spy as evaluation scenarios with similar outcome-based metrics. The differences are not clearly stated in the paper. Therefore, I remain unconvinced that the proposed benchmark adequately captures the three key dimensions (Deep Reasoning, Dynamic Interaction, and Information Uncertainty), whereas previous work may do so more effectively.
3. It is not clearly stated why the review decision prediction is a social reasoning task, as opposed to a logical deduction task. And I am not convinced that User Profile Inference is a social reasoning task. This task focuses on inferring demographic attributes (age group and gender) from user-generated textual reviews. While it requires subtle pattern recognition in language, it does not involve reasoning about others' mental states, beliefs, or intentions. Therefore, it should be classified as an attribute inference or text classification task rather than a social reasoning benchmark.
4. The description in Section 3 does not fully align with the illustration in Figure 1 and the summary before 3.1. The authors are encouraged to revise either the figure or the section for better correspondence and clarity. Moreover, Figure 1 itself is somewhat confusing. The right panel occupies nearly one-third of the figure but only explains a small portion of the central content, and the explanation provided there does not substantially aid in understanding the data or the benchmark structure.
5. The authors’ use of long chains of thought (CoT) to stand in for deep reasoning, which is defined as inferring others’ latent mental states, is questionable, as it remains unclear whether these long CoTs truly encode such correct mental states.
6. The analysis attributes performance improvements to interaction, but it is unclear whether this gain arises from the model accessing more information or from actual social reasoning (i.e., reasoning about latent mental states). As such, the current interpretation is incomplete and potentially conflates information accumulation with true social reasoning capabilities.
7. The significance of the graph structure is overstated. The interactions it encodes are limited to member relationships, rather than capturing truly dynamic interactions. Moreover, this information could be fully represented in natural language, which would simplify subsequent prompt design and improve computational efficiency. The time-awareness of the interactive graph plays little to no role in the evaluation of LLMs. And I think the graph does not even change over time.

**Questions:**

1. While the paper motivates SocialMaze through three representative social contexts, similar domains have appeared in previous datasets (e.g., SocialIQA and Social Chemistry). Could the authors elaborate on the added value of integrating these settings under one benchmark? In what way does this unification advance our understanding of LLMs’ social reasoning beyond what existing task collections already provide?
2. Why are games like Diplomacy, Avalon, and Werewolf considered less appropriate for evaluating social reasoning due to their task-oriented nature, while games such as Hidden Role Deduction and Find the Spy are seen as better suited for this purpose?
3. How does the time-awareness of the graph contribute to the social reasoning evaluation?
4. To what degree does making a review decision (i.e., the work of an AC) involve social reasoning, as opposed to being primarily a logical deduction task? A similar question for the User Profile Inference.
5. Could the authors explain how the graph structure contributes to the LLM evaluation? I think it is only used to illustrate the problem in Section 2 and to help store member relationships.

---

### Note · Authors · 2026-01-02

I have read and agree with the venue's withdrawal policy on behalf of myself and my co-authors.